# Regularized EM Algorithms: A Unified Framework and Statistical Guarantees

**Xinyang Yi**
Dept. of Electrical and Computer Engineering
The University of Texas at Austin
yixy@utexas.edu

**Constantine Caramanis**
Dept. of Electrical and Computer Engineering
The University of Texas at Austin
constantine@utexas.edu

## Abstract

Latent models are a fundamental modeling tool in machine learning applications, but they present significant computational and analytical challenges. The popular EM algorithm and its variants, is a much used algorithmic tool; yet our rigorous understanding of its performance is highly incomplete. Recently, work in [1] has demonstrated that for an important class of problems, EM exhibits linear local convergence. In the high-dimensional setting, however, the $M$-step may not be well defined. We address precisely this setting through a unified treatment using regularization. While regularization for high-dimensional problems is by now well understood, the iterative EM algorithm requires a careful balancing of making progress towards the solution while identifying the right structure (e.g., sparsity or low-rank). In particular, regularizing the $M$-step using the state-of-the-art high-dimensional prescriptions (e.g., à la [19]) is not guaranteed to provide this balance. Our algorithm and analysis are linked in a way that reveals the balance between optimization and statistical errors. We specialize our general framework to sparse gaussian mixture models, high-dimensional mixed regression, and regression with missing variables, obtaining statistical guarantees for each of these examples.

## 1 Introduction

We give general conditions for the convergence of the EM method for high-dimensional estimation. We specialize these conditions to several problems of interest, including high-dimensional sparse and low-rank mixed regression, sparse gaussian mixture models, and regression with missing covariates. As we explain below, the key problem in the high-dimensional setting is the $M$-step. A natural idea is to modify this step via appropriate regularization, yet choosing the appropriate sequence of regularizers is a critical problem. As we know from the theory of regularized M-estimators (e.g., [19]) the regularizer should be chosen proportional to the target estimation error. For EM, however, the target estimation error changes at each step.

The main contribution of our work is technical: we show how to perform this iterative regularization. We show that the regularization sequence must be chosen so that it converges to a quantity controlled by the ultimate estimation error. In existing work, the estimation error is given by the relationship between the population and empirical $M$-step operators, but this too is not well defined in the high-dimensional setting. Thus a key step, related both to our algorithm and its convergence analysis, is obtaining a different characterization of statistical error for the high-dimensional setting.

**Background and Related Work**

EM (e.g., [8, 12]) is a general algorithmic approach for handling latent variable models (including mixtures), popular largely because it is typically computationally highly scalable, and easy to implement. On the flip side, despite a fairly long history of studying EM in theory (e.g., [12, 17, 21]),

very little has been understood about general statistical guarantees until recently. Very recent work in [1] establishes a general local convergence theorem (i.e., assuming initialization lies in a local region around true parameter) and statistical guarantees for EM, which is then specialized to obtain near-optimal rates for several specific *low-dimensional* problems – low-dimensional in the sense of the classical statistical setting where the samples outnumber the dimension. A central challenge in extending EM (and as a corollary, the analysis in [1]) to the high-dimensional regime is the $M$-step. On the algorithm side, the $M$-step will not be stable (or even well-defined in some cases) in the high-dimensional setting. To make matters worse, any analysis that relies on showing that the finite-sample $M$-step is somehow "close" to the $M$-step performed with infinite data (the population-level $M$-step) simply cannot apply in the high-dimensional regime. Recent work in [20] treats high-dimensional EM using a truncated $M$-step. This works in some settings, but also requires specialized treatment for every different setting, precisely because of the difficulty with the $M$-step.

In contrast to work in [20], we pursue a high-dimensional extension via regularization. The central challenge, as mentioned above, is in picking the sequence of regularization coefficients, as this must control the optimization error (related to the special structure of $\boldsymbol{\beta}^*$), as well as the statistical error. Finally, we note that for finite mixture regression, Städler et al.[16] consider an $\ell_1$ regularized EM algorithm for which they develop some asymptotic analysis and oracle inequality. However, this work doesn't establish the theoretical properties of local optima arising from regularized EM. Our work addresses this issue from a local convergence perspective by using a novel choice of regularization.

## 2 Classical EM and Challenges in High Dimensions

The EM algorithm is an iterative algorithm designed to combat the non-convexity of max likelihood due to latent variables. For space concerns we omit the standard derivation, and only give the definitions we need in the sequel. Let $\boldsymbol{Y}, \boldsymbol{Z}$ be random variables taking values in $\mathcal{Y}, \mathcal{Z}$, with joint distribution $f_{\boldsymbol{\beta}}(\mathbf{y}, \mathbf{z})$ depending on model parameter $\boldsymbol{\beta} \subseteq \Omega \subseteq \mathbb{R}^p$. We observe samples of $Y$ but not of the latent variable $Z$. EM seeks to maximize a lower bound on the maximum likelihood function for $\boldsymbol{\beta}$. Letting $\kappa_{\boldsymbol{\beta}}(\mathbf{z}|\mathbf{y})$ denote the conditional distribution of $\boldsymbol{Z}$ given $\boldsymbol{Y} = \mathbf{y}$, letting $y_{\boldsymbol{\beta}^*}(\mathbf{y})$ denote the marginal distribution of $\boldsymbol{Y}$, and defining the function

$$Q_n(\boldsymbol{\beta}'|\boldsymbol{\beta}) := \frac{1}{n} \sum_{i=1}^{n} \int_{\mathcal{Z}} \kappa_{\boldsymbol{\beta}}(\mathbf{z}|\mathbf{y}_i) \log f_{\boldsymbol{\beta}'}(\mathbf{y}_i, \mathbf{z}) d\mathbf{z}, \tag{2.1}$$

one iteration of the EM algorithm, mapping $\boldsymbol{\beta}^{(t)}$ to $\boldsymbol{\beta}^{(t+1)}$, consists of the following two steps:

- E-step: Compute function $Q_n(\boldsymbol{\beta}|\boldsymbol{\beta}^{(t)})$ given $\boldsymbol{\beta}^{(t)}$.
- M-step: $\boldsymbol{\beta}^{(t+1)} \leftarrow \mathcal{M}_n(\boldsymbol{\beta}) := \arg\max_{\boldsymbol{\beta}' \in \Omega} Q_n(\boldsymbol{\beta}'|\boldsymbol{\beta}^{(t)})$.

We can define the population (infinite sample) versions of $Q_n$ and $\mathcal{M}_n$ in a natural manner:

$$Q(\boldsymbol{\beta}'|\boldsymbol{\beta}) := \int_{\mathcal{Y}} y_{\boldsymbol{\beta}^*}(\mathbf{y}) \int_{\mathcal{Z}} \kappa_{\boldsymbol{\beta}}(\mathbf{z}|\mathbf{y}) \log f_{\boldsymbol{\beta}'}(\mathbf{y}, \mathbf{z}) d\mathbf{z}d\mathbf{y} \tag{2.2}$$

$$\mathcal{M}(\boldsymbol{\beta}) = \arg\max_{\boldsymbol{\beta}' \in \Omega} Q(\boldsymbol{\beta}'|\boldsymbol{\beta}). \tag{2.3}$$

This paper is about the high-dimensional setting where the number of samples $n$ may be far less than the dimensionality $p$ of the parameter $\boldsymbol{\beta}$, but where $\boldsymbol{\beta}$ exhibits some special structure, e.g., it may be a sparse vector or a low-rank matrix. In such a setting, the $M$-step of the EM algorithm may be highly problematic. In many settings, for example sparse mixed regression, the $M$-step may not even be well defined. More generally, when $n \ll p$, $\mathcal{M}_n(\boldsymbol{\beta})$ may be far from the population version, $\mathcal{M}(\boldsymbol{\beta})$, and in particular, the minimum estimation error $\|\mathcal{M}_n(\boldsymbol{\beta}^*) - \mathcal{M}(\boldsymbol{\beta}^*)\|$ can be much larger than the signal strength $\|\boldsymbol{\beta}^*\|$. This quantity is used in [1] as well as in follow-up work in [20], as a measure of statistical error. In the high dimensional setting, something else is needed.

## 3 Algorithm

The basis of our algorithm is the by-now well understood concept of regularized high dimensional estimators, where the regularization is tuned to the underlying structure of $\boldsymbol{\beta}^*$, thus defining a regu-

larized $M$-step via

$$\mathcal{M}_n^r(\boldsymbol{\beta}) := \arg\max_{\boldsymbol{\beta}' \in \Omega} Q_n(\boldsymbol{\beta}'|\boldsymbol{\beta}) - \lambda_n \mathcal{R}(\boldsymbol{\beta}'), \tag{3.1}$$

where $\mathcal{R}(\cdot)$ denotes an appropriate regularizer chosen to match the structure of $\boldsymbol{\beta}^*$. The key challenge is how to choose the sequence of regularizers $\{\lambda_n^{(t)}\}$ in the iterative process, so as to control optimization and statistical error. As detailed in Algorithm 1, our sequence of regularizers attempts to match the target estimation error at each step of the EM iteration. For an intuition of what this might look like, consider the estimation error at step $t$: $\|\mathcal{M}_n^r(\boldsymbol{\beta}^{(t)}) - \boldsymbol{\beta}^*\|_2$. By the triangle inequality, we can bound this by a sum of two terms: the optimization error and the final estimation error:

$$\|\mathcal{M}_n^r(\boldsymbol{\beta}^{(t)}) - \boldsymbol{\beta}^*\|_2 \leq \|\mathcal{M}_n^r(\boldsymbol{\beta}^{(t)}) - \mathcal{M}_n^r(\boldsymbol{\beta}^*)\|_2 + \|\mathcal{M}_n^r(\boldsymbol{\beta}^*) - \boldsymbol{\beta}^*\|_2. \tag{3.2}$$

Since we expect (and show) linear convergence of the optimization, it is natural to update $\lambda_n^{(t)}$ via a recursion of the form $\lambda_n^{(t)} = \kappa \lambda_n^{(t-1)} + \Delta$ as in (3.3), where the first term represents the optimization error, and $\Delta$ represents the final statistical error, i.e., the last term above in (3.2). A key part of our analysis shows that this error (and hence $\Delta$) is controlled by $\|\nabla Q_n(\boldsymbol{\beta}^*|\boldsymbol{\beta}) - \nabla Q(\boldsymbol{\beta}^*|\boldsymbol{\beta})\|_{\mathcal{R}^*}$, which in turn can be bounded uniformly for a variety of important applications of EM, including the three discussed in this paper (see Section 5). While a technical point, it is this key insight that enables the right choice of algorithm and its analysis. In the cases we consider, we obtain min-max optimal rates of convergence, demonstrating that no algorithm, let alone another variant of EM, can perform better.

---

**Algorithm 1** Regularized EM Algorithm

---

**Input** Samples $\{\mathbf{y}_i\}_{i=1}^n$, regularizer $\mathcal{R}$, number of iterations $T$, initial parameter $\boldsymbol{\beta}^{(0)}$, initial regularization parameter $\lambda_n^{(0)}$, estimated statistical error $\Delta$, contractive factor $\kappa < 1$.
1: **For** $t = 1, 2, \ldots, T$ **do**
2:    **Regularization parameter update:**

$$\lambda_n^{(t)} \leftarrow \kappa \lambda_n^{(t-1)} + \Delta. \tag{3.3}$$

3:    **E-step**: Compute function $Q_n(\cdot|\boldsymbol{\beta}^{(t-1)})$ according to (2.1).
4:    **Regularized M-step:**

$$\boldsymbol{\beta}^{(t)} \leftarrow \mathcal{M}_n^r(\boldsymbol{\beta}^{(t-1)}) := \arg\max_{\boldsymbol{\beta} \in \Omega} Q_n(\boldsymbol{\beta}|\boldsymbol{\beta}^{(t-1)}) - \lambda_n^{(t)} \cdot \mathcal{R}(\boldsymbol{\beta}).$$

5: **End For**
**Output** $\boldsymbol{\beta}^{(T)}$.

---

## 4 Statistical Guarantees

We now turn to the theoretical analysis of regularized EM algorithm. We first set up a general analytical framework for regularized EM where the key ingredients are decomposable regularizer and several technical conditions on the population based $Q(\cdot|\cdot)$ and the sample based $Q_n(\cdot|\cdot)$. In Section 4.3, we provide our main result (Theorem 1) that characterizes both computational and statistical performance of the proposed variant of regularized EM algorithm.

### 4.1 Decomposable Regularizers

Decomposable regularizers (e.g., [3, 6, 14, 19]), have been shown to be useful both empirically and theoretically for high dimensional structural estimation, and they also play an important role in our analytical framework. Recall that for $\mathcal{R} : \mathbb{R}^p \to \mathbb{R}^+$ a norm, and a pair of subspaces $(\mathcal{S}, \overline{\mathcal{S}})$ in $\mathbb{R}^p$ such that $\mathcal{S} \subseteq \overline{\mathcal{S}}$, we have the following definition:

**Definition 1** (Decomposability). *Regularizer $\mathcal{R} : \mathbb{R}^p \to \mathbb{R}^+$ is decomposable with respect to $(\mathcal{S}, \overline{\mathcal{S}})$ if*

$$\mathcal{R}(\mathbf{u} + \mathbf{v}) = \mathcal{R}(\mathbf{u}) + \mathcal{R}(\mathbf{v}), \text{ for any } \mathbf{u} \in \mathcal{S}, \mathbf{v} \in \overline{\mathcal{S}}^\perp.$$

Typically, the structure of model parameter $\boldsymbol{\beta}^*$ can be characterized by specifying a subspace $\mathcal{S}$ such that $\boldsymbol{\beta}^* \in \mathcal{S}$. The common use of a regularizer is thus to penalize the compositions of solution that

live outside $\mathcal{S}$. We are interested in bounding the estimation error in some norm $\|\cdot\|$. The following quantity is critical in connecting $\mathcal{R}$ to $\|\cdot\|$.

**Definition 2** (Subspace Compatibility Constant). *For any subspace $\mathcal{S} \subseteq \mathbb{R}^p$, a given regularizer $\mathcal{R}$ and some norm $\|\cdot\|$, the subspace compatibility constant of $\mathcal{S}$ with respect to $\mathcal{R}, \|\cdot\|$ is given by*

$$\Psi(\mathcal{S}) := \sup_{\mathbf{u} \in \mathcal{S}\backslash\{\mathbf{0}\}} \frac{\mathcal{R}(\mathbf{u})}{\|\mathbf{u}\|}.$$

As is standard, the dual norm of $\mathcal{R}$ is defined as $\mathcal{R}^*(\mathbf{v}) := \sup_{\mathcal{R}(\mathbf{u}) \leq 1} \langle \mathbf{u}, \mathbf{v} \rangle$. To simplify notation, we let $\|\mathbf{u}\|_{\mathcal{R}} := \mathcal{R}(\mathbf{u})$ and $\|\mathbf{u}\|_{\mathcal{R}^*} := \mathcal{R}^*(\mathbf{u})$.

## 4.2 Conditions on $Q(\cdot|\cdot)$ and $Q_n(\cdot|\cdot)$

Next, we review three technical conditions, originally proposed by [1], on the population level $Q(\cdot|\cdot)$ function, and then we give two important conditions that the empirical function $Q_n(\cdot|\cdot)$ must satisfy, including one that characterizes the statistical error.

It is well known that performance of EM algorithm is sensitive to initialization. Following the low-dimensional development in [1], our results are local, and apply to an $r$-neighborhood region around $\boldsymbol{\beta}^*$: $\mathcal{B}(r; \boldsymbol{\beta}^*) := \{\mathbf{u} \in \Omega, \|\mathbf{u} - \boldsymbol{\beta}^*\| \leq r\}$.

We first require that $Q(\cdot|\boldsymbol{\beta}^*)$ is *self consistent* as stated below. This is satisfied, in particular, when $\boldsymbol{\beta}^*$ maximizes the population log likelihood function, as happens in most settings of interest [12].

**Condition 1** (Self Consistency). *Function $Q(\cdot|\boldsymbol{\beta}^*)$ is self consistent, namely*

$$\boldsymbol{\beta}^* = \arg\max_{\boldsymbol{\beta} \in \Omega} Q(\boldsymbol{\beta}|\boldsymbol{\beta}^*).$$

We also require that the function $Q(\cdot|\cdot)$ satisfies a certain strong concavity condition and is smooth over $\Omega$.

**Condition 2** (Strong Concavity and Smoothness $(\gamma, \mu, r)$). *$Q(\cdot|\boldsymbol{\beta}^*)$ is $\gamma$-strongly concave over $\Omega$, i.e.,*

$$Q(\boldsymbol{\beta}_2|\boldsymbol{\beta}^*) - Q(\boldsymbol{\beta}_1|\boldsymbol{\beta}^*) - \langle \nabla Q(\boldsymbol{\beta}_1|\boldsymbol{\beta}^*), \boldsymbol{\beta}_2 - \boldsymbol{\beta}_1 \rangle \leq -\frac{\gamma}{2}\|\boldsymbol{\beta}_2 - \boldsymbol{\beta}_1\|^2, \ \forall \ \boldsymbol{\beta}_1, \boldsymbol{\beta}_2 \in \Omega. \quad (4.1)$$

*For any $\boldsymbol{\beta} \in \mathcal{B}(r; \boldsymbol{\beta}^*)$, $Q(\cdot|\boldsymbol{\beta})$ is $\mu$-smooth over $\Omega$, i.e.,*

$$Q(\boldsymbol{\beta}_2|\boldsymbol{\beta}) - Q(\boldsymbol{\beta}_1|\boldsymbol{\beta}) - \langle \nabla Q(\boldsymbol{\beta}_1|\boldsymbol{\beta}), \boldsymbol{\beta}_2 - \boldsymbol{\beta}_1 \rangle \geq -\frac{\mu}{2}\|\boldsymbol{\beta}_2 - \boldsymbol{\beta}_1\|^2, \ \forall \ \boldsymbol{\beta}_1, \boldsymbol{\beta}_2 \in \Omega. \quad (4.2)$$

The next condition is key in guaranteeing the curvature of $Q(\cdot|\boldsymbol{\beta})$ is similar to that of $Q(\cdot|\boldsymbol{\beta}^*)$ when $\boldsymbol{\beta}$ is close to $\boldsymbol{\beta}^*$. It has also been called *First Order Stability* in [1].

**Condition 3** (Gradient Stability $(\tau, r)$). *For any $\boldsymbol{\beta} \in \mathcal{B}(r; \boldsymbol{\beta}^*)$, we have*

$$\|\nabla Q(\mathcal{M}(\boldsymbol{\beta})|\boldsymbol{\beta}) - \nabla Q(\mathcal{M}(\boldsymbol{\beta})|\boldsymbol{\beta}^*)\| \leq \tau\|\boldsymbol{\beta} - \boldsymbol{\beta}^*\|.$$

The above condition only requires that the gradient be stable at one point $\mathcal{M}(\boldsymbol{\beta})$. This is sufficient for our analysis. In fact, for many concrete examples, one can verify a stronger version of Condition 3 that is $\|\nabla Q(\boldsymbol{\beta}'|\boldsymbol{\beta}) - \nabla Q(\boldsymbol{\beta}'|\boldsymbol{\beta}^*)\| \leq \tau\|\boldsymbol{\beta} - \boldsymbol{\beta}^*\|, \ \forall \ \boldsymbol{\beta}' \in \mathcal{B}(r; \boldsymbol{\beta}^*)$.

Next we require two conditions on the empirical function $Q_n(\cdot|\cdot)$, which is computed from finite number of samples according to (2.1). Our first condition, parallel to Condition 2, imposes a curvature constraint on $Q_n(\cdot|\cdot)$. In order to guarantee that the estimation error $\|\boldsymbol{\beta}^{(t)} - \boldsymbol{\beta}^*\|$ in step $t$ of the EM algorithm is well controlled, we would like $Q_n(\cdot|\boldsymbol{\beta}^{(t-1)})$ to be strongly concave at $\boldsymbol{\beta}^*$. However, in the setting where $n \ll p$, there might exist directions along which $Q_n(\cdot|\boldsymbol{\beta}^{(t-1)})$ is flat, e.g., as in mixed linear regression and missing covariate regression. In contrast with Condition 2, we only require $Q_n(\cdot|\cdot)$ to be strongly concave over a particular set $\mathcal{C}(\mathcal{S}, \overline{\mathcal{S}}; \mathcal{R})$ that is defined in terms of the subspace pair $(\mathcal{S}, \overline{\mathcal{S}})$ and regularizer $\mathcal{R}$. This set is defined as follows:

$$\mathcal{C}(\mathcal{S}, \overline{\mathcal{S}}; \mathcal{R}) := \left\{\mathbf{u} \in \mathbb{R}^p : \left\|\Pi_{\overline{\mathcal{S}}^\perp}(\mathbf{u})\right\|_{\mathcal{R}} \leq 2 \cdot \left\|\Pi_{\overline{\mathcal{S}}}(\mathbf{u})\right\|_{\mathcal{R}} + 2 \cdot \Psi(\overline{\mathcal{S}}) \cdot \|\mathbf{u}\|\right\}, \quad (4.3)$$

where the projection operator $\Pi_{\mathcal{S}} : \mathbb{R}^p \to \mathbb{R}^p$ is defined as $\Pi_{\mathcal{S}}(\mathbf{u}) := \arg\min_{\mathbf{v} \in \mathcal{S}} \|\mathbf{v} - \mathbf{u}\|$. The restricted strong concavity (RSC) condition is as follows.

**Condition 4** (RSC $(\gamma_n, \mathcal{S}, \overline{\mathcal{S}}, r, \delta)$). *For any fixed $\boldsymbol{\beta} \in \mathcal{B}(r; \boldsymbol{\beta}^*)$, with probability at least $1 - \delta$, we have that for all $\boldsymbol{\beta}' - \boldsymbol{\beta}^* \in \Omega \bigcap \mathcal{C}(\mathcal{S}, \overline{\mathcal{S}}; \mathcal{R})$,*

$$Q_n(\boldsymbol{\beta}'|\boldsymbol{\beta}) - Q_n(\boldsymbol{\beta}^*|\boldsymbol{\beta}) - \langle \nabla Q_n(\boldsymbol{\beta}^*|\boldsymbol{\beta}), \boldsymbol{\beta}' - \boldsymbol{\beta}^* \rangle \leq -\frac{\gamma_n}{2} \|\boldsymbol{\beta}' - \boldsymbol{\beta}^*\|^2.$$

The above condition states that $Q_n(\cdot|\boldsymbol{\beta})$ is strongly concave in directions $\boldsymbol{\beta}' - \boldsymbol{\beta}^*$ that belong to $\mathcal{C}(\mathcal{S}, \overline{\mathcal{S}}; \mathcal{R})$. It is instructive to compare Condition 4 with a related condition proposed by [14] for analyzing high dimensional M-estimators. They require the loss function to be strongly convex over the cone $\{\mathbf{u} \in \mathbb{R}^p : \|\Pi_{\overline{\mathcal{S}}^\perp}(\mathbf{u})\|_{\mathcal{R}} \lesssim \|\Pi_{\overline{\mathcal{S}}}(\mathbf{u})\|_{\mathcal{R}}\}$. Therefore our restrictive set (4.3) is similar to the cone but has the additional term $2\Psi(\overline{\mathcal{S}})\|\mathbf{u}\|$. The main purpose of the term $2\Psi(\overline{\mathcal{S}})\|\mathbf{u}\|$ is to allow the regularization parameter $\lambda_n$ to jointly control optimization and statistical error. We note that while Condition 4 is stronger than the usual RSC condition in M-estimator, in typical settings the difference is immaterial. This is because $\left\|\Pi_{\overline{\mathcal{S}}}(\mathbf{u})\right\|_{\mathcal{R}}$ is within a constant factor of $\Psi(\overline{\mathcal{S}}) \cdot \left\|\mathbf{u}\right\|$, and hence checking RSC over $\mathcal{C}$ amounts to checking it over $\|\Pi_{\overline{\mathcal{S}}^\perp}(\mathbf{u})\|_{\mathcal{R}} \lesssim \Psi(\overline{\mathcal{S}})\|\mathbf{u}\|$, which is indeed what is typically also done in the M-estimator setting.

Finally, we establish the condition that characterizes the achievable statistical error.

**Condition 5** (Statistical Error $(\Delta_n, r, \delta)$). *For any fixed $\boldsymbol{\beta} \in \mathcal{B}(r; \boldsymbol{\beta}^*)$, with probability at least $1 - \delta$, we have*

$$\left\|\nabla Q_n(\boldsymbol{\beta}^*|\boldsymbol{\beta}) - \nabla Q(\boldsymbol{\beta}^*|\boldsymbol{\beta})\right\|_{\mathcal{R}^*} \leq \Delta_n. \tag{4.4}$$

This quantity replaces the term $\|\mathcal{M}_n(\boldsymbol{\beta}) - \mathcal{M}(\boldsymbol{\beta})\|$ which appears in [1] and [20], and which presents problems in the high dimensional regime.

### 4.3 Main Results

In this section, we provide the theoretical guarantees for a *resampled version* of our regularized EM algorithm: we split the whole dataset into $T$ pieces and use a fresh piece of data in each iteration of regularized EM. As in [1], resampling makes it possible to check that Conditions 4-5 are satisfied without requiring them to hold uniformly for all $\boldsymbol{\beta} \in \mathcal{B}(r; \boldsymbol{\beta}^*)$ with high probability. Our empirical results indicate that it is not in fact required and is an artifact of the analysis. We refer to this resampled version as **Algorithm 2**. In the sequel, we let $m := n/T$ to denote the sample complexity in each iteration. We let $\alpha := \sup_{\mathbf{u} \in \mathbb{R}^p \backslash \{0\}} \|\mathbf{u}\|_*/\|\mathbf{u}\|$, where $\|\cdot\|_*$ is the dual norm of $\|\cdot\|$.

For Algorithm 2, our main result is as follows. The proof is deferred to the Supplemental Material.

**Theorem 1.** *Assume the model parameter $\boldsymbol{\beta}^* \in \mathcal{S}$ and regularizer $\mathcal{R}$ is decomposable with respect to $(\mathcal{S}, \overline{\mathcal{S}})$ where $\mathcal{S} \subseteq \overline{\mathcal{S}} \subseteq \mathbb{R}^p$. Assume $r > 0$ is such that $\mathcal{B}(r; \boldsymbol{\beta}^*) \subseteq \Omega$. Further, assume function $Q(\cdot|\cdot)$, defined in (2.2), is self consistent and satisfies Conditions 2-3 with parameters $(\gamma, \mu, r)$ and $(\tau, r)$. Given $n$ samples and $T$ iterations, let $m := n/T$. Assume $Q_m(\cdot|\cdot)$, computed from any $m$ i.i.d. samples according to (2.1), satisfies Conditions 4-5 with parameters $(\gamma_m, \mathcal{S}, \overline{\mathcal{S}}, r, 0.5\delta/T)$ and $(\Delta_m, r, 0.5\delta/T)$. Let $\kappa^* := 5\frac{\alpha\mu\tau}{\gamma\gamma_m}$, and assume $0 < \tau < \gamma$ and $0 < \kappa^* \leq 3/4$. Define $\overline{\Delta} := r\gamma_m/[60\Psi(\overline{\mathcal{S}})]$ and assume $\Delta_m$ is such that $\Delta_m \leq \overline{\Delta}$.*

*Consider Algorithm 2 with initialization $\boldsymbol{\beta}^{(0)} \in \mathcal{B}(r; \boldsymbol{\beta}^*)$ and with regularization parameters given by*

$$\lambda_m^{(t)} = \kappa^t \frac{\gamma_m}{5\Psi(\overline{\mathcal{S}})} \|\boldsymbol{\beta}^{(0)} - \boldsymbol{\beta}^*\| + \frac{1 - \kappa^t}{1 - \kappa}\Delta, \ t = 1, 2, \ldots, T \tag{4.5}$$

*for any $\Delta \in [3\Delta_m, 3\overline{\Delta}]$, $\kappa \in [\kappa^*, 3/4]$. Then with probability at least $1 - \delta$, we have that for any $t \in [T]$,*

$$\|\boldsymbol{\beta}^{(t)} - \boldsymbol{\beta}^*\| \leq \kappa^t \|\boldsymbol{\beta}^{(0)} - \boldsymbol{\beta}^*\| + \frac{5}{\gamma_m} \frac{1 - \kappa^t}{1 - \kappa}\Psi(\overline{\mathcal{S}})\Delta. \tag{4.6}$$

The estimation error is bounded by a term decaying linearly with number of iterations $t$, which we can think of as the *optimization error* and a second term that characterizes the ultimate *estimation error* of our algorithm. With $T = O(\log n)$ and suitable choice of $\Delta$ such that $\Delta = O(\Delta_{n/T})$, we bound the ultimate estimation error as

$$\|\boldsymbol{\beta}^{(T)} - \boldsymbol{\beta}^*\| \lesssim \frac{1}{(1 - \kappa)\gamma_{n/T}}\Psi(\overline{\mathcal{S}})\Delta_{n/T}. \tag{4.7}$$

We note that overestimating the initial error, $\|\boldsymbol{\beta}^{(0)} - \boldsymbol{\beta}^*\|$ is not important, as it may slightly increase the overall number of iterations, but will not impact the ultimate estimation error.

The constraint $\Delta_m \lesssim r\gamma_m/\Psi(\overline{\mathcal{S}})$ ensures that $\boldsymbol{\beta}^{(t)}$ is contained in $\mathcal{B}(r; \boldsymbol{\beta}^*)$ for all $t \in [T]$. This constraint is quite mild in the sense that if $\Delta_m = \Omega(r\gamma_m/\Psi(\overline{\mathcal{S}}))$, $\boldsymbol{\beta}^{(0)}$ is a decent estimator with estimation error $O(\Psi(\overline{\mathcal{S}})\Delta_m/\gamma_m)$ that already matches our expectation.

# 5 Examples: Applying the Theory

Now we introduce three well known latent variable models. For each model, we first review the standard EM algorithm formulations, and discuss the extensions to the high dimensional setting. Then we apply Theorem 1 to obtain the statistical guarantee of the regularized EM with data splitting (Algorithm 2). The key ingredient underlying these results is to check the technical conditions in Section 4 hold for each model. We postpone these tedious details to the Supplemental Material.

## 5.1 Gaussian Mixture Model

We consider the balanced isotropic Gaussian mixture model (GMM) with two components where the distribution of random variables $(Y, Z) \in \mathbb{R}^p \times \{-1, 1\}$ is characterized as

$$\Pr(Y = \mathbf{y}|Z = z) = \phi(\mathbf{y}; z \cdot \boldsymbol{\beta}^*, \sigma^2 \mathbf{I}_p), \ \Pr(Z = 1) = \Pr(Z = -1) = 1/2.$$

Here we use $\phi(\cdot|\boldsymbol{\mu}, \boldsymbol{\Sigma})$ to denote the probability density function of $\mathcal{N}(\boldsymbol{\mu}, \boldsymbol{\Sigma})$. In this example, $Z$ is the latent variable that indicates the cluster id of each sample. Given $n$ i.i.d. samples $\{\mathbf{y}_i\}_{i=1}^n$, function $Q_n(\cdot|\cdot)$ defined in (2.1) corresponds to

$$Q_n^{GMM}(\boldsymbol{\beta}'|\boldsymbol{\beta}) = -\frac{1}{2n}\sum_{i=1}^n \left[w(\mathbf{y}_i; \boldsymbol{\beta})\|\mathbf{y}_i - \boldsymbol{\beta}'\|_2^2 + (1 - w(\mathbf{y}_i; \boldsymbol{\beta}))\|\mathbf{y}_i + \boldsymbol{\beta}'\|_2^2\right], \qquad (5.1)$$

where $w(\mathbf{y}; \boldsymbol{\beta}) := \exp\left(-\frac{\|\mathbf{y}-\boldsymbol{\beta}\|_2^2}{2\sigma^2}\right)[\exp\left(-\frac{\|\mathbf{y}-\boldsymbol{\beta}\|_2^2}{2\sigma^2}\right) + \exp\left(-\frac{\|\mathbf{y}+\boldsymbol{\beta}\|_2^2}{2\sigma^2}\right)]^{-1}$. We assume $\boldsymbol{\beta}^* \in \mathcal{B}_0(s; p) := \{\mathbf{u} \in \mathbb{R}^p : |\text{supp}(\mathbf{u})| \le s\}$. Naturally, we choose the regularizer $\mathcal{R}(\cdot)$ to be the $\ell_1$ norm. We define the signal-to-noise ratio $\text{SNR} := \|\boldsymbol{\beta}^*\|_2/\sigma$.

**Corollary 1** (Sparse Recovery in GMM)**.** *There exist constants $\rho, C$ such that if* $\text{SNR} \ge \rho$, $n/T \ge [80C(\|\boldsymbol{\beta}^*\|_\infty + \sigma)/\|\boldsymbol{\beta}^*\|_2]^2 s\log p$, $\boldsymbol{\beta}^{(0)} \in \mathcal{B}(\|\boldsymbol{\beta}^*\|_2/4; \boldsymbol{\beta}^*)$*; then with probability at least* $1 - T/p$ *Algorithm 2 with parameters* $\Delta = C(\|\boldsymbol{\beta}^*\|_\infty + \sigma)\sqrt{T\log p/n}$, $\lambda_{n/T}^{(0)} = 0.2\|\boldsymbol{\beta}^{(0)} - \boldsymbol{\beta}^*\|_2/\sqrt{s}$*, any* $\kappa \in [1/2, 3/4]$ *and $\ell_1$ regularization generates $\boldsymbol{\beta}^{(t)}$ that has estimation error*

$$\|\boldsymbol{\beta}^{(t)} - \boldsymbol{\beta}^*\|_2 \le \kappa^t\|\boldsymbol{\beta}^{(0)} - \boldsymbol{\beta}^*\|_2 + \frac{5C(\|\boldsymbol{\beta}^*\|_\infty + \sigma)}{1 - \kappa}\sqrt{\frac{s\log p}{n}}T, \text{ for all } t \in [T]. \qquad (5.2)$$

Note that by setting $T \asymp \log(n/\log p)$, the order of final estimation error turns out to be $(\|\boldsymbol{\beta}^*\|_\infty + \delta)\sqrt{(s\log p)/n}\log(n/\log p)$. The minimax rate for estimating $s$-sparse vector in a single Gaussian cluster is $\sqrt{s\log p/n}$, thereby the rate is optimal on $(n, p, s)$ up to a log factor.

## 5.2 Mixed Linear Regression

Mixed linear regression (MLR), as considered in some recent work [5, 7, 22], is the problem of recovering two or more linear vectors from mixed linear measurements. In the case of mixed linear regression with two symmetric and balanced components, the response-covariate pair $(Y, X) \in \mathbb{R} \times \mathbb{R}^p$ is linked through

$$Y = \langle X, \ Z \cdot \boldsymbol{\beta}^* \rangle + W,$$

where $W$ is the noise term and $Z$ is the latent variable that has Rademacher distribution over $\{-1, 1\}$. We assume $X \sim \mathcal{N}(0, \mathbf{I}_p)$, $W \sim \mathcal{N}(0, \sigma^2)$. In this setting, with $n$ i.i.d. samples $\{y_i, \mathbf{x}_i\}_{i=1}^n$ of pair $(Y, X)$, function $Q_n(\cdot|\cdot)$ then corresponds to

$$Q_n^{MLR}(\boldsymbol{\beta}'|\boldsymbol{\beta}) = -\frac{1}{2n}\sum_{i=1}^n \left[w(y_i, \mathbf{x}_i; \boldsymbol{\beta})(y_i - \langle \mathbf{x}_i, \boldsymbol{\beta}' \rangle)^2 + (1 - w(y_i, \mathbf{x}_i; \boldsymbol{\beta}))(y_i + \langle \mathbf{x}_i, \boldsymbol{\beta}' \rangle)^2\right],$$

$$(5.3)$$

where $w(y, \mathbf{x}; \boldsymbol{\beta}) := \exp\left(-\frac{(y - \langle \mathbf{x}, \boldsymbol{\beta} \rangle)^2}{2\sigma^2}\right)\left[\exp\left(-\frac{(y - \langle \mathbf{x}, \boldsymbol{\beta} \rangle)^2}{2\sigma^2}\right) + \exp\left(-\frac{(y + \langle \mathbf{x}, \boldsymbol{\beta} \rangle)^2}{2\sigma^2}\right)\right]^{-1}$.

We consider two kinds of structure on $\boldsymbol{\beta}^*$:

**Sparse Recovery.** Assume $\boldsymbol{\beta}^* \in \mathcal{B}_0(s; p)$. Then let $\mathcal{R}$ be the $\ell_1$ norm, as in the previous section. We define $\mathrm{SNR} := \|\boldsymbol{\beta}^*\|_2/\sigma$.

**Corollary 2** (Sparse recovery in MLR). *There exist constant $\rho, C, C'$ such that if $\mathrm{SNR} \geq \rho$, $n/T \geq C'\left[(\|\boldsymbol{\beta}^*\|_2 + \delta)/\|\boldsymbol{\beta}^*\|_2\right]^2 s \log p$, $\boldsymbol{\beta}^{(0)} \in \mathcal{B}(\|\boldsymbol{\beta}^*\|_2/240, \boldsymbol{\beta}^*)$; then with probability at least $1 - T/p$ Algorithm 2 with parameters $\Delta = C(\|\boldsymbol{\beta}^*\|_2 + \delta)\sqrt{T \log p/n}$, $\lambda_{n/T}^{(0)} = \|\boldsymbol{\beta}^{(0)} - \boldsymbol{\beta}^*\|_2/(15\sqrt{s})$, any $\kappa \in [1/2, 3/4]$ and $\ell_1$ regularization generates $\boldsymbol{\beta}^{(t)}$ that has estimation error*

$$\|\boldsymbol{\beta}^{(t)} - \boldsymbol{\beta}^*\|_2 \leq \kappa^t \|\boldsymbol{\beta}^{(0)} - \boldsymbol{\beta}^*\|_2 + \frac{15C(\|\boldsymbol{\beta}^*\|_2 + \delta)}{1 - \kappa}\sqrt{\frac{s \log p}{n}}T, \text{ for all } t \in [T].$$

Performing $T \asymp \log(n/(s \log p))$ iterations gives us estimation rate $(\|\boldsymbol{\beta}^*\|_2 + \delta)\sqrt{(s \log p/n)\log(n/(s \log p))}$ which is near-optimal on $(s, p, n)$. The dependence on $\|\boldsymbol{\beta}^*\|_2$, which also appears in the analysis of EM in the classical (low dimensional) setting [1], arises from fundamental limits of EM. Removing such dependence for MLR is possible by convex relaxation [7]. It is interesting to study how to remove it in the high dimensional setting.

**Low Rank Recovery.** Second we consider the setting where the model parameter is a matrix $\boldsymbol{\Gamma}^* \in \mathbb{R}^{p_1 \times p_2}$ with $\mathrm{rank}(\boldsymbol{\Gamma}^*) = \theta \ll \min(p_1, p_2)$. We further assume $X \in \mathbb{R}^{p_1 \times p_2}$ is an i.i.d. Gaussian matrix, i.e., entries of $X$ are independent random variables with distribution $\mathcal{N}(0, 1)$. We apply nuclear norm regularization to serve the low rank structure, i.e, $\mathcal{R}(\boldsymbol{\Gamma}) = \sum_{i=1}^{p_1, p_2} |s_i(\boldsymbol{\Gamma})|$, where $s_i(\boldsymbol{\Gamma})$ is the $i$th singular value of $\boldsymbol{\Gamma}$. Similarly, we let $\mathrm{SNR} := \|\boldsymbol{\Gamma}^*\|_F/\sigma$.

**Corollary 3** (Low rank recovery in MLR). *There exist constant $\rho, C, C'$ such that if $\mathrm{SNR} \geq \rho$, $n/T \geq C'\left[(\|\boldsymbol{\Gamma}^*\|_F + \sigma)/\|\boldsymbol{\Gamma}^*\|_F\right]^2 \theta(p_1 + p_2)$, $\boldsymbol{\Gamma}^{(0)} \in \mathcal{B}(\|\boldsymbol{\Gamma}^*\|_F/1600, \boldsymbol{\Gamma}^*)$; then with probability at least $1 - T \exp(-p_1 - p_2)$ Algorithm 2 with parameters $\Delta = C(\|\boldsymbol{\Gamma}^*\|_F + \sigma)\sqrt{T(p_1 + p_2)/n}$, $\lambda_{n/T}^{(0)} = 0.01\|\boldsymbol{\Gamma}^{(0)} - \boldsymbol{\Gamma}^*\|_F/\sqrt{2\theta}$, any $\kappa \in [1/2, 3/4]$ and nuclear norm regularization generates $\boldsymbol{\Gamma}^{(t)}$ that has estimation error*

$$\|\boldsymbol{\Gamma}^{(t)} - \boldsymbol{\Gamma}^*\|_F \leq \kappa^t \|\boldsymbol{\Gamma}^{(0)} - \boldsymbol{\Gamma}^*\|_F + \frac{100C'(\|\boldsymbol{\Gamma}^*\|_F + \sigma)}{1 - \kappa}\sqrt{\frac{2\theta(p_1 + p_2)}{n}}T, \text{ for all } t \in [T].$$

The standard low rank matrix recovery with a single component, including other sensing matrix designs beyond the Gaussianity, has been studied extensively (e.g., [2, 4, 13, 15]). To the best of our knowledge, the theoretical study of the mixed low rank matrix recovery has not been considered.

### 5.3 Missing Covariate Regression

As our last example, we consider the missing covariate regression (MCR) problem. To parallel standard linear regression, $\{y_i, \mathbf{x}_i\}_{i=1}^n$ are samples of $(Y, X)$ linked through $Y = \langle X, \boldsymbol{\beta}^* \rangle + W$. However, we assume each entry of $\mathbf{x}_i$ is missing independently with probability $\epsilon \in (0, 1)$. Therefore, the observed covariate vector $\widetilde{\mathbf{x}}_i$ takes the form

$$\widetilde{x}_{i,j} = \begin{cases} x_{i,j} & \text{with probability } 1 - \epsilon \\ * & \text{otherwise} \end{cases}.$$

We assume the model is under Gaussian design $X \sim \mathcal{N}(\mathbf{0}, \mathbf{I}_p), W \sim \mathcal{N}(0, \sigma^2)$. We refer the reader to our Supplementary Material for the specific $Q_n(\cdot|\cdot)$ function. In high dimensional case, we assume $\boldsymbol{\beta}^* \in \mathcal{B}_0(s; p)$. We define $\rho := \|\boldsymbol{\beta}^*\|_2/\sigma$ to be the SNR and $\omega := r/\|\boldsymbol{\beta}^*\|_2$ to be the *relative contractivity radius*. In particular, let $\zeta := (1 + \omega)\rho$.

**Corollary 4** (Sparse Recovery in MCR). *There exist constants $C, C', C_0, C_1$ such that if $(1+\omega)\rho \leq C_0 < 1$, $\epsilon < C_1$, $n/T \geq C' \max\{\sigma^2(\omega\rho)^{-1}, 1\}s \log p$, $\boldsymbol{\beta}^{(0)} \in \mathcal{B}(\omega\|\boldsymbol{\beta}^*\|_2, \boldsymbol{\beta}^*)$; then with probability at least $1 - T/p$ Algorithm 2 with parameters $\Delta = C\sigma\sqrt{T \log p/n}$, $\lambda_{n/T}^{(0)} = \|\boldsymbol{\beta}^{(0)} - \boldsymbol{\beta}^*\|_2/(45\sqrt{s})$, any $\kappa \in [1/2, 3/4]$ and $\ell_1$ regularization generates $\boldsymbol{\beta}^{(t)}$ that has estimation error*

$$\|\boldsymbol{\beta}^{(t)} - \boldsymbol{\beta}^*\|_2 \leq \kappa^t \|\boldsymbol{\beta}^{(0)} - \boldsymbol{\beta}^*\|_2 + \frac{45C\sigma}{1 - \kappa}\sqrt{\frac{s \log p}{n}}T, \text{ for all } t \in [T],$$

Unlike the previous two models, we require an upper bound on the signal to noise ratio. This unusual constraint is in fact unavoidable [10]. By optimizing $T$, the order of final estimation error turns out to be $\sigma \sqrt{s \log p / n \log(n/(s \log p))}$.

## 6  Simulations

We now provide some simulation results to back up our theory. Note that while Theorem 1 requires resampling, we believe in practice this is unnecessary. This is validated by our results, where we apply Algorithm 1 to the four latent variable models discussed in Section 5.

*Convergence Rate.* We first evaluate the convergence of Algorithm 1 assuming only that the initialization is a bounded distance from $\boldsymbol{\beta}^*$. For a given error $\omega\|\boldsymbol{\beta}^*\|_2$, the initial parameter $\boldsymbol{\beta}^{(0)}$ is picked randomly from the sphere centered around $\boldsymbol{\beta}^*$ with radius $\omega\|\boldsymbol{\beta}^*\|_2$. We use Algorithm 1 with $T = 7$, $\kappa = 0.7$, $\lambda_n^{(0)}$ in Theorem 1. The choice of the critical parameter $\Delta$ is given in the Supplementary Material. For every single trial, we report *estimation error* $\|\boldsymbol{\beta}^{(t)} - \boldsymbol{\beta}^*\|_2$ and *optimization error* $\|\boldsymbol{\beta}^{(t)} - \boldsymbol{\beta}^{(T)}\|_2$ in every iteration. We plot the log of errors over iteration $t$ in Figure 1.

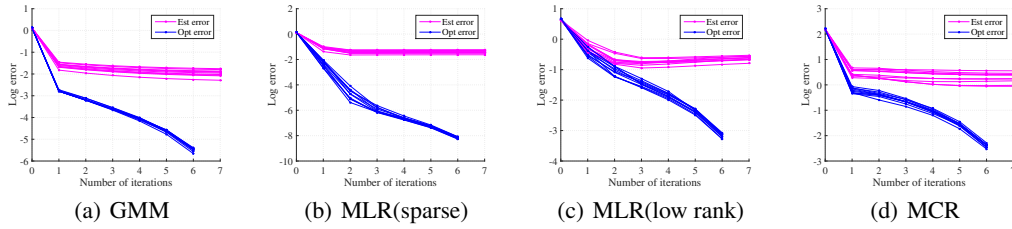

| (a) GMM | (b) MLR(sparse) | (c) MLR(low rank) | (d) MCR |

Figure 1: Convergence of regularized EM algorithm. In each panel, one curve is plotted from single independent trial. **Settings**: (a,b,d) $(n, p, s) = (500, 800, 5)$; (d) $(n, p, \theta) = (600, 30, 3)$; (a-c) SNR $= 5$; (d) $(\text{SNR}, \epsilon) = (0.5, 0.2)$; (a-d) $\omega = 0.5$.

*Statistical Rate.* We now evaluate the statistical rate. We set $T = 7$ and compute estimation error on $\widehat{\boldsymbol{\beta}} := \boldsymbol{\beta}^{(T)}$. In Figure 2, we plot $\|\widehat{\boldsymbol{\beta}} - \boldsymbol{\beta}^*\|_2$ over normalized sample complexity, i.e., $n/(s \log p)$ for $s$-sparse parameter and $n/(\theta p)$ for rank $\theta$ $p$-by-$p$ parameter. We refer the reader to Figure 1 for other settings. We observe that the same normalized sample complexity leads to almost identical estimation error in practice, which thus supports the corresponding statistical rate established in Section 5.

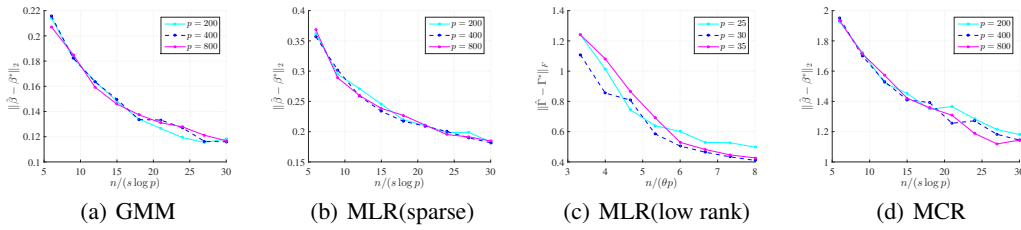

| (a) GMM | (b) MLR(sparse) | (c) MLR(low rank) | (d) MCR |

Figure 2: Statistical rates. Each point is an average of 20 independent trials. **Settings**: (a,b,d) $s = 5$; (c) $\theta = 3$.

## Acknowledgments

The authors would like to acknowledge NSF grants 1056028, 1302435 and 1116955. This research was also partially supported by the U.S. Department of Transportation through the Data-Supported Transportation Operations and Planning (D-STOP) Tier 1 University Transportation Center.

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
