[Supplementary Material · EM_supp.pdf]

# Regularized EM: Supplemental Material

In this supplemental document, we present full proofs of the results in our main paper and provide detailed discussions. The rest of this document is organized as follows. we give the proof of the main result in Section A. The proofs of the results showing specializations of example models are given in Section B. We collect several technical lemmas in Section C.

## A   Proof of Main Result

In this section, we provide the proof of Theorem 1 that characterizes the computational and statistical performance of the regularized EM algorithm with resampling. We first present a result which shows that the population EM operator $\mathcal{M} : \Omega \to \Omega$ is contractive when $\tau < \gamma$.

**Lemma 1.** *Suppose $Q(\cdot|\cdot)$ satisfies all the corresponding conditions stated in Theorem 1. Mapping $\mathcal{M}$ is contractive over $\mathcal{B}(r; \boldsymbol{\beta}^*)$, namely*

$$\|\mathcal{M}(\boldsymbol{\beta}) - \boldsymbol{\beta}^*\| \leq \frac{\tau}{\gamma}\|\boldsymbol{\beta} - \boldsymbol{\beta}^*\|, \ \forall \ \boldsymbol{\beta} \in \mathcal{B}(r; \boldsymbol{\beta}^*).$$

*Proof.* A similar result is proved in [1]. The slight difference is that [1] shows Lemma 1 with $\ell_2$ norm. Extending $\ell_2$ norm to arbitrary norm is trivial, so we omit the details. □

Now we are ready to prove Theorem 1.

*Proof of Theorem 1.* We first consider one iteration of Algorithm 1 and show the relationship between $\|\boldsymbol{\beta}^{(t)} - \boldsymbol{\beta}^*\|$ and $\|\boldsymbol{\beta}^{(t-1)} - \boldsymbol{\beta}^*\|$. Recall that

$$\boldsymbol{\beta}^{(t)} = \arg\max_{\boldsymbol{\beta}' \in \Omega} Q_m(\boldsymbol{\beta}'|\boldsymbol{\beta}^{(t-1)}) - \lambda_m^{(t)} \cdot \mathcal{R}(\boldsymbol{\beta}'),$$

where $m = n/T$ is the number of samples in each step. We assume $\boldsymbol{\beta}^{(t-1)} \in \mathcal{B}(r; \boldsymbol{\beta}^*)$. To simplify the notation, we drop the superscripts of $\boldsymbol{\beta}^{(t-1)}$, $\lambda_m^{(t)}$ and denote $\boldsymbol{\beta}^{(t)}$ as $\boldsymbol{\beta}^+$. From the optimality of $\boldsymbol{\beta}^+$, we have

$$Q_m(\boldsymbol{\beta}^+|\boldsymbol{\beta}) - \lambda_m \cdot \mathcal{R}(\boldsymbol{\beta}^+) \geq Q_m(\boldsymbol{\beta}^*|\boldsymbol{\beta}) - \lambda_m \cdot \mathcal{R}(\boldsymbol{\beta}^*). \tag{A.1}$$

Equivalently,

$$\lambda_m \cdot \mathcal{R}(\boldsymbol{\beta}^+) - \lambda_m \cdot \mathcal{R}(\boldsymbol{\beta}^*) \leq Q_m(\boldsymbol{\beta}^+|\boldsymbol{\beta}) - Q_m(\boldsymbol{\beta}^*|\boldsymbol{\beta}). \tag{A.2}$$

Using the fact that $Q_m(\cdot|\boldsymbol{\beta})$ is a concave function, the right hand side of the above inequality can be bounded as

$$Q_m(\boldsymbol{\beta}^+|\boldsymbol{\beta}) - Q_m(\boldsymbol{\beta}^*|\boldsymbol{\beta}) \leq \langle \nabla Q_m(\boldsymbol{\beta}^*|\boldsymbol{\beta}), \boldsymbol{\beta}^+ - \boldsymbol{\beta}\rangle \leq \underbrace{\left|\langle \nabla Q_m(\boldsymbol{\beta}^*|\boldsymbol{\beta}), \boldsymbol{\beta}^+ - \boldsymbol{\beta}\rangle\right|}_{A}. \tag{A.3}$$

A key ingredient of our proof is to bound the term $A$. Letting $\Theta := \boldsymbol{\beta}^+ - \boldsymbol{\beta}^*$, we have

$$
\begin{aligned}
\left|\langle \nabla Q_m(\boldsymbol{\beta}^*|\boldsymbol{\beta}), \boldsymbol{\beta}^+ - \boldsymbol{\beta}\rangle\right| &= \left|\langle \nabla Q_m(\boldsymbol{\beta}^*|\boldsymbol{\beta}) - \nabla Q(\boldsymbol{\beta}^*|\boldsymbol{\beta}) + \nabla Q(\boldsymbol{\beta}^*|\boldsymbol{\beta}), \Theta\rangle\right| \\
&\leq \left|\langle \nabla Q_m(\boldsymbol{\beta}^*|\boldsymbol{\beta}) - \nabla Q(\boldsymbol{\beta}^*|\boldsymbol{\beta}), \Theta\rangle\right| + \left|\langle \nabla Q(\boldsymbol{\beta}^*|\boldsymbol{\beta}), \Theta\rangle\right| \\
&\overset{(a)}{\leq} \left\|Q_m(\boldsymbol{\beta}^*|\boldsymbol{\beta}) - \nabla Q(\boldsymbol{\beta}^*|\boldsymbol{\beta})\right\|_{\mathcal{R}^*} \cdot \mathcal{R}(\Theta) + \left\|\nabla Q(\boldsymbol{\beta}^*|\boldsymbol{\beta})\right\|_* \times \|\Theta\| \\
&\overset{(b)}{\leq} \Delta_m \mathcal{R}(\Theta) + \alpha\left\|\nabla Q(\boldsymbol{\beta}^*|\boldsymbol{\beta})\right\| \times \|\Theta\| \\
&\overset{(c)}{\leq} \Delta_m \mathcal{R}(\Theta) + \alpha\left\|\nabla Q(\boldsymbol{\beta}^*|\boldsymbol{\beta}) - \nabla Q(\mathcal{M}(\boldsymbol{\beta})|\boldsymbol{\beta})\right\| \times \|\Theta\| \\
&\overset{(d)}{\leq} \Delta_m \mathcal{R}(\Theta) + \alpha\mu\left\|\mathcal{M}(\boldsymbol{\beta}) - \boldsymbol{\beta}^*\right\| \times \|\Theta\| \\
&\overset{(e)}{\leq} \Delta_m \mathcal{R}(\Theta) + \frac{\alpha\mu\tau}{\gamma}\left\|\boldsymbol{\beta} - \boldsymbol{\beta}^*\right\| \times \|\Theta\| \tag{A.4}
\end{aligned}
$$

where $(a)$ follows from the Cauchy-Schwarz inequality, $(b)$ follows from the statistical error Condition 5 and the definition of $\alpha$, $(c)$ follows from the fact that $\mathcal{M}(\boldsymbol{\beta})$ maximizes $Q(\cdot|\boldsymbol{\beta})$, $(d)$ follows

from the smoothness Condition 2, and $(e)$ follows from Lemma 1. For inequality $(c)$, note that we assume that $\mathcal{B}(r; \boldsymbol{\beta}^*) \subseteq \Omega$. From Lemma 1, we know that if $\boldsymbol{\beta} \in \mathcal{B}(r; \boldsymbol{\beta}^*)$, under condition $\tau < \gamma$, we must have $\mathcal{M}(\boldsymbol{\beta}) \in \mathcal{B}(r\tau/\gamma; \boldsymbol{\beta}^*) \subseteq \mathcal{B}(r; \boldsymbol{\beta}^*)$. Therefore $\mathcal{M}(\boldsymbol{\beta})$ lies in the interior of $\Omega$ thus the optimality condition corresponds to $\nabla Q(\mathcal{M}(\boldsymbol{\beta})|\boldsymbol{\beta}) = \mathbf{0}$.

Plugging (A.4) back into (A.3), we obtain

$$Q_m(\boldsymbol{\beta}^+|\boldsymbol{\beta}) - Q_m(\boldsymbol{\beta}^*|\boldsymbol{\beta}) \leq \Delta_m \mathcal{R}(\Theta) + \frac{\alpha\mu\tau}{\gamma}\|\boldsymbol{\beta} - \boldsymbol{\beta}^*\| \times \|\Theta\|.$$

Using the above result and (A.2), we have

$$\lambda_m \mathcal{R}(\boldsymbol{\beta}^* + \Theta) - \lambda_m \mathcal{R}(\boldsymbol{\beta}^*) \leq \Delta_m \mathcal{R}(\Theta) + \frac{\alpha\mu\tau}{\gamma}\|\boldsymbol{\beta} - \boldsymbol{\beta}^*\| \times \|\Theta\|. \tag{A.5}$$

To ease notation, we use $\mathbf{u}_{\mathcal{S}}$ to denote the projection operator $\Pi_{\mathcal{S}}(\mathbf{u})$ defined in (??). From the decomposability of $\mathcal{R}$, we have

$$\mathcal{R}(\boldsymbol{\beta}^* + \Theta) - \mathcal{R}(\boldsymbol{\beta}^*) \geq \mathcal{R}(\boldsymbol{\beta}^* + \Theta_{\overline{\mathcal{S}}^\perp}) - \mathcal{R}(\Theta_{\overline{\mathcal{S}}}) - \mathcal{R}(\boldsymbol{\beta}^*)$$
$$= \mathcal{R}(\Theta_{\overline{\mathcal{S}}^\perp}) - \mathcal{R}(\Theta_{\overline{\mathcal{S}}}),$$

where the inequality is from the triangle inequality and the equality is from decomposability of $\mathcal{R}$. Plugging the above result back into (A.5) yields that

$$\lambda_m \cdot \left(\mathcal{R}(\Theta_{\overline{\mathcal{S}}^\perp}) - \mathcal{R}(\Theta_{\overline{\mathcal{S}}})\right) \leq \Delta_m \mathcal{R}(\Theta) + \frac{\alpha\mu\tau}{\gamma}\|\boldsymbol{\beta} - \boldsymbol{\beta}^*\| \times \|\Theta\|.$$

By choosing $\lambda_m$ so that it satisfies the following condition

$$\lambda_m \geq 3\Delta_m + \frac{\alpha\mu\tau}{\gamma\Psi(\overline{\mathcal{S}})}\|\boldsymbol{\beta} - \boldsymbol{\beta}^*\|, \tag{A.6}$$

we have that

$$\mathcal{R}(\Theta_{\overline{\mathcal{S}}^\perp}) - \mathcal{R}(\Theta_{\overline{\mathcal{S}}}) \leq \frac{\Delta_m}{\lambda_m}\mathcal{R}(\Theta) + \frac{\alpha\mu\tau\|\boldsymbol{\beta} - \boldsymbol{\beta}^*\|}{\gamma\lambda_m}\|\Theta\| \leq \frac{1}{3}\mathcal{R}(\Theta) + \Psi(\overline{\mathcal{S}})\|\Theta\|.$$

Plugging $\mathcal{R}(\Theta) \leq \mathcal{R}(\Theta_{\overline{\mathcal{S}}}) + \mathcal{R}(\Theta_{\overline{\mathcal{S}}^\perp})$ into the above inequality, we obtain

$$2\mathcal{R}(\Theta_{\overline{\mathcal{S}}^\perp}) \leq 4\mathcal{R}(\Theta_{\overline{\mathcal{S}}}) + 3\Psi(\overline{\mathcal{S}}) \cdot \|\Theta\|. \tag{A.7}$$

Therefore, we have shown that $\Theta$ lies in the quasi cone $\mathcal{C}(\mathcal{S}, \overline{\mathcal{S}}; \mathcal{R})$ defined in (4.3). Recall that Condition 4 states that for any fixed $\boldsymbol{\beta} \in \mathcal{B}(r; \boldsymbol{\beta}^*)$, $Q_m(\cdot|\boldsymbol{\beta})$ is strongly concave over the set $\Omega \bigcap \left(\{\boldsymbol{\beta}^*\} + \mathcal{C}(\mathcal{S}, \overline{\mathcal{S}}; \mathcal{R})\right)$. Using this condition yields that

$$Q_m(\boldsymbol{\beta}^* + \Theta|\boldsymbol{\beta}) - Q_m(\boldsymbol{\beta}^*|\boldsymbol{\beta}) \leq \langle\nabla Q_m(\boldsymbol{\beta}^*|\boldsymbol{\beta}), \Theta\rangle - \frac{\gamma_m}{2}\|\Theta\|^2$$
$$\leq \Delta_m\mathcal{R}(\Theta) + \frac{\alpha\mu\tau}{\gamma}\|\boldsymbol{\beta} - \boldsymbol{\beta}^*\| \times \|\Theta\| - \frac{\gamma_m}{2}\|\Theta\|^2, \tag{A.8}$$

where the second inequality follows from (A.4).

Now we turn back to optimality condition (A.2), following which we have

$$Q_m(\boldsymbol{\beta}^* + \Theta|\boldsymbol{\beta}) - Q_m(\boldsymbol{\beta}^*|\boldsymbol{\beta}) \geq \lambda_m \cdot \mathcal{R}(\boldsymbol{\beta}^* + \Theta) - \lambda_m \cdot \mathcal{R}(\boldsymbol{\beta}^*) \geq -\lambda_m\mathcal{R}(\Theta_{\overline{\mathcal{S}}}). \tag{A.9}$$

Putting (A.8) and (A.9) together gives us

$$\frac{\gamma_m}{2}\|\Theta\|^2 \leq \lambda_m\mathcal{R}(\Theta_{\overline{\mathcal{S}}}) + \Delta_m\mathcal{R}(\Theta) + \frac{\alpha\mu\tau}{\gamma}\|\boldsymbol{\beta} - \boldsymbol{\beta}^*\| \times \|\Theta\|.$$

Using $\mathcal{R}(\Theta) \leq \mathcal{R}(\Theta_{\overline{\mathcal{S}}^\perp}) + \mathcal{R}(\Theta_{\overline{\mathcal{S}}}) \leq (9/2)\Psi(\overline{\mathcal{S}})\|\Theta\|$, we further have

$$\frac{\gamma_m}{2}\|\Theta\|^2 \leq \lambda_m\Psi(\overline{\mathcal{S}})\|\Theta\| + \frac{9}{2}\Delta_m\Psi(\overline{\mathcal{S}})\|\Theta\| + \frac{\alpha\mu\tau}{\gamma}\|\boldsymbol{\beta} - \boldsymbol{\beta}^*\| \times \|\Theta\|.$$

Canceling the term $\|\Theta\|$ on both sides of the above inequality yields that

$$\|\Theta\| \leq 2\Psi(\overline{\mathcal{S}})\frac{\lambda_m}{\gamma_m} + \frac{\Psi(\overline{\mathcal{S}})}{\gamma_m}\left(9\Delta_m + 2\frac{\alpha\mu\tau}{\gamma\Psi(\overline{\mathcal{S}})}\|\boldsymbol{\beta} - \boldsymbol{\beta}^*\|\right) \leq 5\Psi(\overline{\mathcal{S}})\frac{\lambda_m}{\gamma_m}. \tag{A.10}$$

The last inequality follows from our assumption (A.6). Putting (A.6) and (A.10) together, we reach the conclusion that if $\boldsymbol{\beta}^{(t-1)} \in \mathcal{B}(r; \boldsymbol{\beta}^*)$ and

$$\lambda_m^{(t)} \geq 3\Delta_m + \frac{\alpha\mu\tau}{\gamma\Psi(\overline{\mathcal{S}})}\|\boldsymbol{\beta}^{(t-1)} - \boldsymbol{\beta}^*\|, \tag{A.11}$$

then we have

$$\|\boldsymbol{\beta}^{(t)} - \boldsymbol{\beta}^*\| \leq 5\Psi(\overline{\mathcal{S}})\frac{\lambda_m^{(t)}}{\gamma_m}. \tag{A.12}$$

As in the statement of the theorem, let $\kappa^* := 5\frac{\alpha\mu\tau}{\gamma\gamma_m}$ and assume $\kappa^* \leq 3/4$. Then for any $\kappa \in [\kappa^*, 3/4], \Delta \geq 3\Delta_m$, we can set

$$\lambda_m^{(t)} = \frac{1 - \kappa^t}{1 - \kappa}\Delta + \kappa^t \frac{\gamma_m}{5\Psi(\overline{\mathcal{S}})}\|\boldsymbol{\beta}^{(0)} - \boldsymbol{\beta}^*\| \tag{A.13}$$

for all $t \in [T]$. When $t = 1$, we have $\boldsymbol{\beta}^{(0)} \in \mathcal{B}(r; \boldsymbol{\beta}^*)$ and one can check inequality (A.11) holds by setting $t = 1$ in (A.13), thereby applying (A.12) yields that

$$\|\boldsymbol{\beta}^{(1)} - \boldsymbol{\beta}^*\| \leq 5\Psi(\overline{\mathcal{S}})\frac{\lambda_m^{(1)}}{\gamma_m} = \frac{5\Psi(\overline{\mathcal{S}})}{\gamma_m}\frac{1 - \kappa}{1 - \kappa}\Delta + \kappa\|\boldsymbol{\beta}^{(0)} - \boldsymbol{\beta}^*\|.$$

Now we prove Theorem 1 by induction. Assume that for some $t \geq 1$,

$$\|\boldsymbol{\beta}^{(t)} - \boldsymbol{\beta}^*\| \leq \frac{5\Psi(\overline{\mathcal{S}})}{\gamma_m}\frac{1 - \kappa^t}{1 - \kappa}\Delta + \kappa^t\|\boldsymbol{\beta}^{(0)} - \boldsymbol{\beta}^*\|. \tag{A.14}$$

Under condition $\Delta \leq 3\overline{\Delta}, \kappa \leq 3/4$, we have

$$\|\boldsymbol{\beta}^{(t)} - \boldsymbol{\beta}^*\| \leq \frac{15\Psi(\overline{\mathcal{S}})}{\gamma_m}\frac{1 - (3/4)^t}{1 - 3/4}\overline{\Delta} + (3/4)^t\|\boldsymbol{\beta}^{(0)} - \boldsymbol{\beta}^*\| \leq \frac{15\Psi(\overline{\mathcal{S}})}{\gamma_m}\frac{1 - (3/4)^t}{1 - 3/4}\overline{\Delta} + (3/4)^t \cdot r$$
$$= (1 - (3/4)^t) \cdot r + (3/4)^t \cdot r = r,$$

where the first equality is from our definition of $\overline{\Delta}$. Consequently, we have $\boldsymbol{\beta}^{(t)} \in \mathcal{B}(r; \boldsymbol{\beta}^*)$. Now we check that by our choice of $\lambda_m^{(t+1)}$, inequality (A.11) holds. Note that

$$3\Delta_m + \frac{\alpha\mu\tau}{\gamma\Psi(\overline{\mathcal{S}})}\|\boldsymbol{\beta}^{(t)} - \boldsymbol{\beta}^*\| \leq \Delta + \frac{5\alpha\mu\tau}{\gamma\gamma_m}\frac{1 - \kappa^t}{1 - \kappa}\Delta + \frac{\alpha\mu\tau}{\gamma\Psi(\overline{\mathcal{S}})}\kappa^t\|\boldsymbol{\beta}^{(0)} - \boldsymbol{\beta}^*\|$$
$$\leq \Delta + \kappa\frac{1 - \kappa^t}{1 - \kappa}\Delta + \kappa^{t+1}\frac{\gamma_m}{5\Psi(\overline{\mathcal{S}})}\|\boldsymbol{\beta}^{(0)} - \boldsymbol{\beta}^*\| = \frac{1 - \kappa^{t+1}}{1 - \kappa}\Delta + \kappa^{t+1}\frac{\gamma_m}{5\Psi(\overline{\mathcal{S}})}\|\boldsymbol{\beta}^{(0)} - \boldsymbol{\beta}^*\| = \lambda_m^{(t+1)},$$

where the first inequality is from (A.14) and the second inequality is from the fact $\kappa \geq \kappa^* = 5\frac{\alpha\mu\tau}{\gamma\gamma_m}$. Therefore (A.11) holds for $t + 1$. Then applying (A.12) with $t + 1$ implies that

$$\|\boldsymbol{\beta}^{(t+1)} - \boldsymbol{\beta}^*\| \leq \frac{5\Psi(\overline{\mathcal{S}})}{\gamma_m}\frac{1 - \kappa^{t+1}}{1 - \kappa}\Delta + \kappa^{t+1}\|\boldsymbol{\beta}^{(0)} - \boldsymbol{\beta}^*\|.$$

Putting pieces together we prove that (A.14) holds for all $t \in [T]$ when Conditions 4 and 5 hold in every step. Applying probabilistic union bound, we reach the conclusion. $\qquad\square$

# B    Applications to Example Models

We fill in the details for the example models discussed in Section 5 in the main body: Gaussian mixture models, mixed linear regression (with sparse and low-rank regressors) and missing covariate regression.

## B.1 Gaussian Mixture Model

Recall that we consider the isotropic, balanced Gaussian Mixture Model with two components where sample $\mathbf{y}_i$ is generated from either $\mathcal{N}(\boldsymbol{\beta}^*, \sigma^2 \mathbf{I}_p)$ or $\mathcal{N}(-\boldsymbol{\beta}^*, \sigma^2 \mathbf{I}_p)$.

We focus on the high SNR regime where we assume $\mathrm{SNR} \geq \rho$ for some constant $\rho$. Note that the work in [11] provides empirical and theoretical evidence that in the low SNR regime, where the overlap density of two Gaussian clusters is small, the standard EM algorithm suffers from sublinear convergence asymptotically. Therefore the high SNR condition is necessary for showing exponential/linear convergence of the EM algorithm and our high dimensional variant. In particular, we are interested in quantizing estimation error using $\ell_2$ norm. We thus set the norm $\| \cdot \|$ in our framework to be $\| \cdot \|_2$ in this section. Recall that we set regularizer $\mathcal{R}$ to be the $\ell_1$ norm. For any subset $\mathcal{S} \subseteq \{1, \ldots, p\}$, $\ell_1$ norm is decomposable with respect to $(\mathcal{S}, \overline{\mathcal{S}})$. For any $\boldsymbol{\beta}^* \in \mathcal{B}_0(s; p)$, by letting $\mathcal{S} = \mathrm{supp}(\boldsymbol{\beta}^*), \overline{\mathcal{S}} = \mathrm{supp}(\boldsymbol{\beta}^*)$, we have $\Psi(\mathcal{S}) = \sqrt{s}$ and $\mathcal{C}(\mathcal{S}, \overline{\mathcal{S}}; \mathcal{R})$ corresponds to $\{\|\mathbf{u}_{\mathcal{S}^\perp}\|_1 \leq 2\|\mathbf{u}_{\mathcal{S}}\|_1 + 2\sqrt{s}\|\mathbf{u}\|_2\}$.

According to the $Q_n^{GMM}(\cdot|\cdot)$ introduced in (5.1), by taking its expectation, we have

$$Q^{GMM}(\boldsymbol{\beta}'|\boldsymbol{\beta}) = -\frac{1}{2}\mathbb{E}\left[w(Y;\boldsymbol{\beta})\|Y - \boldsymbol{\beta}'\|_2^2 + (1 - w(Y;\boldsymbol{\beta}))\|Y + \boldsymbol{\beta}'\|_2^2\right]. \tag{B.1}$$

We now check that Conditions 1-3 hold for $Q^{GMM}(\cdot|\cdot)$. We begin with proving the following result.

**Lemma 2** (Self consistency of GMM)**.** *Consider the Gaussian mixture model with $Q^{GMM}(\cdot|\cdot)$ given in* (B.1)*. For model parameter $\boldsymbol{\beta}^*$ we have*

$$\boldsymbol{\beta}^* = \arg\max_{\boldsymbol{\beta} \in \mathbb{R}^p} Q^{GMM}(\boldsymbol{\beta}|\boldsymbol{\beta}^*).$$

*Proof.* In this example, we have

$$\mathcal{M}(\boldsymbol{\beta}^*) = 2\mathbb{E}\left[w(Y;\boldsymbol{\beta}^*)Y\right] = 2\mathbb{E}\left[\frac{1}{1 + \exp(-\frac{2}{\sigma^2}\langle Z \cdot \boldsymbol{\beta}^* + W, \boldsymbol{\beta}^* \rangle)}(Z \cdot \boldsymbol{\beta}^* + W)\right],$$

where $W \sim \mathcal{N}(0, \sigma^2)$ and $Z$ has Rademacher distribution over $\{-1, 1\}$. Due to the rotation invariance of Gaussianity, without loss of generality, we assume $\boldsymbol{\beta}^* = Ae_1$. It is easy to check $\mathrm{supp}(\mathcal{M}(\boldsymbol{\beta}^*)) = \{1\}$. Moreover, the first coordinate of $\mathcal{M}(\boldsymbol{\beta}^*)$ takes form

$$(\mathcal{M}(\boldsymbol{\beta}^*))_1 = 2\mathbb{E}\left[\frac{1}{1 + \exp(-\frac{2}{\sigma^2}(AZ + W_1))}(AZ + W_1)\right] = A,$$

where the last equality follows by the substitution $X = W_1, Z = Z, \gamma = 0, a = A$ in Lemma 25. Therefore, $\mathcal{M}(\boldsymbol{\beta}^*) = \boldsymbol{\beta}^*$. $\qquad\square$

The above result shows that $Q^{GMM}(\cdot|\cdot)$ satisfies Condition 1. It is easy to see $\nabla^2 Q^{GMM}(\boldsymbol{\beta}'|\boldsymbol{\beta}) = -\mathbf{I}_p$, which implies that $Q^{GMM}(\cdot|\cdot)$ satisfies Condition 2 with parameters $(\gamma, \mu, r) = (1, 1, r)$ for any $r > 0$. Next we present a result showing that $Q^{GMM}(\cdot|\cdot)$ satisfies Condition 3 with arbitrarily small stability factor $\tau$ when SNR is sufficiently large.

**Lemma 3** (Gradient stability of GMM)**.** *Consider the Gaussian Mixture Model with $Q^{GMM}(\cdot|\cdot)$ given in* (B.1)*. Suppose $\mathrm{SNR} > \rho$. Function $Q^{GMM}(\cdot|\cdot)$ satisfies Condition 3 with parameters $(\tau, \|\boldsymbol{\beta}^*\|_2/4)$, where $\tau \leq \exp(-C\rho^2)$ for some absolute constant $C$.*

*Proof.* See the proof of Lemma 3 in [1]. $\qquad\square$

Now we turn to the conditions on $Q_n^{GMM}(\cdot|\cdot)$.

**Lemma 4** (RSC of GMM)**.** *Consider the Gaussian mixture model with any $\boldsymbol{\beta}^* \in \mathcal{B}_0(s; p)$ and $Q_n^{GMM}(\cdot|\cdot)$ given in* (5.1)*. For any $r > 0$, we have $Q_n^{GMM}(\cdot|\cdot)$ satisfies Condition 4 with parameters $(\gamma_n, \mathcal{S}, \overline{\mathcal{S}}, r, \delta)$, where*

$$\gamma_n = 1, \ \delta = 0, \ (\mathcal{S}, \overline{\mathcal{S}}) = (supp(\boldsymbol{\beta}^*), supp(\boldsymbol{\beta}^*)).$$

*Proof.* Although Condition 4 is a stochastic condition, for Gaussian mixture model in particular, it is satisfied deterministically. Note that

$$Q_n^{GMM}(\boldsymbol{\beta}'|\boldsymbol{\beta}) = -\frac{1}{2n}\sum_{i=1}^n \left[ w(\mathbf{y}_i;\boldsymbol{\beta})\|\mathbf{y}_i - \boldsymbol{\beta}'\|_2^2 + (1 - w(\mathbf{y}_i;\boldsymbol{\beta}))\|\mathbf{y}_i + \boldsymbol{\beta}'\|_2^2 \right].$$

We have that for any $\boldsymbol{\beta}', \boldsymbol{\beta} \in \mathbb{R}^p$, $\nabla^2 Q_n^{GMM}(\boldsymbol{\beta}'|\boldsymbol{\beta}) = -\mathbf{I}_p$, which implies that $Q_n^{GMM}(\boldsymbol{\beta}'|\boldsymbol{\beta})$ is strongly concave with parameter 1. Consequently, Condition 4 holds with $\gamma_n = 1$. □

This above result indicates that the restricted strong concavity condition holds deterministically in this example. The next lemma validates the statistical error condition and provides the corresponding parameters.

**Lemma 5** (Statistical error of GMM). *Consider the Gaussian mixture model with $Q_n^{GMM}(\cdot|\cdot)$ and $Q^{GMM}(\cdot|\cdot)$ given in (5.1) and (B.1) respectively. For any $r > 0$, $\delta \in (0,1)$ and some absolute constant $C$, Condition 5 holds with parameters $(\Delta_n, r, \delta)$ where*

$$\Delta_n = C(\|\boldsymbol{\beta}^*\|_\infty + \sigma)\sqrt{\frac{\log p + \log(2e/\delta)}{n}}.$$

*Proof.* Note that $\mathcal{R}^*$ is $\|\cdot\|_\infty$ in this example. Following the specific formulations of $Q_n^{GMM}(\cdot|\cdot)$ and $Q^{GMM}(\cdot|\cdot)$ in (5.1) and (B.1), we have

$$\nabla Q_n^{GMM}(\boldsymbol{\beta}^*|\boldsymbol{\beta}) - \nabla Q^{GMM}(\boldsymbol{\beta}^*|\boldsymbol{\beta}) = -\frac{1}{n}\sum_{i=1}^n \mathbf{y}_i + \frac{2}{n}\sum_{i=1}^n w(\mathbf{y}_i;\boldsymbol{\beta})\mathbf{y}_i - 2\mathbb{E}\left[w(Y;\boldsymbol{\beta})Y\right].$$

Therefore,

$$\left\|\nabla Q_n^{GMM}(\boldsymbol{\beta}^*|\boldsymbol{\beta}) - \nabla Q^{GMM}(\boldsymbol{\beta}^*|\boldsymbol{\beta})\right\|_\infty \leq \underbrace{\left\|\frac{1}{n}\sum_{i=1}^n \mathbf{y}_i\right\|_\infty}_{(a)} + \underbrace{\left\|\frac{2}{n}\sum_{i=1}^n w(\mathbf{y}_i;\boldsymbol{\beta})\mathbf{y}_i - 2\mathbb{E}\left[w(Y;\boldsymbol{\beta})Y\right]\right\|_\infty}_{(b)}$$

Next we bound the two terms $(a)$ and $(b)$ respectively.

**Term** $(a)$. Let $\boldsymbol{\zeta} := \frac{1}{n}\sum_{i=1}^n \mathbf{y}_i$. Let $\mathbf{y}_i = (y_{i,1}, \ldots, y_{i,p})^\top$ for all $i \in [n]$. Consider the $j$-th coordinate $\zeta_j$ of $\boldsymbol{\zeta}$, we have

$$\zeta_j = \frac{1}{n}\sum_{i=1}^n y_{i,j}.$$

Note that $\{y_{i,j}\}_{i=1}^n$ are independent copies of random variable $Y_j$ that is

$$Y_j = Z \cdot \beta_j^* + V, \tag{B.2}$$

where $Z$ is Rademacher random variable taking values in $\{-1, 1\}$ and $V$ has distribution $\mathcal{N}(0, \sigma^2)$. Since $Z \cdot \beta_j^*$ and $V$ are both sub-Gaussian random variables with norm $\|Z \cdot \beta_j^*\|_{\psi_2} \leq |\beta_j^*|$ and $\|V\|_{\psi_2} \lesssim \delta$. Following the rotation invariance sub-Gaussian random variables (e.g., Lemma 5.9 in [18]), we have that

$$\|Y_j\|_{\psi_2} \lesssim \sqrt{\|Z \cdot \beta_j^*\|_{\psi_2}^2 + \|V\|_{\psi_2}^2} \lesssim \sqrt{\|\boldsymbol{\beta}^*\|_\infty^2 + \sigma^2}.$$

Following the standard sub-Gaussian concentration argument in Lemma 19, there exists some constant $C$ such that for any $j \in [p]$ and all $t \geq 0$,

$$\Pr\left(|\zeta_j| \geq t\right) \leq e \cdot \exp\left(-\frac{Cnt^2}{\|\boldsymbol{\beta}^*\|_\infty^2 + \sigma^2}\right).$$

Then by applying union bound, we have

$$\Pr\left(\sup_{j\in[p]} |\zeta_j| \geq t\right) \leq pe \cdot \exp\left(-\frac{Cnt^2}{\|\boldsymbol{\beta}^*\|_\infty^2 + \sigma^2}\right).$$

Setting the right hand side to be $\delta$, we have that, with probability at least $1 - \delta/2$,

$$\left\| \frac{1}{n} \sum_{i=1}^{n} \mathbf{y}_i \right\|_{\infty} \lesssim (\|\boldsymbol{\beta}^*\|_{\infty} + \delta) \sqrt{\frac{\log p + \log(2e/\delta)}{n}}. \tag{B.3}$$

**Term** $(b)$. Now let $\boldsymbol{\zeta} := \frac{2}{n} \sum_{i=1}^{n} w(\mathbf{y}_i; \boldsymbol{\beta}) \mathbf{y}_i - 2\mathbb{E}\left[w(Y; \boldsymbol{\beta})Y\right]$. We also consider the $j$-th coordinate $\zeta_j$ of $\boldsymbol{\zeta}$, which takes form

$$\zeta_j = \frac{2}{n} \sum_{i=1}^{n} \left\{ w(\mathbf{y}_i; \boldsymbol{\beta}) y_{i,j} - \mathbb{E}(w(Y; \boldsymbol{\beta}) Y_j) \right\}.$$

Note that $w(\mathbf{y}_i; \boldsymbol{\beta}) y_{i,j} - \mathbb{E}(w(Y; \boldsymbol{\beta}) Y_j), i = 1, \ldots, n$ are independent copies of random variable $w(Y; \boldsymbol{\beta}) Y_j - \mathbb{E}(w(Y; \boldsymbol{\beta}) Y_j)$ where $Y_j$ is given in (B.2). We have shown that $Y_j$ is sub-Gaussian random variable. Note that $w(Y; \boldsymbol{\beta})$ is random variable taking values in $[0, 1]$. We thus always have

$$\Pr\left(|w(Y; \boldsymbol{\beta}) Y_j| \geq t\right) \leq \Pr(|Y_j| > t) \leq \exp(1 - Ct^2 / \|Y_j\|_{\psi_2}^2).$$

Using the equivalent properties of sub-Gaussian (see Lemma 5.5 in [18]) , we conclude that $w(Y; \boldsymbol{\beta}) Y_j$ is sub-Gaussian random variable with norm $\|w(Y; \boldsymbol{\beta}) Y_j\|_{\psi_2} \leq \|Y_j\|_{\psi_2} \lesssim \sqrt{\|\boldsymbol{\beta}^*\|_{\infty}^2 + \sigma^2}$. Following Lemma 21, we have $\|w(Y; \boldsymbol{\beta}) Y_j - \mathbb{E}\left[w(Y; \boldsymbol{\beta}) Y_j\right]\|_{\psi_2} \leq 2\|w(Y; \boldsymbol{\beta}) Y_j\|_{\psi_2}$. Using the concentration result from Lemma 19 yields that for any $j \in [p]$ and some constant $C$,

$$\Pr\left(|\zeta_j| \geq t\right) = \Pr\left\{ \left| \frac{2}{n} \sum_{i=1}^{n} w(\mathbf{y}_i; \boldsymbol{\beta}) y_{i,j} - \mathbb{E}(w(Y; \boldsymbol{\beta})Y) \right| > t \right\} \leq e \cdot \exp\left( -\frac{Cnt^2}{\|\boldsymbol{\beta}^*\|_{\infty}^2 + \sigma^2} \right).$$

Applying union bound over $p$ coordinates, we have

$$\Pr\left( \sup_{j \in [p]} |\zeta_j| > t \right) \leq pe \cdot \exp\left( -\frac{Cnt^2}{\|\boldsymbol{\beta}^*\|_{\infty}^2 + \sigma^2} \right),$$

which implies that, with probability at least $1 - \delta/2$,

$$\left\| \frac{2}{n} \sum_{i=1}^{n} w(\mathbf{y}_i; \boldsymbol{\beta}) \mathbf{y}_i - 2\mathbb{E}\left[w(Y; \boldsymbol{\beta})Y\right] \right\|_{\infty} \lesssim (\|\boldsymbol{\beta}^*\|_{\infty} + \sigma) \sqrt{\frac{\log p + \log(2e/\delta)}{n}}. \tag{B.4}$$

Putting (B.3) and (B.4) together completes the proof. $\qquad\square$

Now we give the guarantees of Algorithm 2 for the Gaussian mixture model.

*Proof of Corollary 1.* This result follows from Theorem 1. First, recall that the minimum contractive factor $\kappa^*$ is $\kappa^* = 5\frac{\alpha\mu\tau}{\gamma\gamma_{n/T}}$. For the $\ell_2$ norm, we have $\alpha = 1$. Following the fact that $(\gamma, \mu) = (1, 1)$ and Lemma 3-4, we have $\kappa^* \leq 20\exp(-C\rho^2)$ for some constant $C$. We further have $\kappa^* \leq \frac{1}{2}$ when $\rho$ is sufficiently large. Second, based on Lemma 5, we set $\delta = 1/p$ and choose $\Delta$ as $\Delta = C(\|\boldsymbol{\beta}^*\|_{\infty} + \sigma)\sqrt{T \log p/n}$ with sufficiently large $C$ such that $\Delta \geq 3\Delta_{n/T}$. By the assumption on $n/T$, we have that $\Delta \leq 3\overline{\Delta}$ where $\overline{\Delta} = \|\boldsymbol{\beta}^*\|_2/(240\sqrt{s})$ in this example. Finally, we choose $\lambda_{n/T}^{(0)} = \|\boldsymbol{\beta}^{(0)} - \boldsymbol{\beta}^*\|/(5\sqrt{s})$ by following Theorem 1. Packing up these ingredients and following Theorem 1, we have that by choosing any $\kappa \in [1/2, 3/4]$, $\|\boldsymbol{\beta}^{(t)} - \boldsymbol{\beta}^*\|_2 \leq \kappa^t\|\boldsymbol{\beta}^{(0)} - \boldsymbol{\beta}^*\|_2 + 5\sqrt{s}\Delta/(1 - \kappa)$, which thus completes the proof. $\qquad\square$

## B.2 Mixed Linear Regression

Recall that for Mixed Linear Regression (MLR) model, we consider two sets of model parameters: $\boldsymbol{\beta}^* \in \mathcal{B}_0(s; p)$ and $\boldsymbol{\Gamma}^* \in \mathbb{R}^{p_1 \times p_2}$ with $\text{rank}(\boldsymbol{\Gamma}^*) = \theta$. For the two settings, the population level analysis is identical under i.i.d. Gaussian covariate design. Without loss of generality, we begin with treating the model parameter as a vector $\boldsymbol{\beta}^* \in \mathbb{R}^p$ and validate Conditions 1-3 for $Q^{MLR}(\cdot|\cdot)$ in this example. Given function $Q_n^{MLR}(\cdot|\cdot)$ in (5.3), taking its expectation, yields

$$Q^{MLR}(\boldsymbol{\beta}'|\boldsymbol{\beta}) = -\frac{1}{2}\mathbb{E}\left[ w(Y, X; \boldsymbol{\beta})(Y - \langle X, \boldsymbol{\beta}'\rangle)^2 + (1 - w(Y, X; \boldsymbol{\beta}))(Y + \langle X, \boldsymbol{\beta}'\rangle)^2 \right]. \tag{B.5}$$

For now, we set the norm $\|\cdot\|$ in our framework to $\|\cdot\|_2$. We begin by checking the self consistency condition.

**Lemma 6** (Self consistency of MLR). *Consider mixed linear regression with model parameter $\boldsymbol{\beta}^* \in \mathbb{R}^p$ and $Q^{MLR}(\cdot|\cdot)$ given in (B.5). We have*

$$\boldsymbol{\beta}^* = \arg\max_{\boldsymbol{\beta} \in \mathbb{R}^p} Q^{MLR}(\boldsymbol{\beta}|\boldsymbol{\beta}^*).$$

*Proof.* In this example, we have

$$\mathcal{M}(\boldsymbol{\beta}^*) = 2\mathbb{E}\left[w(Y, X; \boldsymbol{\beta}^*)YX\right] = 2\mathbb{E}\left[\frac{1}{1 + \exp(-\frac{2(\langle X, Z \cdot \boldsymbol{\beta}^*\rangle + W)\langle X, \boldsymbol{\beta}^*\rangle}{\sigma^2})}(Z \cdot \boldsymbol{\beta}^* + W)X\right],$$

where $X \sim \mathcal{N}(\mathbf{0}, \mathbf{I}_p), W \sim \mathcal{N}(0, \sigma^2)$, $Z$ has Rademacher distribution. Due to the rotation invariance of Gaussianity, without loss of generality, we can assume $\boldsymbol{\beta}^* = A\boldsymbol{e}_1$. It is easy to check $\text{supp}(\mathcal{M}(\boldsymbol{\beta}^*)) = \{1\}$. Moreover,

$$(\mathcal{M}(\boldsymbol{\beta}^*))_1 = 2\mathbb{E}\left[\frac{1}{1 + \exp(-\frac{2}{\sigma^2}(AZX_1 + W)AX_1)}(AZX_1^2 + X_1W)\right] = \mathbb{E}(AX_1^2) = A,$$

where the second inequality follows by the substitution $X = W, Z = Z, \gamma = 0, a = AX_1$ in Lemma 25. We thus have $\mathcal{M}(\boldsymbol{\beta}^*) = \boldsymbol{\beta}^*$. $\qquad\square$

It is easy to check $\nabla^2 Q^{MLR}(\boldsymbol{\beta}'|\boldsymbol{\beta}) = -\mathbf{I}_p$. Therefore, $Q^{MLR}(\cdot|\cdot)$ satisfies Condition 2 with parameters $(\gamma, \mu, r) = (1, 1, r)$ for any $r > 0$. Similar to the Gaussian mixture model, we introduce the following SNR quantity to characterize the difficulty of the problem.

$$\text{SNR} := \|\boldsymbol{\beta}^*\|/\sigma.$$

The work in [7] shows that there exists an unavoidable phase transition of statistical rate from high SNR to low SNR. In detail, in low-dimensional setting, the obtainable statistical error is $\Omega(\sqrt{p/n})$ that matches the standard linear regression when $\text{SNR} \geq \rho$ for some constant $\rho$. Meanwhile, the unavoidable rate becomes $\Omega((p/n)^{1/4})$ when $\text{SNR} \ll \rho$. We conjecture such transition phenomenon still exists in high dimensional setting. For now we focus on the high SNR regime and show our algorithm achieves statistical rate that matches the standard sparse linear regression and low rank matrix recovery (up to logarithmic factor) in the end.

The following result shows Condition 3 holds with arbitrarily small stability factor $\tau$ when SNR is sufficiently large and the radius $r$ of ball $\mathcal{B}(r; \boldsymbol{\beta}^*)$ is sufficiently small.

**Lemma 7** (Gradient Stability of MLR). *Consider mixed linear regression model with function $Q^{MLR}(\cdot|\cdot)$ given in (B.5). For any $\omega \in [0, 1/4]$, let $r = \omega\|\boldsymbol{\beta}^*\|_2$. Suppose $\text{SNR} \geq \rho$ for some constant $\rho$. Then for any $\boldsymbol{\beta} \in \mathcal{B}(r; \boldsymbol{\beta}^*)$, we have*

$$\|\nabla Q^{MLR}(\mathcal{M}(\boldsymbol{\beta})|\boldsymbol{\beta}) - \nabla Q^{MLR}(\mathcal{M}(\boldsymbol{\beta})|\boldsymbol{\beta}^*)\|_2 \leq \tau\|\boldsymbol{\beta} - \boldsymbol{\beta}^*\|_2$$

*with*

$$\tau = \frac{17}{\rho} + 7.3\omega.$$

*Proof.* Recall that we hope to find $\tau$ such that for any $\boldsymbol{\beta} \in \mathcal{B}(r; \boldsymbol{\beta}^*)$

$$\|\nabla Q^{MLR}(\mathcal{M}(\boldsymbol{\beta})|\boldsymbol{\beta}) - \nabla Q^{MLR}(\mathcal{M}(\boldsymbol{\beta})|\boldsymbol{\beta}^*)\|_2 \leq \tau\|\boldsymbol{\beta} - \boldsymbol{\beta}^*\|_2.$$

In this example, we have

$$\mathcal{M}(\boldsymbol{\beta}) = 2\mathbb{E}\left[w(Y, X; \boldsymbol{\beta})YX\right],$$

and

$$\nabla Q^{MLR}(\boldsymbol{\beta}'|\boldsymbol{\beta}) = 2\mathbb{E}\left[w(Y, X; \boldsymbol{\beta})YX\right] - \boldsymbol{\beta}'.$$

Therefore,

$$\nabla Q^{MLR}(\mathcal{M}(\boldsymbol{\beta})|\boldsymbol{\beta}) - \nabla Q^{MLR}(\mathcal{M}(\boldsymbol{\beta})|\boldsymbol{\beta}^*)$$
$$= 2\mathbb{E}\left[w(Y, X; \boldsymbol{\beta})YX\right] - 2\mathbb{E}\left[w(Y, X; \boldsymbol{\beta}^*)YX\right] = 2\mathbb{E}\left[w(Y, X; \boldsymbol{\beta})YX\right] - \boldsymbol{\beta}^*,$$

where the last equality is from the self consistent property of $Q^{MLR}(\cdot|\cdot)$. Due to the rotation invariance of Gaussianity, without loss of generality, we assume $\boldsymbol{\beta}^* = A\boldsymbol{e}_1, \boldsymbol{\beta} = (1+\epsilon_1)A\boldsymbol{e}_1 + \epsilon_2 A\boldsymbol{e}_2$, where $A = \|\boldsymbol{\beta}^*\|_2, \|\boldsymbol{\beta} - \boldsymbol{\beta}^*\|_2 = A\sqrt{\epsilon_1^2 + \epsilon_2^2}$. Let random vector $T$ be

$$T := w(Y, X; \boldsymbol{\beta})YX - \frac{1}{2}\boldsymbol{\beta}^*.$$

Note that for any $\boldsymbol{\beta} \in \mathbb{R}^p$,

$$w(Y, X; \boldsymbol{\beta}) = \frac{\exp(-\frac{(Y-\langle X, \boldsymbol{\beta}\rangle)^2}{2\sigma^2})}{\exp(-\frac{(Y-\langle X, \boldsymbol{\beta}\rangle)^2}{2\sigma^2}) + \exp(-\frac{(Y+\langle X, \boldsymbol{\beta}\rangle)^2}{2\sigma^2})} = \frac{1}{1 + \exp(-\frac{2Y\langle X, \boldsymbol{\beta}\rangle}{\sigma^2})},$$

thereby

$$\begin{aligned}
T &= \frac{1}{1 + \exp(-\frac{2Y\langle X, \boldsymbol{\beta}\rangle}{\sigma^2})}YX - \frac{1}{2}\boldsymbol{\beta}^* \\
&= \frac{1}{1 + \exp(-\frac{2(ZAX_1 + W)(A(1+\epsilon_1)X_1 + \epsilon_2 X_2)}{\sigma^2})}(ZAX_1 + W)X - \frac{1}{2}A\boldsymbol{e}_1,
\end{aligned}$$

where $Z$ is Rademacher random variable taking values in $\{-1, 1\}$, $W$ is stochastic noise with distribution $\mathcal{N}(0, \sigma^2)$, $X_1$ and $X_2$ are the first two coordinates of $X$. It is easy to note that $\mathbb{E}[T_i] = 0$ for $i = 3, \ldots, p$. We focus on characterizing the first two coordinates $T_1, T_2$ of $T$.

**Coordinate $T_1$.**
First, we compute the expectation of $T_1$. Particularly we let $\gamma = \epsilon_1 + \epsilon_2 X_2/X_1$. Then we have

$$\begin{aligned}
|\mathbb{E}[T_1]| &= \left| \mathbb{E}\left[ \frac{X_1(W + ZAX_1)}{1 + \exp(-\frac{2AX_1(1+\gamma)}{\sigma^2}(W + ZAX_1))} - \frac{1}{2}AX_1^2 \right] \right| \\
&\leq \mathbb{E}\left[ |X_1| \cdot \left| \frac{(W + ZAX_1)}{1 + \exp(-\frac{2AX_1(1+\gamma)}{\sigma^2}(W + ZAX_1))} - \frac{1}{2}AX_1 \right| \right] \\
&= \mathbb{E}_{X_1, X_2}\left\{ |X_1| \cdot \mathbb{E}_{W,Z}\left[ \left| \frac{(W + ZAX_1)}{1 + \exp(-\frac{2AX_1(1+\gamma)}{\sigma^2}(W + ZAX_1))} - \frac{1}{2}AX_1 \right| \right] \right\} \\
&\leq \mathbb{E}_{X_1, X_2}\left[ |X_1| \cdot \min\left\{ \frac{1}{2}A \cdot |X_1\gamma| \cdot \exp(\frac{\gamma^2(AX_1)^2 - (AX_1)^2}{2\sigma^2}), \frac{\sigma}{\sqrt{2\pi}} + A|X_1| \right\} \right], \quad \text{(B.6)}
\end{aligned}$$

where the last inequality follows from Lemma 25 by replacing the parameters $(X, Z, a, \gamma)$ in the statement with $(W, Z, AX_1, \gamma)$. Let event $\mathcal{E}$ be $\mathcal{E} := \{\gamma^2 \leq 0.9\}$. Computing the expectation in (B.6) conditioning on $\mathcal{E}$ and $\mathcal{E}^c$ yields that

$$\begin{aligned}
|\mathbb{E}[T_1]| \leq &\mathbb{E}\left[ \frac{1}{2}|\gamma|AX_1^2 \exp(\frac{\gamma^2(AX_1)^2 - (AX_1)^2}{2\sigma^2}) \,\Big|\, \mathcal{E} \right] \cdot \Pr(\mathcal{E}) \\
&+ \mathbb{E}\left[ \frac{\sigma|X_1|}{\sqrt{2\pi}} + AX_1^2 \,\Big|\, \mathcal{E}^c \right] \cdot \Pr(\mathcal{E}^c).
\end{aligned} \quad \text{(B.7)}$$

We bound the two terms on the right hand side of the above inequality respectively. For the first term we have

$$\begin{aligned}
\mathbb{E}\left[ \frac{1}{2}|\gamma|AX_1^2 \exp(\frac{\gamma^2(AX_1)^2 - (AX_1)^2}{2\sigma^2}) \,\Big|\, \mathcal{E} \right] \cdot \Pr(\mathcal{E}) &\leq \mathbb{E}\left[ \frac{1}{2}|\gamma|AX_1^2 \exp(\frac{-(AX_1)^2}{20\sigma^2}) \,\Big|\, \mathcal{E} \right] \cdot \Pr(\mathcal{E}) \\
&\leq \mathbb{E}\left[ \frac{1}{2}|\gamma|AX_1^2 \exp(\frac{-(AX_1)^2}{20\sigma^2}) \right] \leq \mathbb{E}\left[ \frac{1}{2}A(|\epsilon_1| \cdot X_1^2 + |\epsilon_2 X_1 X_2|) \exp(-\frac{1}{20}\rho^2 X_1^2) \right] \\
&= \frac{1}{2}A\frac{|\epsilon_1|}{(1 + 0.1\rho^2)^{3/2}} + \frac{1}{\pi}A\frac{|\epsilon_2|}{1 + 0.1\rho^2} \leq \frac{1}{2}A\frac{1}{1 + 0.1\rho^2}(|\epsilon_1| + |\epsilon_2|), \quad \text{(B.8)}
\end{aligned}$$

where the third inequality is from $\|\boldsymbol{\beta}^*\|_2/\sigma \geq \rho$. For the second term in (B.7), first note that

$$\sqrt{\epsilon_1^2 + \epsilon_2^2} \leq \frac{\|\boldsymbol{\beta} - \boldsymbol{\beta}^*\|_2}{\|\boldsymbol{\beta}^*\|_2} \leq \omega \leq 1/4,$$

thereby
$$|\gamma| \leq |\epsilon_1| + |\epsilon_2| \cdot |X_2/X_1| \leq 1/4 + |\epsilon_2| \cdot |X_2/X_1|.$$
We define event $\mathcal{E}' := \{X_2^2/X_1^2 \geq (2.1\epsilon_2^2)^{-1}\}$. Note that $\mathcal{E}^c = \{\gamma^2 \geq 0.9\}$, we thus have $\mathcal{E}^c \subseteq \mathcal{E}'$, i.e., the occurrence of $\mathcal{E}^c$ must lead to the occurrence of $\mathcal{E}'$. For the second term in (B.7), we have

$$\mathbb{E}\left[\frac{\sigma|X_1|}{\sqrt{2\pi}} + AX_1^2 \,\bigg|\, \mathcal{E}^c\right] \cdot \Pr(\mathcal{E}^c) \leq \mathbb{E}\left[\frac{\sigma|X_1|}{\sqrt{2\pi}} + AX_1^2 \,\bigg|\, \mathcal{E}'\right] \cdot \Pr(\mathcal{E}')$$

$$\leq \mathbb{E}\left[\frac{\sigma|X_1|}{\sqrt{2\pi}} + \sqrt{2.1\epsilon_2^2}A|X_1 X_2| \,\bigg|\, \mathcal{E}'\right] \cdot \Pr(\mathcal{E}') \tag{B.9}$$

$$= \frac{\sigma}{\pi}\left[1 - \sqrt{\frac{1}{1 + 2.1\epsilon_2^2}}\right] + \sqrt{2.1\epsilon_2^2}A\frac{2}{\pi}\frac{2.1\epsilon_2^2}{1 + 2.1\epsilon_2^2} \leq \frac{\sqrt{2.1}\sigma}{\pi}|\epsilon_2| + \frac{2\sqrt{2.1}^3}{\pi}A|\epsilon_2|^3, \tag{B.10}$$

where the equality is from Lemma 24 by setting $C$ in the statement to be $\sqrt{2.1\epsilon_2^2}$.

Putting (B.8) and (B.9) together, we have

$$|\mathbb{E}[T_1]| \leq \frac{1}{2}A\frac{1}{1 + 0.1\rho^2}(|\epsilon_1| + |\epsilon_2|) + \frac{\sqrt{2.1}\sigma}{\pi}|\epsilon_2| + \frac{2\sqrt{2.1}^3}{\pi}A|\epsilon_2|^3. \tag{B.11}$$

**Coordinate $T_2$.**
Now we turn to the second coordinate $T_2$. Using $\mathbb{E}[X_1 X_2] = 0$, we have

$$|\mathbb{E}[T_2]| = \left|\mathbb{E}\left[\frac{X_2(W + ZAX_1)}{1 + \exp(-\frac{2AX_1(1+\gamma)}{\sigma^2}(W + ZAX_1))} - \frac{1}{2}AX_1 X_2\right]\right|$$

$$\leq \mathbb{E}\left[|X_2| \cdot \left|\frac{(W + ZAX_1)}{1 + \exp(-\frac{2AX_1(1+\gamma)}{\sigma^2}(W + ZAX_1))} - \frac{1}{2}AX_1\right|\right].$$

Similar to (B.6), using Lemma 25 leads to

$$|\mathbb{E}[T_2]| \leq \mathbb{E}\left[|X_2| \cdot \min\left\{\frac{1}{2}A \cdot |X_1\gamma| \cdot \exp(\frac{\gamma^2(AX_1)^2 - (AX_1)^2}{2\sigma^2}), \; \frac{\sigma}{\sqrt{2\pi}} + A|X_1|\right\}\right]$$

$$\leq \mathbb{E}\left[\frac{1}{2}A|\gamma| \cdot |X_1 X_2| \exp(\frac{\gamma^2(AX_1)^2 - (AX_1)^2}{2\sigma^2}) \,\bigg|\, \mathcal{E}\right] \cdot \Pr(\mathcal{E})$$

$$+ \mathbb{E}\left[\frac{\sigma|X_2|}{\sqrt{2\pi}} + A|X_1 X_2| \,\bigg|\, \mathcal{E}^c\right] \cdot \Pr(\mathcal{E}^c).$$

We bound the two terms in the right hand side of the above inequality respectively. For the first term, we have

$$\mathbb{E}\left[\frac{1}{2}A|\gamma| \cdot |X_1 X_2| \exp(\frac{\gamma^2(AX_1)^2 - (AX_1)^2}{2\sigma^2}) \,\bigg|\, \mathcal{E}\right] \cdot \Pr(\mathcal{E})$$

$$\leq \mathbb{E}\left[\frac{1}{2}A|\gamma| \cdot |X_1 X_2| \exp(\frac{-0.1(AX_1)^2}{2\sigma^2}) \,\big|\, \mathcal{E}\right] \cdot \Pr(\mathcal{E}) \leq \mathbb{E}\left[\frac{1}{2}A|\gamma| \cdot |X_1 X_2| \exp(\frac{-0.1(AX_1)^2}{2\sigma^2})\right]$$

$$\leq \mathbb{E}\left[\frac{1}{2}A\left(|\epsilon_1 X_1 X_2| + |\epsilon_2|X_2^2\right)\exp(-\frac{1}{20}\rho^2 X_1^2)\right] = \frac{1}{\pi}A\frac{|\epsilon_1|}{1 + 0.1\rho^2} + \frac{1}{2}A\frac{|\epsilon_2|}{\sqrt{1 + 0.1\rho^2}} \tag{B.12}$$

For the second term, recall that event $\mathcal{E}'$ is defined as $\{X_2^2/X_1^2 \geq (2.1\epsilon_2^2)^{-1}\}$, we have

$$\mathbb{E}\left[\frac{\sigma|X_2|}{\sqrt{2\pi}} + A|X_1 X_2| \,\bigg|\, \mathcal{E}^c\right] \cdot \Pr(\mathcal{E}^c) \leq \mathbb{E}\left[\frac{\sigma|X_2|}{\sqrt{2\pi}} + A|X_1 X_2| \,\bigg|\, \mathcal{E}'\right] \cdot \Pr(\mathcal{E}')$$

$$= \frac{\sigma}{\pi}\frac{\sqrt{2.1}\epsilon_2}{\sqrt{1 + 2.1\epsilon_2^2}} + \frac{2A}{\pi}\frac{2.1\epsilon_2^2}{1 + 2.1\epsilon_2^2} \leq \frac{\sqrt{2.1}\sigma}{\pi}|\epsilon_2| + \frac{4.2A}{\pi}\epsilon_2^2. \tag{B.13}$$

where the equality follows from Lemma 24 by setting $C$ in the statement to be $\sqrt{2.1\epsilon_2^2}$. Putting (B.12) and (B.13) together, we have

$$|\mathbb{E}[T_2]| \leq \frac{1}{\pi}A\frac{|\epsilon_1|}{1 + 0.1\rho^2} + \frac{1}{2}A\frac{|\epsilon_2|}{\sqrt{1 + 0.1\rho^2}} + \frac{\sqrt{2.1}\sigma}{\pi}|\epsilon_2| + \frac{4.2A}{\pi}\epsilon_2^2. \tag{B.14}$$

Now based on (B.11) and (B.14), we conclude that

$$\mathbb{E}\left[\|T\|_2\right] = \mathbb{E}\left[\sqrt{T_1^2 + T_2^2}\right] \leq \mathbb{E}\left[|T_1| + |T_2|\right]$$

$$\leq A\frac{1}{\sqrt{1+0.1\rho^2}}(|\epsilon_1| + |\epsilon_2|) + \frac{\sqrt{2.1}\sigma}{\pi}|\epsilon_2| + \frac{2\sqrt{2.1}^3}{\pi}A|\epsilon_2|^3 + \frac{\sqrt{2.1}\sigma}{\pi}|\epsilon_2| + \frac{4.2A}{\pi}\epsilon_2^2$$

$$\leq A\left(\frac{1}{\sqrt{1+0.1\rho^2}}(|\epsilon_1| + |\epsilon_2|) + |\epsilon_2|/\rho + 1.83\omega|\epsilon_2|\right)$$

$$\leq A(|\epsilon_1| + |\epsilon_2|) \cdot \left(\frac{4.2}{\rho} + 1.83\omega\right) \leq 2A\sqrt{\epsilon_1^2 + \epsilon_2^2} \cdot \left(\frac{4.2}{\rho} + 1.83\omega\right)$$

$$= 2\left(\frac{4.2}{\rho} + 1.83\omega\right)\|\boldsymbol{\beta} - \boldsymbol{\beta}^*\|_2.$$

Note that $\nabla Q^{MLR}(\mathcal{M}(\boldsymbol{\beta})|\boldsymbol{\beta}) - \nabla Q^{MLR}(\mathcal{M}(\boldsymbol{\beta})|\boldsymbol{\beta}^*) = 2T$, thereby we conclude that for any $\omega \leq 1/4$, $Q^{MLR}(\cdot|\cdot)$ satisfies gradient stability condition over $\mathcal{B}(\omega\|\boldsymbol{\beta}^*\|_2; \boldsymbol{\beta}^*)$ with parameter

$$\tau = \frac{17}{\rho} + 7.3\omega.$$

$\square$

In [1], it is proved that when $r = \frac{1}{32}\|\boldsymbol{\beta}^*\|_2$, there exists $\tau \in [0, 1/2]$ such that $Q^{MLR}(\cdot|\cdot)$ satisfies Condition 3 with parameter $\tau$ when $\rho$ is sufficiently large. Note that Lemma 7 recovers this result. Moreover, Lemma 7 provides an explicit function to characterize the relationship between $\tau$ and $\rho, \omega$.

Next we turn to validate the two technical conditions of $Q_n^{MLR}(\cdot|\cdot)$ and establish the computational and statistical guarantees of estimating mixed linear parameters in the high dimensional regime. We consider two different structures of linear parameters: (1) model parameter $\boldsymbol{\beta}^*$ is a sparse vector; (2) model parameter $\boldsymbol{\Gamma}^*$ is a low rank matrix. Note that we assume $X$ is a fully random Gaussian vector/matrix, thereby the population level conditions on $Q^{MLR}(\cdot|\cdot)$ hold in both settings.

**Sparse Recovery.** We assume model parameter $\boldsymbol{\beta}^*$ is $s$-sparse, i.e., $\boldsymbol{\beta}^* \in \mathcal{B}_0(s; p)$. Recall that, in order to serve sparse structure, we choose $\mathcal{R}$ to be $\ell_1$ norm. Setting $\mathcal{S} = \overline{\mathcal{S}} = \text{supp}(\boldsymbol{\beta}^*)$, set $\mathcal{C}(\mathcal{S}, \overline{\mathcal{S}}; \mathcal{R})$ corresponds to $\{\mathbf{u} : \|\mathbf{u}_{\mathcal{S}^\perp}\|_1 \leq 2\|\mathbf{u}_{\mathcal{S}}\|_1 + 2\sqrt{s}\|\mathbf{u}\|_2\}$. Restricted concavity of $Q^{MLR}(\cdot|\cdot)$ is validated in the following result.

**Lemma 8** (RSC of MLR with sparsity)**.** *Consider mixed linear regression with any model parameter* $\boldsymbol{\beta}^* \in \mathcal{B}_0(s; p)$ *and function* $Q_n^{MLR}(\cdot|\cdot)$ *defined in (5.3). There exit absolute constants* $\{C_i\}_{i=0}^3$ *such that, if* $n \geq C_0 s \log p$, *then for any* $r > 0$, $Q_n^{MLR}(\cdot|\cdot)$ *satisfies Condition 4 with parameters* $(\gamma_n, \mathcal{S}, \overline{\mathcal{S}}, r, \delta)$, *where*

$$\gamma_n = \frac{1}{3}, \ (\mathcal{S}, \overline{\mathcal{S}}) = (\text{supp}(\boldsymbol{\beta}^*), \text{supp}(\boldsymbol{\beta}^*)), \ \delta = C_1 \exp(-C_2 n).$$

*Proof.* Recall that

$$Q_n^{MLR}(\boldsymbol{\beta}'|\boldsymbol{\beta}) = -\frac{1}{2n}\sum_{i=1}^n \left[w(y_i, \mathbf{x}_i; \boldsymbol{\beta})(y_i - \langle \mathbf{x}_i, \boldsymbol{\beta}'\rangle)^2 + (1 - w(y_i, \mathbf{x}_i; \boldsymbol{\beta}))(y_i + \langle \mathbf{x}_i, \boldsymbol{\beta}'\rangle)^2\right].$$

For any $\boldsymbol{\beta}, \boldsymbol{\beta}' \in \mathbb{R}^p$, we have

$$Q_n^{MLR}(\boldsymbol{\beta}'|\boldsymbol{\beta}) - Q_n^{MLR}(\boldsymbol{\beta}^*|\boldsymbol{\beta}) - \langle \nabla Q_n^{MLR}(\boldsymbol{\beta}^*|\boldsymbol{\beta}), \boldsymbol{\beta}' - \boldsymbol{\beta}^*\rangle = -\frac{1}{2}(\boldsymbol{\beta}' - \boldsymbol{\beta}^*)^\top \left(\frac{1}{n}\sum_{i=1}^n \mathbf{x}_i\mathbf{x}_i^\top\right)(\boldsymbol{\beta}' - \boldsymbol{\beta}^*).$$

(B.15)

Note that we want to find $\gamma_n$ such that the right hand side of (B.15) is less than $-\frac{\gamma_n}{2}\|\boldsymbol{\beta}' - \boldsymbol{\beta}\|_2^2$ for any $\boldsymbol{\beta}' - \boldsymbol{\beta}^* \in \mathcal{C}(\mathcal{S}, \overline{\mathcal{S}}; \mathcal{R})$. In this example, we have $\mathcal{C}(\mathcal{S}, \overline{\mathcal{S}}; \mathcal{R}) = $

$\{\mathbf{u} \in \mathbb{R}^p : \|\mathbf{u}_{\mathcal{S}^{\perp}}\|_1 \leq 2\|\mathbf{u}_{\mathcal{S}}\|_1 + 2\sqrt{s}\|\mathbf{u}\|_2\}$. It is sufficient to prove that the sample covariance matrix has restricted eigenvalues over set $\mathcal{C}(\mathcal{S}, \overline{\mathcal{S}}; \mathcal{R})$. The following statement is follows by the substitution $\mathbf{\Sigma} = \mathbf{I}_p$ and $X = X$ in Lemma 23: there exist constants $\{C_i\}_{i=0}^2$ such that

$$\frac{1}{n}\sum_{i=1}^n \langle \mathbf{x}_i, \mathbf{u}\rangle^2 \geq \frac{1}{2}\|\mathbf{u}\|_2^2 - C_0\frac{\log p}{n}\|\mathbf{u}\|_1^2, \text{ for all } \mathbf{u} \in \mathbb{R}^p, \tag{B.16}$$

with probability at least $1 - C_1 \exp(-C_2 n)$. For any $\mathbf{u} \in \mathcal{C}(\mathcal{S}, \overline{\mathcal{S}}; \mathcal{R})$, we have

$$\|\mathbf{u}\|_1 = \|\mathbf{u}_{\mathcal{S}}\|_1 + \|\mathbf{u}_{\mathcal{S}^{\perp}}\|_1 \leq 3\|\mathbf{u}_{\mathcal{S}}\|_1 + 2\sqrt{s}\|\mathbf{u}\|_2 \leq 5\sqrt{s}\|\mathbf{u}\|_2.$$

Applying (B.16) yields that

$$\frac{1}{n}\sum_{i=1}^n \langle \boldsymbol{x}_i, \mathbf{u}\rangle^2 \geq \frac{1}{2}\|\mathbf{u}\|_2^2 - 25C_0\frac{s\log p}{n}\|\mathbf{u}\|_2^2, \text{ for all } \mathbf{u} \in \mathcal{C}(\mathcal{S}, \overline{\mathcal{S}}; \mathcal{R}).$$

Consequently, when $n \geq C_3 s \log p$ for sufficiently large $C_3$, $\frac{1}{n}\sum_{i=1}^n \langle \boldsymbol{x}_i, \mathbf{u}\rangle^2 \geq 1/3\|\mathbf{u}\|_2^2$, which implies $\gamma_n = 1/3$. □

Lemma 8 states that using $n = O(s \log p)$ samples makes $Q_n^{MLR}(\cdot|\cdot)$ be strongly concave over $\mathcal{C}$ with high probability.

**Lemma 9** (Statistical error of MLR with sparsity). *Consider mixed linear regression model with any* $\boldsymbol{\beta}^* \in \mathcal{B}_0(s; p)$ *and functions* $Q_n^{MLR}(\cdot|\cdot), Q^{MLR}(\cdot|\cdot)$ *defined in (5.3) and (B.5) respectively. There exist constants $C$ and $C_1$ such that, for any $r > 0$ and $\delta \in (0, 1)$, if $n \geq C_1(\log p + \log(6/\delta))$, then*

$$\|\nabla Q_n^{MLR}(\boldsymbol{\beta}^*|\boldsymbol{\beta}) - \nabla Q^{MLR}(\boldsymbol{\beta}^*|\boldsymbol{\beta})\|_\infty \leq C(\|\boldsymbol{\beta}^*\|_2 + \delta)\sqrt{\frac{\log p + \log(6/\delta)}{n}} \text{ for all } \boldsymbol{\beta} \in \mathcal{B}(r; \boldsymbol{\beta}^*)$$

*with probability at least $1 - \delta$.*

*Proof.* According to the formulations of $Q_n^{MLR}(\cdot|\cdot)$ and $Q^{MLR}(\cdot|\cdot)$ in (5.3) and (B.5), we have

$$\nabla Q_n^{MLR}(\boldsymbol{\beta}^*|\boldsymbol{\beta}) - \nabla Q^{MLR}(\boldsymbol{\beta}^*|\boldsymbol{\beta})$$
$$= \boldsymbol{\beta}^* - \left(\frac{1}{n}\sum_{i=1}^n \mathbf{x}_i\mathbf{x}_i^\top\right)\boldsymbol{\beta}^* + \frac{2}{n}\sum_{i=1}^n w(y_i, \mathbf{x}_i; \boldsymbol{\beta})y_i\mathbf{x}_i - 2\mathbb{E}\left[w(Y, X; \boldsymbol{\beta})YX\right] - \frac{1}{n}\sum_{i=1}^n y_i\mathbf{x}_i. \tag{B.17}$$

So

$$\|\nabla Q_n^{MLR}(\boldsymbol{\beta}^*|\boldsymbol{\beta}) - \nabla Q^{MLR}(\boldsymbol{\beta}^*|\boldsymbol{\beta})\|_\infty$$
$$\leq \underbrace{\left\|\frac{1}{n}\sum_{i=1}^n y_i\mathbf{x}_i\right\|_\infty}_{(a)} + \underbrace{\left\|\boldsymbol{\beta}^* - \left(\frac{1}{n}\sum_{i=1}^n \mathbf{x}_i\mathbf{x}_i^\top\right)\boldsymbol{\beta}^*\right\|_\infty}_{(b)} + \underbrace{\left\|\frac{2}{n}\sum_{i=1}^n w(y_i, \mathbf{x}_i; \boldsymbol{\beta})y_i\mathbf{x}_i - 2\mathbb{E}\left[w(Y, X; \boldsymbol{\beta})YX\right]\right\|_\infty}_{(c)}.$$

Next we bound the above three terms $(a), (b)$ and $(c)$ respectively.

**Term** $(a)$. We let vector $\boldsymbol{\zeta} := \frac{1}{n}\sum_{i=1}^n y_i\mathbf{x}_i$. Consider $j$th coordinate of $\boldsymbol{\zeta}$. For any $j \in [p]$, we have

$$\zeta_j = \frac{1}{n}\sum_{i=1}^n y_i x_{i,j},$$

where $x_{i,j}$ is the $j$th coordinate of $\mathbf{x}_i$. Note that $\{y_i x_{ij}\}_{i=1}^n$ are independent copies of random variables $(\langle X, Z \cdot \boldsymbol{\beta}^*\rangle + W)X_j$ where $X \sim \mathcal{N}(0, \mathbf{I}_p)$, $W \sim \mathcal{N}(0, \sigma^2)$ and $Z$ has Rademacher distribution. $\langle X, Z \cdot \boldsymbol{\beta}^*\rangle + W$ is sub-Gaussian random variable that has norm $\|\langle X, Z \cdot \boldsymbol{\beta}^*\rangle + W\|_{\psi_2} \lesssim \sqrt{\|\boldsymbol{\beta}^*\|_2^2 + \sigma^2}$. Also $X_j$ is sub-Gaussian random variable that has norm $\|X_j\|_{\psi_2} \lesssim 1$. Then based on Lemma 22, $(\langle X, Z \cdot \boldsymbol{\beta}^*\rangle + W)X_j$ is sub-exponential with norm $\|(\langle X, Z \cdot \boldsymbol{\beta}^*\rangle + W)X_j\|_{\psi_1} \lesssim$

$\sqrt{\|\boldsymbol{\beta}^*\|_2^2 + \sigma^2}$. Following standard concentration result of sub-exponential random variables (e.g., Lemma 20), there exists some constant $C$ such that the following inequality

$$\Pr\left(|\zeta_j| \geq t\right) \leq 2\exp\left(-C\frac{t^2 n}{\|\boldsymbol{\beta}^*\|_2^2 + \sigma^2}\right)$$

holds for sufficiently small $t > 0$. Therefore,

$$\Pr\left(\sup_{j \in [p]} |\zeta_j| > t\right) \leq 2p\exp\left(-C\frac{t^2 n}{\|\boldsymbol{\beta}^*\|_2^2 + \sigma^2}\right).$$

Setting the right hand side to be $\delta/3$, we have that, when $n$ is sufficiently large (i.e., $n \geq C(\log p + \log(6/\delta))$ for some constant $C$), with probability at least $1 - \delta/3$.

$$\left\|\frac{1}{n}\sum_{i=1}^{n} y_i \mathbf{x}_i\right\|_\infty \lesssim (\|\boldsymbol{\beta}^*\|_2 + \sigma)\sqrt{\frac{\log p + \log(6/\delta)}{n}}. \tag{B.18}$$

**Term** $(b)$. Now we let $\boldsymbol{\zeta} = \boldsymbol{\beta}^* - \frac{1}{n}\mathbf{x}_i\mathbf{x}_i\boldsymbol{\beta}^*$. For any $j \in [p]$,

$$\zeta_j = \frac{1}{n}\sum_{i=1}^{n} \beta_j^* - x_{i,j}\langle \mathbf{x}_i, \boldsymbol{\beta}^*\rangle.$$

Note that $\{\beta_j^* - x_{i,j}\langle \mathbf{x}_i, \boldsymbol{\beta}^*\rangle\}_{i=1}^{n}$ are independent copies of random variable $\beta_j^* - X_j\langle X, \boldsymbol{\beta}^*\rangle$. Using similar analysis in bounding term $(a)$, we claim that $\beta_j^* - X_j\langle X, \boldsymbol{\beta}^*\rangle$ is centered sub-exponential random variable with norm $\|\beta_j^* - X_j\langle X, \boldsymbol{\beta}^*\rangle\|_{\psi_1} \lesssim \|\boldsymbol{\beta}^*\|_2$. Therefore, for sufficiently small $t$ and some constant $C$,

$$\Pr\left(|\zeta_j| \geq t\right) \leq 2\exp\left(-C\frac{t^2 n}{\|\boldsymbol{\beta}^*\|_2^2}\right).$$

Using union bound implies that

$$\Pr\left(\sup_{j \in [p]} |\zeta_j| \geq t\right) \leq 2p \cdot \exp\left(-C\frac{t^2 n}{\|\boldsymbol{\beta}^*\|_2^2}\right).$$

Setting the right hand side to be $\delta/3$, we have that, when $n$ is sufficiently large,

$$\left\|\boldsymbol{\beta}^* - \left(\frac{1}{n}\sum_{i=1}^{n} \mathbf{x}_i\mathbf{x}_i^\top\right)\boldsymbol{\beta}^*\right\|_\infty \lesssim \|\boldsymbol{\beta}^*\|_2\sqrt{\frac{\log p + \log(6/\delta)}{n}} \tag{B.19}$$

holds with probability at least $1 - \delta/3$.

**Term** $(c)$. The analysis of this term is similar to the previous two terms. We let

$$\boldsymbol{\zeta} := \frac{1}{n}\sum_{i=1}^{n} w(\boldsymbol{y}_i, \mathbf{x}_i; \boldsymbol{\beta})y_i\mathbf{x}_i - \mathbb{E}\left[w(Y, X; \boldsymbol{\beta})YX\right].$$

For any $j \in [p]$,

$$\zeta_j = \frac{1}{n}\sum_{i=1}^{n} w(\boldsymbol{y}_i, \mathbf{x}_i; \boldsymbol{\beta})y_i x_{i,j} - \mathbb{E}\left[w(Y, X; \boldsymbol{\beta})YX\right].$$

Note that $\{w(\boldsymbol{y}_i, \mathbf{x}_i; \boldsymbol{\beta})y_i x_{i,j}\}_{i=1}^{n}$ are independent copies of random variable $w(Y, X; \boldsymbol{\beta})YX_j$. We know that $Y$ is sub-Gaussian with norm $\|Y\|_{\psi_2} \lesssim \sqrt{\|\boldsymbol{\beta}^*\|_2^2 + \sigma^2}$. Since $w(Y, X; \boldsymbol{\beta})$ is bounded, $w(Y, X; \boldsymbol{\beta})Y$ is also sub-Gaussian. Consequently, $w(Y, X; \boldsymbol{\beta})YX_j$ is sub-exponential. By standard concentration result, for some constant $C$ and sufficiently small $t$,

$$\Pr(|\zeta_j| \geq t) \leq 2\exp\left(-C\frac{nt^2}{\|\boldsymbol{\beta}^*\|_2^2 + \sigma^2}\right).$$

Therefore,

$$\Pr(\sup_{j \in [p]} |\zeta_j| \geq t) \leq 2\exp\left(-C\frac{nt^2}{\|\boldsymbol{\beta}^*\|_2^2 + \sigma^2}\right).$$

Setting the right hand side to be $\delta/3$, we have that, when $n$ is sufficiently large,

$$\left\| \frac{2}{n} \sum_{i=1}^{n} w(\boldsymbol{y}_i, \mathbf{x}_i; \boldsymbol{\beta}) y_i \mathbf{x}_i - 2\mathbb{E}\left[w(Y, X; \boldsymbol{\beta}) Y X\right] \right\|_{\infty} \lesssim (\|\boldsymbol{\beta}^*\|_2 + \delta) \sqrt{\frac{\log p + \log(6/\delta)}{n}} \quad \text{(B.20)}$$

with probability at least $1 - \delta/3$.

Putting (B.18), (B.19) and (B.20) together completes the proof. $\qquad\square$

Lemma 9 implies Condition 5 hold with parameters $\Delta_n = O\left((\|\boldsymbol{\beta}^*\|_2 + \delta)\sqrt{\log p/n}\right)$, any $r > 0$ and $\delta = 1/p$. Putting all the ingredients together leads to the following guarantee about sparse recovery in mixed linear regression using regularized EM algorithm.

*Proof of Corollary 2.* The result follows from Theorem 1. First, we note that the minimum contractive factor $\kappa^* = 5\frac{\alpha\mu\tau}{\gamma\gamma_{n/T}} = 15\tau$ in this example since $\alpha = 1, \mu = \gamma = 1$ and $\gamma_{n/T} = 1/3$ w.h.p when $n \gtrsim s \log p$ (see Lemma 8). Following Lemma 7, $\kappa^* \leq 1/2$ when $w \leq 1/240$ and $\rho$ is sufficiently large. Second, by choosing $n/T \gtrsim s \log p$, we have $\Delta_{n/T} \lesssim (\|\boldsymbol{\beta}^*\|_2 + \delta)\sqrt{\frac{T \log p}{n}}$ w.h.p., as proved in Lemma 9. Lastly, we have $\Delta \leq 3\overline{\Delta}$ by assuming $n/T \gtrsim \left[(\|\boldsymbol{\beta}^*\|_2 + \delta)/\|\boldsymbol{\beta}^*\|_2\right]^2 s \log p$. Putting these ingredients together and plugging the established parameters into (4.6) complete the proof. $\qquad\square$

**Low Rank Recovery.** In the sequel, we assume model parameter $\boldsymbol{\Gamma}^* \in \mathbb{R}^{p_1 \times p_2}$ is a low rank matrix that has $\text{rank}(\boldsymbol{\Gamma}^*) = \theta \ll \min\{p_1, p_2\}$. We focus on measuring the estimation error in Frobenius norm thus set $\|\cdot\|$ in our framework to be $\|\cdot\|_F$. Note that by treating $\boldsymbol{\Gamma}^*$ as a vector, Frobenius norm is equivalent to $\ell_2$ norm, thereby we still have Lemma 6-7 in this setting. Moreover, SNR is similarly defined as

$$\text{SNR} := \|\boldsymbol{\Gamma}^*\|_F / \sigma.$$

In order to serve the low rank structure, we choose $\mathcal{R}$ to be nuclear norm $\|\cdot\|_*$. For any matrix $\mathbf{M}$, we let $\text{row}(\mathbf{M})$ denote the subspace spanned by the rows of $\mathbf{M}$ and $\text{col}(\mathbf{M})$ denote the subspace spanned by the columns of $\mathbf{M}$. Moreover, for subspace represented by the columns of matrix $\mathbf{U}$, we denote the subspace orthogonal to $\mathbf{U}$ as $\mathbf{U}^\perp$. For $\boldsymbol{\Gamma}^*$ with singular value decomposition $\mathbf{U}^* \boldsymbol{\Sigma} \mathbf{V}^{*\top}$, we thus let

$$\mathcal{S} = \left\{\mathbf{M} \in \mathbb{R}^{p_1 \times p_2} : \text{col}(\mathbf{M}) \subseteq \mathbf{U}^*, \text{row}(\mathbf{M}) \subseteq \mathbf{V}^*\right\} \quad \text{(B.21)}$$

and

$$\overline{\mathcal{S}}^\perp = \left\{\mathbf{M} \in \mathbb{R}^{p_1 \times p_2} : \text{col}(\mathbf{M}) \subseteq \mathbf{U}^{*\perp}, \text{row}(\mathbf{M}) \subseteq \mathbf{V}^{*\perp}\right\}. \quad \text{(B.22)}$$

So $\mathcal{S}$ contains all matrices with rows (and columns) living in the row (and column) space of $\boldsymbol{\Gamma}^*$. Subspace $\overline{\mathcal{S}}^\perp$ contains all matrices with rows (and columns) orthogonal to the row (and column) space of $\boldsymbol{\Gamma}^*$. Nuclear norm is decomposable with respect to $(\mathcal{S}, \overline{\mathcal{S}})$. We have $\Psi(\overline{\mathcal{S}}) = \sup_{\mathbf{M} \in \overline{\mathcal{S}} \setminus \{\mathbf{0}\}} \|\mathbf{M}\|_* / \|\mathbf{M}\|_F \leq \sqrt{2\theta}$ since matrix in $\overline{\mathcal{S}}$ has rank at most $2\theta$. Similar to Lemma 8 and 9 for sparse structure, we have the following two results for low rank structure.

**Lemma 10** (RSC of MLR with low rank structure). *Consider mixed linear regression with model parameter $\boldsymbol{\Gamma}^* \in \mathbb{R}^{p_1 \times p_2}$ that has $\text{rank}(\boldsymbol{\Gamma}^*) = \theta$. There exists constants $\{C_i\}_{i=0}^{2}$ such that, if $n \geq C_0 \theta \max\{p_1, p_2\}$, then for any $\theta \in (0, \min\{p_1, p_2\})$, $Q_n^{MLR}(\cdot|\cdot)$ satisfies Condition 4 with parameters $(\gamma_n, \mathcal{S}, \overline{\mathcal{S}}, r, \delta)$, where $(\mathcal{S}, \overline{\mathcal{S}})$ are given in (B.21) and (B.22),*

$$\gamma_n = \frac{1}{20}, \ \delta = C_1 \exp(-C_2 n).$$

*Proof.* Similarly to (B.15), we have that for any $\boldsymbol{\Gamma}', \boldsymbol{\Gamma} \in \mathbb{R}^{p_1 \times p_2}$,

$$Q_n^{MLR}(\boldsymbol{\Gamma}'|\boldsymbol{\Gamma}) - Q_n^{MLR}(\boldsymbol{\Gamma}^*|\boldsymbol{\Gamma}) - \langle \nabla Q_n^{MLR}(\boldsymbol{\Gamma}^*|\boldsymbol{\Gamma}), \boldsymbol{\Gamma}' - \boldsymbol{\Gamma}^* \rangle = -\frac{1}{2n} \sum_{i=1}^{n} \langle \mathbf{X}_i, \boldsymbol{\Gamma}' - \boldsymbol{\Gamma}^* \rangle^2. \quad \text{(B.23)}$$

Note that $\boldsymbol{\Gamma}' - \boldsymbol{\Gamma}^* \in \mathcal{C}(\mathcal{S}, \overline{\mathcal{S}}; \|\cdot\|_*)$. Let $\Theta := \boldsymbol{\Gamma}' - \boldsymbol{\Gamma}^*$, we thus have

$$\|\Theta_{\overline{\mathcal{S}}^\perp}\|_* \leq 2 \cdot \|\Theta_{\overline{\mathcal{S}}}\|_* + 2 \cdot \sqrt{2\theta}\|\Theta\|_F.$$

We make use of the following result.

**Lemma 11.** *Let $\{\mathbf{X}_i\}_{i=1}^n$ be $n$ independent samples of random matrix $X \in \mathbb{R}^{p_1 \times p_2}$ where the entries are i.i.d. Gaussian random variable with distribution $\mathcal{N}(0,1)$. There exits constants $C_1, C_2$ such that*

$$\frac{1}{\sqrt{n}}\sqrt{\sum_{i=1}^n \langle \mathbf{X}_i, \Theta \rangle^2} \geq \frac{1}{4}\|\Theta\|_F - 12\left(\sqrt{\frac{p_1}{n}} + \sqrt{\frac{p_2}{n}}\right)\|\Theta\|_*, \text{ for all } \Theta \in \mathbb{R}^{p_1 \times p_2},$$

*with probability at least $1 - C_1 \exp(-C_2 n)$.*

*Proof.* See Proposition 1 in [13] for detailed proof. □

Then for our $\Theta$, using the above result yields that

$$\frac{1}{\sqrt{n}}\sqrt{\sum_{i=1}^n \langle \mathbf{X}_i, \Theta \rangle^2} \geq \frac{1}{4}\|\Theta\|_F - 12\left(\sqrt{\frac{p_1}{n}} + \sqrt{\frac{p_2}{n}}\right)(\|\Theta_{\overline{\mathcal{S}}}\|_* + \|\Theta_{\overline{\mathcal{S}}^\perp}\|_*)$$

$$\geq \frac{1}{4}\|\Theta\|_F - 12\left(\sqrt{\frac{p_1}{n}} + \sqrt{\frac{p_2}{n}}\right)\left(3\|\Theta_{\overline{\mathcal{S}}}\|_* + 2\sqrt{2r}\|\Theta\|_F\right)$$

$$\geq \left[\frac{1}{4} - 60\sqrt{2\theta}\left(\sqrt{\frac{p_1}{n}} + \sqrt{\frac{p_2}{n}}\right)\right]\|\Theta\|_F.$$

So when $n \geq C\theta \max\{p_1, p_2\}$ for sufficient large $C$, we have $\frac{1}{\sqrt{n}}\sqrt{\sum_{i=1}^n \langle \mathbf{X}_i, \Theta \rangle^2} \geq \|\Theta\|_F/\sqrt{20}$. Plugging this result back into (B.23) gives us $\gamma_n = 1/20$ thus completes the proof. □

**Lemma 12** (Statistical error of MLR with low rank structure). *Consider the mixed linear regression with any $\mathbf{\Gamma}^* \in \mathbb{R}^{p_1 \times p_2}$. There exists constants $C$ and $C_1$ such that, for any fixed $\mathbf{\Gamma} \in \mathbb{R}^{p_1 \times p_2}$ and $\delta \in (0,1)$, if $n \geq C_1(p_1 + p_2 + \log(6/\delta))$, then*

$$\|\nabla Q^{MLR}(\mathbf{\Gamma}^*|\mathbf{\Gamma}) - \nabla Q_n^{MLR}(\mathbf{\Gamma}^*|\mathbf{\Gamma})\|_2 \leq C(\|\mathbf{\Sigma}^*\|_F + \sigma)\sqrt{\frac{p_1 + p_2 + \log(6/\delta)}{n}}$$

*with probability at least $1 - \delta$.*

*Proof.* Parallel to (B.17), we have

$$\nabla Q_n^{MLR}(\mathbf{\Gamma}^*|\mathbf{\Gamma}) - \nabla Q^{MLR}(\mathbf{\Gamma}^*|\mathbf{\Gamma})$$

$$= \mathbf{\Gamma}^* - \frac{1}{n}\sum_{i=1}^n \langle \mathbf{X}_i, \mathbf{\Gamma}^* \rangle \mathbf{\Gamma}^* + \frac{2}{n}\sum_{i=1}^n w(y_i, \mathbf{X}_i; \mathbf{\Gamma})y_i\mathbf{X}_i - 2\mathbb{E}[w(Y, X; \mathbf{\Gamma})YX] - \frac{1}{n}\sum_{i=1}^n y_i\mathbf{X}_i.$$

The dual norm of nuclear norm is spectral norm. So we are interested in bounding the following term for fixed $\mathbf{\Gamma}$:

$$\left\|\nabla Q_n^{MLR}(\mathbf{\Gamma}^*|\mathbf{\Gamma}) - \nabla Q^{MLR}(\mathbf{\Gamma}^*|\mathbf{\Gamma})\right\|_2$$

$$\leq \underbrace{\left\|\frac{1}{n}\sum_{i=1}^n y_i\mathbf{X}_i\right\|_2}_{U_1} + \underbrace{\left\|\mathbf{\Gamma}^* - \frac{1}{n}\sum_{i=1}^n \langle \mathbf{X}_i, \mathbf{\Gamma}^* \rangle \mathbf{X}_i\right\|_2}_{U_2} + \underbrace{\left\|\frac{2}{n}\sum_{i=1}^n w(y_i, \mathbf{X}_i; \mathbf{\Gamma})y_i\mathbf{X}_i - 2\mathbb{E}[w(Y, X; \mathbf{\Gamma})YX]\right\|_2}_{U_3}.$$

Next we bound the three terms $U_1, U_2$ and $U_3$ respectively.

**Term $U_1$.** We first note that

$$U_1 = \sup_{\substack{\mathbf{u} \in \mathbb{S}^{p_1-1} \\ \mathbf{v} \in \mathbb{S}^{p_2-1}}} \frac{1}{n}\sum_{i=1}^n y_i\langle \mathbf{u}\mathbf{v}^\top, \mathbf{X}_i \rangle.$$

In particular, we let

$$Z(a,b) = \sup_{\substack{\mathbf{u} \in a\mathbb{S}^{p_1-1} \\ \mathbf{v} \in b\mathbb{S}^{p_2-1}}} \frac{1}{n}\sum_{i=1}^n y_i\langle \mathbf{u}\mathbf{v}^\top, \mathbf{X}_i \rangle.$$

We thus have $Z(a, b) = ab Z(1, 1)$. We construct $1/4$-covering sets of $\mathbb{S}^{p_1-1}$ and $\mathbb{S}^{p_2-1}$, which we denote as $\mathcal{N}_1$ and $\mathcal{N}_2$ respectively. Therefore, for any $\boldsymbol{u} \in \mathbb{S}^{p-1}, \boldsymbol{v} \in \mathbb{S}^{p_2-1}$, we can always find $\mathbf{u}' \in \mathcal{N}_1, \mathbf{v}' \in \mathcal{N}_2$ such that $\|\mathbf{u} - \mathbf{u}'\|_2 \leq 1/4, \|\mathbf{v} - \mathbf{v}'\|_2 \leq 1/4$. Moreover, we have the following decomposition $\mathbf{u}\mathbf{v}^\top = \mathbf{u}'\mathbf{v}'^\top + (\mathbf{u} - \mathbf{u}')\mathbf{v}'^\top + \mathbf{u}'(\mathbf{v} - \mathbf{v}')^\top + (\mathbf{u} - \mathbf{u}')(\mathbf{v} - \mathbf{v}')^\top$. Therefore, we have

$$Z(1, 1) \leq \max_{\mathbf{u} \in \mathcal{N}_1, \mathbf{v} \in \mathcal{N}_2} \frac{1}{n} \sum_{i=1}^{n} y_i \langle \mathbf{u}\mathbf{v}^\top, \mathbf{X}_i \rangle + Z(1/4, 1) + Z(1/4, 1) + Z(1/4, 1/4),$$

which implies that

$$Z(1, 1) \leq \frac{16}{7} \max_{\mathbf{u} \in \mathcal{N}_1, \mathbf{v} \in \mathcal{N}_2} \frac{1}{n} \sum_{i=1}^{n} y_i \langle \mathbf{u}\mathbf{v}^\top, \mathbf{X}_i \rangle.$$

For any fixed $\mathbf{u}$ and $\mathbf{v}$, $\{y_i \langle \mathbf{u}\mathbf{v}^\top, \mathbf{X}_i \rangle\}_{i=1}^{n}$ are $n$ independent copies of random variable $Y \langle \mathbf{u}\mathbf{v}^\top, X \rangle$ where $Y$ is sub-Gaussian with norm $\|Y\|_{\psi_2} \lesssim \sqrt{\|\mathbf{\Gamma}^*\|_F^2 + \sigma^2}$, $\langle \mathbf{u}\mathbf{v}^\top, X \rangle$ is zero mean Gaussian with variance 1. Following Lemma 22, $Y \langle \mathbf{u}\mathbf{v}^\top, X \rangle$ is sub-exponential with norm $\|Y \langle \mathbf{u}\mathbf{v}^\top, X \rangle\|_{\psi_1} \lesssim \sqrt{\|\mathbf{\Gamma}^*\|_F^2 + \sigma^2}$. Using concentration result in Lemma 20, we have

$$\Pr\left( \left| \frac{1}{n} \sum_{i=1}^{n} y_i \langle \mathbf{u}\mathbf{v}^\top, \mathbf{X}_i \rangle \right| \geq t \right) \leq 2 \exp\left( -\frac{C t^2 n}{\|\mathbf{\Gamma}^*\|_F^2 + \sigma^2} \right)$$

for sufficiently small $t > 0$. Note that $|\mathcal{N}_1| \leq 9^{p_1}, |\mathcal{N}_2| \leq 9^{p_2}$. By applying union bounds over $\mathcal{N}_1$ and $\mathcal{N}_2$, we have

$$\Pr\left( \max_{\mathbf{u} \in \mathcal{N}_1, \mathbf{v} \in \mathcal{N}_2} \frac{1}{n} \sum_{i=1}^{n} y_i \langle \mathbf{u}\mathbf{v}^\top, \mathbf{X}_i \rangle \geq t \right) \leq 2 \cdot 9^{(p_1+p_2)} \exp\left( -\frac{C t^2 n}{\|\mathbf{\Gamma}^*\|_F^2 + \sigma^2} \right).$$

By setting the right hand side to be $\delta/3$, we have that if $n \geq C(p_1 + p_2 + \log(6/\delta))$ for sufficiently large $C$, then

$$U_1 \lesssim (\|\mathbf{\Gamma}^*\|_F + \sigma) \sqrt{\frac{p_1 + p_2 + \log(6/\delta)}{n}} \tag{B.24}$$

with probability at least $1 - \delta/3$.

**Term $U_2$.** Parallel to the analysis of term $U_1$, we have

$$U_2 = \sup_{\substack{\mathbf{u} \in \mathbb{S}^{p_1-1} \\ \mathbf{v} \in \mathbb{S}^{p_2-1}}} \langle \mathbf{u}\mathbf{v}^\top, \mathbf{\Gamma}^* \rangle - \frac{1}{n} \sum_{i=1}^{n} \langle \mathbf{X}_i, \mathbf{\Gamma}^* \rangle \cdot \langle \mathbf{u}\mathbf{v}^\top, \mathbf{X}_i \rangle.$$

We construct $1/4$-nets $\mathcal{N}_1, \mathcal{N}_2$ of $\mathbb{S}^{p_1-1}$ and $\mathbb{S}^{p_2-1}$ respectively. Then

$$U_2 \leq \frac{16}{7} \max_{\mathbf{u} \in \mathcal{N}_1, \mathbf{v} \in \mathcal{N}_2} \langle \mathbf{u}\mathbf{v}^\top, \mathbf{\Gamma}^* \rangle - \frac{1}{n} \sum_{i=1}^{n} \langle \mathbf{X}_i, \mathbf{\Gamma}^* \rangle \cdot \langle \mathbf{u}\mathbf{v}^\top, \mathbf{X}_i \rangle.$$

For any fixed $\mathbf{u}, \mathbf{v}$, note that $\{\langle \mathbf{X}_i, \mathbf{\Gamma}^* \rangle \cdot \langle \mathbf{u}\mathbf{v}^\top, \mathbf{X}_i \rangle\}_{i=1}^{n}$ are $n$ independent samples of random variable $\langle X, \mathbf{\Gamma}^* \rangle \cdot \langle \mathbf{u}\mathbf{v}^\top, X \rangle$ where $\langle X, \mathbf{\Gamma}^* \rangle \sim \mathcal{N}(0, \|\mathbf{\Gamma}^*\|_F^2)$ and $\langle \mathbf{u}\mathbf{v}^\top, X \rangle \sim \mathcal{N}(0, 1)$. So $\langle X, \mathbf{\Gamma}^* \rangle \cdot \langle \mathbf{u}\mathbf{v}^\top, X \rangle$ is sub-exponential with norm $O(\|\mathbf{\Gamma}^*\|_F)$. Using the centering argument (Lemma 21) and concentration result (Lemma 20), we have

$$\Pr\left( \left| \langle \mathbf{u}\mathbf{v}^\top, \mathbf{\Gamma}^* \rangle - \frac{1}{n} \sum_{i=1}^{n} \langle \mathbf{X}_i, \mathbf{\Gamma}^* \rangle \cdot \langle \mathbf{u}\mathbf{v}^\top, \mathbf{X}_i \rangle \right| \geq t \right) \leq 2 \cdot \exp\left( -C \frac{t^2 n}{\|\mathbf{\Gamma}^*\|_F^2} \right)$$

for sufficiently small $t$. Using the union bound over sets $\mathcal{N}_1, \mathcal{N}_2$, we conclude that when $n \geq C(p_1 + p_2 + \log(6/\delta))$ for sufficiently large $C$, we have

$$U_2 \lesssim \|\mathbf{\Gamma}^*\|_F \sqrt{\frac{p_1 + p_2 + \log(6/\delta)}{n}} \tag{B.25}$$

with probability at least $1 - \delta/3$.

**Term** $U_3$. We first have

$$U_3 = \sup_{\substack{\mathbf{u} \in \mathbb{S}^{p_1-1} \\ \mathbf{v} \in \mathbb{S}^{p_2-1}}} \frac{2}{n} \sum_{i=1}^n w \cdot y_i \langle \mathbf{uv}^\top, \mathbf{X}_i \rangle - 2\mathbb{E}\left[w \cdot Y \langle \mathbf{uv}^\top, X \rangle\right].$$

Similar to the analysis of the first two terms, by constructing $\mathcal{N}_1, \mathcal{N}_2$, we have

$$U_3 \leq \frac{16}{7} \max_{\mathbf{u} \in \mathcal{N}_1, \mathbf{v} \in \mathcal{N}_2} \frac{2}{n} \sum_{i=1}^n w \cdot y_i \langle \mathbf{uv}^\top, \mathbf{X}_i \rangle - 2\mathbb{E}\left[w \cdot Y \langle \mathbf{uv}^\top, X \rangle\right].$$

Note that $\{wy_i \langle \mathbf{uv}^\top, \mathbf{X}_i \rangle\}_{i=1}^n$ are $n$ independent samples of random variable $wY \langle \mathbf{uv}^\top, X \rangle$ where $\langle \mathbf{uv}^\top, X \rangle \sim \mathcal{N}(0,1)$ and $wY$ is sub-Gaussian with norm $\|wY\|_{\psi_2} \lesssim \sqrt{\|\mathbf{\Gamma}^*\|_F^2 + \sigma^2}$ since $|w| \leq 1$. We thus have $wY \langle \mathbf{uv}^\top, X \rangle$ is sub-exponential with norm $\|wY \langle \mathbf{uv}^\top, X \rangle\|_{\psi_1} \lesssim \sqrt{\|\mathbf{\Gamma}^*\|_F^2 + \sigma^2}$. Then following the similar steps in analyzing the first two terms, we reach the conclusion that

$$U_3 \lesssim (\|\mathbf{\Gamma}^*\|_F + \sigma) \sqrt{\frac{p_1 + p_2 + \log(6/\delta)}{n}} \tag{B.26}$$

with probability at least $1 - \delta/3$ when $n \gtrsim p_1 + p_2 + \log(6/\delta)$.

Putting (B.24), (B.25) and (B.26) together completes the proof. $\qquad\square$

Setting $\delta = 6\exp(-(p_1 + p_2))$ in Lemma 12 suggests that Condition 5 holds with parameters $(\Delta_n, r, \delta)$ where $\Delta_n \lesssim (\|\mathbf{\Gamma}^*\|_F + \delta)\sqrt{(p_1 + p_2)/n}$, $\delta = \exp(-(p_1 + p_2))$ and $r$ can be any positive number. Putting these pieces together leads to the following guarantee about low rank recovery.

*Proof of Corollary 3.* This result is parallel to Corollary 2 for sparse recovery thus can be proved similarly. We omit the details. $\qquad\square$

## B.3 Missing Covariate Regression

We now turn to missing covariate regression. We first reveal function $Q_n^{MCR}(\cdot|\cdot)$ and $Q^{MCR}(\cdot|\cdot)$. To ease notation, we introduce vector $\mathbf{z}_i \in \{0,1\}^p$ to indicate the positions of missing entries, i.e., $z_{i,j} = 1$ if $x_{i,j}$ is missing. In this example, the E step involves computing the distribution of missing entries given current parameter guess $\boldsymbol{\beta}$. Under Gaussian design $X \sim \mathcal{N}(\mathbf{0}, \mathbf{I}_p), W \sim \mathcal{N}(0, \sigma^2)$, given observed covariate entries $(\mathbf{1} - \mathbf{z}_i) \odot \mathbf{x}_i$ and $y_i$, the conditional mean vector of $\widetilde{\mathbf{x}}_i$ has form

$$\boldsymbol{\mu}_{\boldsymbol{\beta}}(y_i, \mathbf{z}_i, \mathbf{x}_i) := \mathbb{E}[\widetilde{\mathbf{x}}_i | \boldsymbol{\beta}, y_i, (\mathbf{1}-\mathbf{z}_i)\odot\mathbf{x}_i] = (\mathbf{1}-\mathbf{z}_i)\odot\mathbf{x}_i + \frac{y_i - \langle\boldsymbol{\beta}, (\mathbf{1}-\mathbf{z}_i)\odot\mathbf{x}_i\rangle}{\sigma^2 + \|\mathbf{z}_i \odot \boldsymbol{\beta}\|_2^2}\mathbf{z}_i\odot\boldsymbol{\beta}, \tag{B.27}$$

and the conditional correlation matrix of $\widetilde{\mathbf{x}}_i$ has form

$$\boldsymbol{\Sigma}_{\boldsymbol{\beta}}(y_i, \mathbf{z}_i, \mathbf{x}_i) := \mathbb{E}\left[\widetilde{\mathbf{x}}_i \widetilde{\mathbf{x}}_i^\top | \boldsymbol{\beta}, y_i, (\mathbf{1} - \mathbf{z}_i) \odot \mathbf{x}_i\right]$$

$$= \boldsymbol{\mu}_{\boldsymbol{\beta}}\boldsymbol{\mu}_{\boldsymbol{\beta}}^\top + \mathrm{diag}(\mathbf{z}_i) - \left(\frac{1}{\sigma^2 + \|\mathbf{z}_i \odot \boldsymbol{\beta}\|_2^2}\right)(\mathbf{z}_i \odot \boldsymbol{\beta})(\mathbf{z}_i \odot \boldsymbol{\beta})^\top. \tag{B.28}$$

Consequently, $Q_n(\cdot|\cdot)$ corresponds to

$$Q_n^{MCR}(\boldsymbol{\beta}'|\boldsymbol{\beta}) = \frac{1}{n} \sum_{i=1}^n \langle y_i \boldsymbol{\mu}_{\boldsymbol{\beta}}(y_i, \mathbf{z}_i, \mathbf{x}_i), \boldsymbol{\beta}' \rangle - \frac{1}{2}\boldsymbol{\beta}^\top \boldsymbol{\Sigma}_{\boldsymbol{\beta}}(y_i, \mathbf{z}_i, \mathbf{x}_i)\boldsymbol{\beta}. \tag{B.29}$$

We thus have that $Q^{MCR}(\cdot|\cdot)$ takes form

$$Q^{MCR}(\boldsymbol{\beta}'|\boldsymbol{\beta}) = \langle \mathbb{E}\left[Y\boldsymbol{\mu}_{\boldsymbol{\beta}}(Y, Z, X)\right], \boldsymbol{\beta}' \rangle - \frac{1}{2}\left\langle \mathbb{E}\left[\boldsymbol{\Sigma}_{\boldsymbol{\beta}}(Y, Z, X)\right], \boldsymbol{\beta}\boldsymbol{\beta}^\top \right\rangle. \tag{B.30}$$

In particular, we let $\overline{\boldsymbol{\Sigma}}_{\boldsymbol{\beta}} := \mathbb{E}\left[\boldsymbol{\Sigma}_{\boldsymbol{\beta}}(Y, Z, X)\right]$. We first present a key result that characterizes the spectral property of $\overline{\boldsymbol{\Sigma}}_{\boldsymbol{\beta}}$.

**Lemma 13.** *For $\overline{\boldsymbol{\Sigma}}_{\boldsymbol{\beta}}$, we have the following decomposition*

$$\overline{\boldsymbol{\Sigma}}_{\boldsymbol{\beta}} = \epsilon \mathbf{I}_p + \boldsymbol{\Sigma}_1 - \boldsymbol{\Sigma}_2,$$

*where*

$$\boldsymbol{\Sigma}_1 = \mathbb{E}\left\{[(\mathbf{1}-Z)\odot X + \nu Z \odot \boldsymbol{\beta}] \cdot [(\mathbf{1}-Z)\odot X + \nu Z \odot \boldsymbol{\beta}]^{\top}\right\},$$

$$\boldsymbol{\Sigma}_2 = \mathbb{E}\left[\frac{1}{\sigma^2 + \|Z\odot\boldsymbol{\beta}\|_2^2}(Z\odot\boldsymbol{\beta})(Z\odot\boldsymbol{\beta})^{\top}\right], \quad \nu = \frac{Y - \langle\boldsymbol{\beta}, (\mathbf{1}-Z)\odot X\rangle}{\sigma^2 + \|Z\odot\boldsymbol{\beta}\|_2^2}.$$

*Let $\zeta := (1+\omega)\rho$, we have*

$$\lambda_{min}(\boldsymbol{\Sigma}_1) \geq 1 - \epsilon - 2\zeta^2\sqrt{\epsilon}, \tag{B.31}$$

$$\lambda_{max}(\boldsymbol{\Sigma}_2) \leq \zeta^2\epsilon, \tag{B.32}$$

$$\lambda_{max}(\overline{\boldsymbol{\Sigma}}_{\boldsymbol{\beta}}) \leq 1 + 2\zeta^2\sqrt{\epsilon} + (1+\zeta^2)\zeta^2\epsilon. \tag{B.33}$$

*In particular, let $\boldsymbol{\beta} = \boldsymbol{\beta}^*$, we have $\overline{\boldsymbol{\Sigma}}_{\boldsymbol{\beta}^*} = \mathbf{I}_p$.*

*Proof.* The decomposition follows by taking expectation of (B.28). For $\boldsymbol{\Sigma}_1$, expanding the bracket leads to

$$\boldsymbol{\Sigma}_1 = (1-\epsilon)\mathbf{I}_p + \underbrace{\mathbb{E}\left\{\nu[(\mathbf{1}-Z)\odot X](Z\odot\boldsymbol{\beta})^{\top} + \nu(Z\odot\boldsymbol{\beta})[(\mathbf{1}-Z)\odot X]^{\top}\right\}}_{\mathbf{M}} + \underbrace{\mathbb{E}\left[\nu^2(Z\odot\boldsymbol{\beta})(Z\odot\boldsymbol{\beta})^{\top}\right]}_{\mathbf{N}}.$$

For term $\mathbf{M}$, consider its spectral norm. Since it is symmetric, we have

$$\|\mathbf{M}\|_2 = \sup_{\mathbf{u}\in\mathbb{S}^{p-1}} 2\left|\mathbb{E}\left[\nu\langle Z\odot\boldsymbol{\beta}, \mathbf{u}\rangle \cdot \langle(\mathbf{1}-Z)\odot X, \mathbf{u}\rangle\right]\right|$$

$$= 2\sup_{\mathbf{u}\in\mathbb{S}^{p-1}}\left|\mathbb{E}\left[\frac{1}{\sigma^2+\|Z\odot\boldsymbol{\beta}\|_2}\langle(\mathbf{1}-Z)\odot(\boldsymbol{\beta}^*-\boldsymbol{\beta}), \mathbf{u}\rangle \cdot \langle Z\odot\boldsymbol{\beta}, \mathbf{u}\rangle\right]\right|$$

$$\leq 2\frac{1}{\sigma^2}\mathbb{E}\left[\|(\mathbf{1}-Z)\odot(\boldsymbol{\beta}^*-\boldsymbol{\beta})\|_2\|Z\odot\boldsymbol{\beta}\|_2\right] \leq 2\frac{1}{\sigma^2}\sqrt{\mathbb{E}\left[\|(\mathbf{1}-Z)\odot(\boldsymbol{\beta}^*-\boldsymbol{\beta})\|_2^2 \cdot \|Z\odot\boldsymbol{\beta}\|_2^2\right]}$$

$$\leq 2\frac{1}{\sigma^2}\sqrt{\epsilon(1-\epsilon)}\|\boldsymbol{\beta}-\boldsymbol{\beta}^*\|_2\|\boldsymbol{\beta}\|_2 \leq 2\rho^2\omega(1+\omega)\sqrt{\epsilon(1-\epsilon)} \leq 2\zeta^2\sqrt{\epsilon}.$$

where the second equality follows by taking expectation of $X$ and Gaussian noise $W$, the last inequality follows from the definitions of $\omega, \rho$ given in Section B.3. Note that $\mathbf{N} \succeq \mathbf{0}$. Then the lower bound of $\lambda_{min}(\boldsymbol{\Sigma}_1)$ follows by using $\lambda_{min}(\boldsymbol{\Sigma}_1) \geq 1 - \epsilon - \|\mathbf{M}\|_2$. For $\boldsymbol{\Sigma}_2$, we have

$$\boldsymbol{\Sigma}_2 = \mathbb{E}\left[\frac{1}{\sigma^2+\|Z\odot\boldsymbol{\beta}\|_2^2}(Z\odot\boldsymbol{\beta})(Z\odot\boldsymbol{\beta})^{\top}\right] \preceq \frac{1}{\sigma^2}\left((\epsilon-\epsilon^2)\mathrm{diag}(\boldsymbol{\beta}\odot\boldsymbol{\beta}) + \epsilon^2\boldsymbol{\beta}\boldsymbol{\beta}^{\top}\right).$$

Therefore, $\lambda_{max}(\boldsymbol{\Sigma}_2) \leq \zeta^2\epsilon$. Note that

$$\mathbf{N} \preceq \frac{1}{\sigma^4}\mathbb{E}\left[(Y - \langle\boldsymbol{\beta}, (\mathbf{1}-Z)\odot X\rangle)^2(Z\odot\boldsymbol{\beta})(Z\odot\boldsymbol{\beta})^{\top}\right]$$

$$= \frac{1}{\sigma^4}\mathbb{E}\left[(\sigma^2 + \|\boldsymbol{\beta}^* - (\mathbf{1}-Z)\odot\boldsymbol{\beta}\|_2^2)(Z\odot\boldsymbol{\beta})(Z\odot\boldsymbol{\beta})^{\top}\right]$$

$$\preceq \frac{1}{\sigma^4}(\sigma^2 + \|\boldsymbol{\beta}^*\|_2^2 + \|\boldsymbol{\beta}-\boldsymbol{\beta}^*\|_2^2)\left((\epsilon-\epsilon^2)\mathrm{diag}(\boldsymbol{\beta}\odot\boldsymbol{\beta}) + \epsilon^2\boldsymbol{\beta}\boldsymbol{\beta}^{\top}\right).$$

We thus have $\lambda_{max}(\mathbf{N}) \leq \frac{1}{\sigma^4}(\sigma^2 + \|\boldsymbol{\beta}^*\|_2^2 + \|\boldsymbol{\beta}-\boldsymbol{\beta}^*\|_2^2)\epsilon\|\boldsymbol{\beta}\|_2^2 \leq (1+\zeta^2)\zeta^2\epsilon$. The corresponding bound for $\lambda_{max}(\overline{\boldsymbol{\Sigma}}_{\boldsymbol{\beta}})$ then follows from $\lambda_{max}(\overline{\boldsymbol{\Sigma}}_{\boldsymbol{\beta}}) \leq 1 + \lambda_{max}(\mathbf{M}) + \lambda_{max}(\mathbf{N})$.

When $\boldsymbol{\beta} = \boldsymbol{\beta}^*$, we have

$$\mathbb{E}_{X,W}(\nu^2) = \frac{\mathbb{E}_{X,W}\left[(\langle X, \boldsymbol{\beta}^*\rangle + W - \langle X, (\mathbf{1}-Z)\odot\boldsymbol{\beta}^*\rangle)^2\right]}{(\sigma^2 + \|Z\odot\boldsymbol{\beta}^*\|_2^2)^2} = \frac{1}{\sigma^2 + \|Z\odot\boldsymbol{\beta}^*\|_2^2}$$

and

$$\mathbb{E}_{X,W}(\nu(\mathbf{1}-Z)\odot X) = \frac{\mathbb{E}\left[(\langle X, \boldsymbol{\beta}^*\rangle + W - \langle X, (\mathbf{1}-Z)\odot\boldsymbol{\beta}^*\rangle)(\mathbf{1}-Z)\odot X\right]}{\sigma^2 + \|Z\odot\boldsymbol{\beta}^*\|_2^2}$$

$$= \frac{(\mathbf{1}-Z)\odot Z\odot\boldsymbol{\beta}^*}{\sigma^2 + \|Z\odot\boldsymbol{\beta}^*\|_2^2} = \mathbf{0}.$$

Therefore, $\mathbf{M} = \mathbf{0}$ and $\mathbf{N} = \boldsymbol{\Sigma}_2$. We thus have $\overline{\boldsymbol{\Sigma}}_{\boldsymbol{\beta}^*} = \epsilon\mathbf{I}_p + (1-\epsilon)\mathbf{I}_p = \mathbf{I}_p$. $\qquad\square$

We now turn to check technical conditions about $Q^{MCR}(\cdot|\cdot)$. First, $\mathcal{M}(\cdot)$ is self consistent as stated below.

**Lemma 14** (Self-consistency of MCR). *Consider missing covariate regression with parameter $\boldsymbol{\beta}^* \in \mathbb{R}^p$ and $Q^{MCR}(\cdot|\cdot)$ given in* (B.30). *We have*

$$\boldsymbol{\beta}^* = \arg \max_{\boldsymbol{\beta} \in \mathbb{R}^p} Q^{MCR}(\boldsymbol{\beta}|\boldsymbol{\beta}^*).$$

*Proof.* In this example

$$\mathcal{M}(\boldsymbol{\beta}^*) = \left(\mathbb{E}\left[\boldsymbol{\Sigma}_{\boldsymbol{\beta}^*}(Y, Z, X)\right]\right)^{-1} \mathbb{E}\left[Y \boldsymbol{\mu}_{\boldsymbol{\beta}^*}(Y, Z, X)\right].$$

Following Lemma 13, we have $\boldsymbol{\Sigma}_{\boldsymbol{\beta}^*}(Y, Z, X) = \mathbf{I}_p$. Meanwhile, we have

$$\begin{aligned}
\mathbb{E}\left[Y \boldsymbol{\mu}_{\boldsymbol{\beta}^*}(Y, Z, X)\right] &= \mathbb{E}\left[(\langle \boldsymbol{\beta}^*, X \rangle + W)\left((\mathbf{1} - Z) \odot X + \frac{\langle Z \odot \boldsymbol{\beta}^*, X \rangle + W}{\sigma^2 + \|Z \odot \boldsymbol{\beta}^*\|_2^2} Z \odot \boldsymbol{\beta}^*\right)\right] \\
&= \mathbb{E}\left[(\mathbf{1} - Z) \odot \boldsymbol{\beta}^* + Z \odot \boldsymbol{\beta}^*\right] = \boldsymbol{\beta}^*.
\end{aligned}$$

Thus $\mathcal{M}(\boldsymbol{\beta}^*) = \boldsymbol{\beta}^*$. $\qquad\qquad\square$

For our analysis, we define $\rho := \|\boldsymbol{\beta}^*\|_2/\sigma$ to be the *signal to noise ratio* and $\omega := r/\|\boldsymbol{\beta}^*\|_2$ to be the *relative contractivity radius*. Let

$$\zeta := (1 + \omega)\rho.$$

Recall that $\epsilon$ is the missing probability of every entry. The next result characterizes the smoothness and concavity of $Q^{MCR}(\cdot|\cdot)$.

**Lemma 15** (Smoothness and concavity of MCR). *Consider missing covariate regression with parameter $\boldsymbol{\beta}^* \in \mathbb{R}^p$ and $Q^{MCR}(\cdot|\cdot)$ given in* (B.30). *For any $\omega > 0$, we have that $Q^{MCR}(\cdot|\cdot)$ satisfies Condition 2 with parameters $(\gamma, \mu, \omega\|\boldsymbol{\beta}^*\|_2)$, where*

$$\gamma = 1, \quad \mu = 1 + 2\zeta^2\sqrt{\epsilon} + (1 + \zeta^2)\zeta^2\epsilon.$$

*Proof.* Following Lemma 13, we have $\overline{\boldsymbol{\Sigma}}_{\boldsymbol{\beta}^*} = \mathbf{I}_p$. Therefore, $Q^{MCR}(\cdot|\boldsymbol{\beta}^*)$ is 1-strongly concave. For any $\boldsymbol{\beta} \in \mathcal{B}(w\|\boldsymbol{\beta}^*\|; \boldsymbol{\beta}^*)$, following (B.33), we have that $Q^{MCR}(\cdot|\boldsymbol{\beta})$ is $\mu$-smooth with $\mu = 1 + 2\zeta^2\sqrt{\epsilon} + (1 + \zeta^2)\zeta^2\epsilon$. $\qquad\square$

We revisit the following result about the gradient stability from [1].

**Lemma 16** (Gradient stability of MCR). *Consider the missing covariate regression with $\boldsymbol{\beta}^* \in \mathbb{R}^p$ and $Q^{MCR}(\cdot|\cdot)$ given in* (B.30). *For any $\omega > 0, \rho > 0$, $Q^{MCR}(\cdot|\cdot)$ satisfies Condition 3 with parameter $(\tau, \omega\|\boldsymbol{\beta}^*\|_2)$ where*

$$\tau = \frac{\zeta^2 + 2\epsilon(1 + \zeta^2)^2}{1 + \zeta^2}.$$

*Proof.* See the proof of Corollary 6 in [1]. $\qquad\qquad\square$

Unlike the previous two models, we require an upper bound on the signal to noise ratio. This unusual constraint is in fact unavoidable, as pointed out in [10].

We now turn to validate the conditions on finite sample function $Q_n^{MCR}(\cdot|\cdot)$. In particular, we have the following two guarantees.

**Lemma 17** (RSC of MCR). *Consider missing covariate regression with any fixed parameter $\boldsymbol{\beta}^* \in \mathcal{B}_0(s; p)$ and $Q_n^{MCR}(\cdot|\cdot)$ given in* (B.29). *There exist constants $\{C_i\}_{i=0}^3$ such that if $\epsilon \leq C_0 \min\{1, \zeta^{-4}\}$ and $n \geq C_1(1 + \zeta)^8 s \log p$, then we have $Q_n^{MCR}(\cdot|\cdot)$ satisfies Condition 4 with parameters $(\gamma_n, \mathcal{S}, \overline{\mathcal{S}}, \omega\|\boldsymbol{\beta}^*\|_2, \delta)$, where*

$$\gamma_n = \frac{1}{9}, \quad (\mathcal{S}, \overline{\mathcal{S}}) = (supp(\boldsymbol{\beta}^*), supp(\boldsymbol{\beta}^*)), \quad \delta = C_2 \exp(-C_3 n(1 + \zeta)^{-8}).$$

*Proof.* In order to show $Q_n^{MCR}(\cdot|\boldsymbol{\beta})$ is $\gamma_n$-strongly concave over $\mathcal{C}(\mathcal{S}, \overline{\mathcal{S}}; \mathcal{R})$, since $Q_n^{MCR}(\cdot|\boldsymbol{\beta})$ is quadratic, it is then equivalent to show

$$\frac{1}{n}\sum_{i=1}^{n}\mathbf{u}^{\top}\boldsymbol{\Sigma}_{\boldsymbol{\beta}}(y_i, \mathbf{z}_i, \mathbf{x}_i)\mathbf{u} \geq \gamma_n\|\mathbf{u}\|_2^2$$

for all $\mathbf{u} \in \mathcal{C}(\mathcal{S}, \overline{\mathcal{S}}, \mathcal{R})$. Expanding $\boldsymbol{\Sigma}_{\boldsymbol{\beta}}$ gives us

$$\frac{1}{n}\sum_{i=1}^{n}\mathbf{u}^{\top}\boldsymbol{\Sigma}_{\boldsymbol{\beta}}(y_i, \mathbf{z}_i, \mathbf{x}_i)\mathbf{u} \geq \underbrace{\frac{1}{n}\sum_{i=1}^{n}\langle\boldsymbol{\mu}_{\boldsymbol{\beta}}(y_i, \mathbf{z}_i, \mathbf{x}_i), \mathbf{u}\rangle^2}_{L_1} - \underbrace{\frac{1}{n}\sum_{i=1}^{n}\left(\frac{1}{\sigma^2 + \|\mathbf{z}_i \odot \boldsymbol{\beta}\|_2^2}\right)\langle\mathbf{z}_i \odot \boldsymbol{\beta}, \mathbf{u}\rangle^2}_{L_2}.$$

We choose to bound each term using restricted eigenvalue argument in Lemma 23. To ease notation, we let $\nu := \frac{y_i - \langle(\mathbf{1}-\mathbf{z}_i)\odot\boldsymbol{\beta}, \mathbf{x}_i\rangle}{\sigma^2 + \|\mathbf{z}_i\odot\boldsymbol{\beta}\|_2^2}$.

**Term $L_1$.** Note that $\boldsymbol{\mu}_{\boldsymbol{\beta}}(y_i, \mathbf{z}_i, \mathbf{x}_i)$ are samples of $\boldsymbol{\mu}_{\boldsymbol{\beta}}(Y, Z, X)$ which is zero mean sub-Gaussian random vector with covariance matrix $\boldsymbol{\Sigma}_1$ given in Lemma 13. Moreover, we have $\lambda_{min}(\boldsymbol{\Sigma}_1) \geq 1 - \epsilon - 2\zeta^2\sqrt{\epsilon}$. By restricting $\epsilon \leq 1/4$ and assuming $\epsilon \leq C\zeta^{-4}$ for sufficiently small $C$, we have $\lambda_{min}(\boldsymbol{\Sigma}_1) \geq \frac{1}{2}$. Moreover

$$\|\boldsymbol{\mu}_{\boldsymbol{\beta}}(Y, Z, X)\|_{\psi_2} \lesssim \|(\mathbf{1}-Z)\odot X\|_{\psi_2} + \|\nu Z \odot \boldsymbol{\beta}\|_{\psi_2} \lesssim 1 + \|\nu Z \odot \boldsymbol{\beta}\|_{\psi_2}.$$

Note that $\|\nu Z \odot \boldsymbol{\beta}\|_{\psi_2} = \sup_{\mathbf{u}\in\mathbb{S}^{p-1}} \|\nu\langle Z\odot\boldsymbol{\beta}, \mathbf{u}\rangle\|_{\psi_2} \leq \|\boldsymbol{\beta}\|_2 \cdot \|\nu\|_{\psi_2} \leq \sigma^{-2}\|\boldsymbol{\beta}\|_2 \cdot \|W + \langle X, \boldsymbol{\beta}^* - (\mathbf{1}-Z)\odot\boldsymbol{\beta}\rangle\|\|_{\psi_2} \lesssim (1+\omega)\rho + (1+\omega)^2\rho^2$. As $\zeta := (1+\omega)\rho$. We thus have $\|\boldsymbol{\mu}_{\boldsymbol{\beta}}(Y, Z, X)\|_{\psi_2} \lesssim (1+\zeta)^2$. Using Lemma 23 with the substitution $\boldsymbol{\Sigma} = \boldsymbol{\Sigma}_1$ and $X = \boldsymbol{\mu}_{\boldsymbol{\beta}}(Y, Z, X)$, we claim that there exist constants $C_i$ such that

$$L_1 \geq \frac{1}{4}\|\mathbf{u}\|_2^2 - C_0(1+\zeta)^8\frac{\log p}{n}\|\mathbf{u}\|_1^2 \text{ for all } \mathbf{u} \in \mathbb{R}^p. \tag{B.34}$$

with probability at least $1 - C_1\exp(-C_2 n(1+\zeta)^{-8})$.

**Term $L_2$.** We now turn to term $L_2$. We introduce $n$ i.i.d. samples $\{p_i\}_{i=1}^n$ of Rademacher random variable $P$ with $\Pr(P=1) = \Pr(P=-1) = 1/2$. Equivalently, we have

$$L_2 = \frac{1}{n}\sum_{i=1}^{n}\frac{1}{\sigma^2 + \|\mathbf{z}_i \odot \boldsymbol{\beta}\|_2^2}\langle p_i\mathbf{z}_i \odot \boldsymbol{\beta}, \mathbf{u}\rangle^2.$$

Note that $\sqrt{(\sigma^2 + \|Z\odot\boldsymbol{\beta}\|_2^2)^{-1}}PZ\odot\boldsymbol{\beta}$ is zero mean sub-Gaussian random vector with covariance matrix $\boldsymbol{\Sigma}_2$ given in Lemma 13. Moreover, we have $\lambda_{max}(\boldsymbol{\Sigma}_2) \leq \zeta^2\epsilon \leq 1/12$, where the last inequality follows by letting $\epsilon \leq C\zeta^{-2}$ for sufficiently small $C$. Also note that

$$\left\|\sqrt{(\sigma^2 + \|Z\odot\boldsymbol{\beta}\|_2^2)^{-1}}PZ\odot\boldsymbol{\beta}\right\|_{\psi_2} \lesssim \sigma^{-1}\|Z\odot\boldsymbol{\beta}\|_{\psi_2} \lesssim \zeta.$$

Using Lemma 23 with substitution $\boldsymbol{\Sigma} = \boldsymbol{\Sigma}_2$ and $X = \sqrt{(\sigma^2 + \|Z\odot\boldsymbol{\beta}\|_2^2)^{-1}}PZ\odot\boldsymbol{\beta}$, we claim there exists constants $C_i'$ such that

$$L_2 \leq \frac{1}{8}\|\mathbf{u}\|_2^2 + C_0'\max\{\zeta^4, 1\}\frac{\log p}{n}\|\mathbf{u}\|_1^2, \text{ for all } \mathbf{u} \in \mathbb{R}^p. \tag{B.35}$$

with probability at least $1 - C_1'\exp(-C_2'n\min\{\zeta^{-4}, 1\})$.

Now we put (B.34) and (B.35) together. So we obtain

$$\frac{1}{n}\sum_{i=1}^{n}\mathbf{u}^{\top}\boldsymbol{\Sigma}_{\boldsymbol{\beta}}(y_i, \mathbf{z}_i, \mathbf{x}_i)\mathbf{u} \geq \frac{1}{8}\|\mathbf{u}\|_2^2 - (C_0 + C_0')(1+\zeta)^8\frac{\log p}{n}\|\mathbf{u}\|_1^2.$$

For any $\mathbf{u} \in \mathcal{C}(\mathcal{S}, \overline{\mathcal{S}}; \mathcal{R})$, we have $\|\mathbf{u}\|_1 \leq 5\sqrt{s}\|\mathbf{u}\|_2$. Consequently, when $n \geq C(1+\zeta)^8 s \log p$ for sufficiently large $C$, we have that, with high probability, $Q_n^{MCR}(\cdot|\boldsymbol{\beta})$ is $\gamma_n$-strongly concave over $\mathcal{C}$ with $\gamma_n = 1/9$. $\qquad\square$

**Lemma 18** (Statistical error of MCR). *Consider missing covariate regression with any fixed parameter $\boldsymbol{\beta}^* \in \mathcal{B}_0(s;p)$ and $Q_n^{MCR}(\cdot|\cdot)$ given in (B.29). There exist constants $C_0, C_1$ such that if $n \geq C_0[\log p + \log(24/\delta)]$, then for any $\delta \in (0,1)$ and any fixed $\boldsymbol{\beta} \in \mathcal{B}(\omega\|\boldsymbol{\beta}^*\|_2, \boldsymbol{\beta}^*)$, we have that for*

$$\|\nabla Q_n^{MCR}(\boldsymbol{\beta}^*|\boldsymbol{\beta}) - Q^{MCR}(\boldsymbol{\beta}^*|\boldsymbol{\beta})\|_\infty \leq C_1(1+\zeta)^5 \sigma \sqrt{\frac{\log p + \log(24/\delta)}{n}}$$

*with probability at least $1 - \delta$.*

*Proof.* In this example,

$$\|\nabla Q_n^{MCR}(\boldsymbol{\beta}^*|\boldsymbol{\beta}) - \nabla Q^{MCR}(\boldsymbol{\beta}^*|\boldsymbol{\beta})\|_{\mathcal{R}^*}$$
$$\leq \underbrace{\left\|\frac{1}{n}\sum_{i=1}^n y_i \boldsymbol{\mu}_{\boldsymbol{\beta}}(y_i, \mathbf{z}_i, \mathbf{x}_i) - \mathbb{E}\left[Y\boldsymbol{\mu}_{\boldsymbol{\beta}}(Y, Z, X)\right]\right\|_\infty}_{U_1} + \underbrace{\left\|\frac{1}{n}\sum_{i=1}^n \boldsymbol{\Sigma}_{\boldsymbol{\beta}}(y_i, \mathbf{z}_i, \mathbf{x}_i)\boldsymbol{\beta}^* - \mathbb{E}\left[\boldsymbol{\Sigma}_{\boldsymbol{\beta}}(Y, Z, X)\right]\boldsymbol{\beta}^*\right\|_\infty}_{U_2}.$$

To ease notation, we let $\nu := \frac{y_i - \langle (\mathbf{1}-\mathbf{z}_i)\odot\boldsymbol{\beta}, \mathbf{x}_i\rangle}{\sigma^2 + \|\mathbf{z}_i \odot \boldsymbol{\beta}\|_2^2}$. Next we bound the term $U_1$ and $U_2$ respectively.

**Term $U_1$.** Consider one coordinate of vector $V := Y\boldsymbol{\mu}_{\boldsymbol{\beta}}(Y, Z, X)$. For any $j \in [p]$, we have

$$V_j = Y[(1 - Z_j)X_j + \nu Z_j \beta_j].$$

So $V_j$ is sub-exponential random variable since $Y$ and $(1 - Z_j)X_j + \nu Z_j \beta_j$ are both sub-Gaussians. Moreover, we have $\|Y\|_{\psi_2} \lesssim \sigma + \|\boldsymbol{\beta}^*\|_2$ and $\|(1 - Z_j)X_j + \nu Z_j \beta_j\|_{\psi_2} \lesssim \|(1 - Z_j)X_j\|_{\psi_2} + \|\nu Z_j \boldsymbol{\beta}_j\|_{\psi_2} \lesssim 1 + \sigma^{-2}(\sigma + \sqrt{1 + \omega^2}\|\boldsymbol{\beta}^*\|_2)\|\boldsymbol{\beta}\|_2$. The last inequality follows from the fact that $\nu$ is sub-Gaussian with $\|\nu\|_{\psi_2} \lesssim \sigma^{-2}(\sigma + \sqrt{1 + \omega^2}\|\boldsymbol{\beta}^*\|_2)$. We have $\|V_i\|_{\psi_1} \lesssim \|Y\|_{\psi_2} \cdot \|(1 - Z_j)X_j + \nu Z_j \beta_j\|_{\psi_2} \lesssim (1 + \zeta)^3 \sigma$, where $\zeta := (1 + \omega)\rho$. By concentration result of sub-exponentials (Lemma 20) and applying union bound, we have that there exists constant $C$ such that for $t \lesssim (1 + \zeta)^3 \sigma$,

$$\Pr(U_1 \geq t) \leq pe \cdot \exp(-\frac{Cnt^2}{(1 + \zeta)^6 \sigma^2}).$$

Setting the right hand side to be $\delta/2$ implies that for $n \gtrsim \log p + \log(2e/\delta)$,

$$U_1 \lesssim (1 + \zeta)^3 \sigma \sqrt{\frac{\log p + \log(2e/\delta)}{n}} \tag{B.36}$$

with probability at least $1 - \delta/2$.

**Term $U_2$.** Term $U_2$ can be further decomposed into several terms as follows

$$U_2 \leq \|\mathbf{a}_1\|_\infty + \|\mathbf{a}_2\|_\infty + \|\mathbf{a}_3\|_\infty + \|\mathbf{a}_4\|_\infty + \sigma^{-2}\|\mathbf{a}_5\|_\infty + \|\mathbf{a}_6\|_\infty,$$

where

$$\mathbf{a}_1 = \frac{1}{n}\sum_{i=1}^n \langle(\mathbf{1} - \mathbf{z}_i)\odot\mathbf{x}_i, \boldsymbol{\beta}^*\rangle(\mathbf{1} - \mathbf{z}_i)\odot\mathbf{x}_i - \mathbb{E}\left[\langle(\mathbf{1} - Z)\odot X, \boldsymbol{\beta}^*\rangle(\mathbf{1} - Z)\odot X\right],$$

$$\mathbf{a}_2 = \frac{1}{n}\sum_{i=1}^n \langle\nu\mathbf{z}_i \odot \boldsymbol{\beta}, \boldsymbol{\beta}^*\rangle(\mathbf{1} - \mathbf{z}_i)\odot\mathbf{x}_i - \mathbb{E}\left[\langle\nu Z \odot \boldsymbol{\beta}, \boldsymbol{\beta}^*\rangle(\mathbf{1} - Z)\odot X\right],$$

$$\mathbf{a}_3 = \frac{1}{n}\sum_{i=1}^n \langle(\mathbf{1} - \mathbf{z}_i)\odot\mathbf{x}_i, \boldsymbol{\beta}^*\rangle\nu\mathbf{z}_i \odot \boldsymbol{\beta} - \mathbb{E}\left[\langle(\mathbf{1} - Z)\odot X, \boldsymbol{\beta}^*\rangle\nu Z \odot \boldsymbol{\beta}\right],$$

$$\mathbf{a}_4 = \frac{1}{n}\sum_{i=1}^n \nu^2\langle\mathbf{z}_i \odot \boldsymbol{\beta}, \boldsymbol{\beta}^*\rangle\mathbf{z}_i \odot \boldsymbol{\beta} - \mathbb{E}\left[\nu^2\langle Z \odot \boldsymbol{\beta}, \boldsymbol{\beta}^*\rangle Z \odot \boldsymbol{\beta}\right],$$

$$\mathbf{a}_5 = \frac{1}{n}\sum_{i=1}^n \langle\mathbf{z}_i \odot \boldsymbol{\beta}, \boldsymbol{\beta}^*\rangle\mathbf{z}_i \odot \boldsymbol{\beta} - \mathbb{E}\left[\langle Z \odot \boldsymbol{\beta}, \boldsymbol{\beta}^*\rangle Z \odot \boldsymbol{\beta}\right], \quad \mathbf{a}_6 = \frac{1}{n}\sum_{i=1}^n \text{diag}(\mathbf{z}_i)\boldsymbol{\beta}^* - \epsilon\boldsymbol{\beta}^*.$$

The key idea to bound the infinite norm of each term $\mathbf{a}_i$ is the same: showing that each coordinate is finite summation of independent sub-Gaussian (or sub-exponential) random variables and applying

concentration result and probabilistic union bound. For each term $\mathbf{a}_i, i = 1, 2, \ldots, 6$, we have that for any $j \in [p]$,

$$\|\langle (\mathbf{1} - Z) \odot X, \boldsymbol{\beta}^* \rangle (1 - Z_j) \odot X_j\|_{\psi_1} \lesssim \|\boldsymbol{\beta}^*\|_2, \quad \|\langle \nu Z \odot \boldsymbol{\beta}, \boldsymbol{\beta}^* \rangle (1 - Z_j) \odot X_j\|_{\psi_1} \lesssim \sigma(1 + \zeta)\zeta^2,$$

$$\|\langle (\mathbf{1} - Z) \odot X, \boldsymbol{\beta}^* \rangle \nu Z_j \beta_j\|_{\psi_1} \lesssim \sigma(1 + \zeta)\zeta^2, \quad \|\nu^2 \langle Z \odot \boldsymbol{\beta}, \boldsymbol{\beta}^* \rangle Z_j \beta_j\|_{\psi_1} \lesssim \sigma(1 + \zeta^2)\zeta^3,$$

$$\sigma^{-2}\|\langle Z \odot \boldsymbol{\beta}, \boldsymbol{\beta}^* \rangle Z_j \odot \beta_j\|_{\psi_2} \lesssim \sigma\zeta^3, \quad \|\epsilon \beta_j^*\|_{\psi_2} \lesssim \epsilon \|\boldsymbol{\beta}^*\|_\infty$$

respectively. For simplicity, we treat coordinates of every $\mathbf{a}_i$ as finite sum of sub-exponentials with $\psi_1$ norm $O(\sigma(1 + \zeta)^5)$. Consequently, by concentration result in Lemma 20, there exists constant $C$ such that

$$\Pr(U_2 \geq t) \leq 12p \cdot \exp\left(-\frac{Cnt^2}{\sigma^2(1 + \zeta)^{10}}\right)$$

for $t \lesssim \sigma(1 + \zeta)^5$. By setting the right hand side to be $\delta/2$ in the above inequality, we have that when $n \gtrsim \log p + \log(24/\delta)$,

$$U_2 \lesssim \sigma(1 + \zeta)^5 \sqrt{\frac{\log p + \log(24/\delta)}{n}}. \tag{B.37}$$

with probability at least $1 - \delta/2$.

Finally, putting (B.36) and (B.37) together completes the proof. $\qquad \square$

By setting $\delta = 1/p$ in Lemma 18 immediately implies that $Q_n^{MCR}$ satisfies Condition 5 with parameters $\Delta_n = O\left((1 + \zeta)^5 \sigma \sqrt{\log p/n}\right), r = \omega\|\boldsymbol{\beta}^*\|_2$ and $\delta = 1/p$.

Putting together all the pieces leads to the following guarantee about resampling version of regularized EM on missing covariate regression.

*Proof of Corollary 4.* Following Theorem 1, we have $\kappa^* = 5\frac{\alpha\mu\tau}{\gamma\gamma_{n/T}}$. For $\ell_2$ norm, $\alpha = 1$. Based on Lemma 17, we have $\gamma_n = 1/9$. Following Lemma 15 and 16, we have $\gamma = 1$ and can always find sufficiently small constants $C_0, C_1$ such that $\mu \leq 10/9$ and $\tau \leq 1/100$. We thus obtain $\kappa^* \leq 1/2$. From Lemma 18, one can check $\Delta > 3\Delta_{n/T}$ under suitable $C$. We choose $n/T \gtrsim \sigma^2(\omega\rho)^{-1} s \log p$ to make sure $\Delta \leq 3\overline{\Delta}$. With these conditions in hand, direct applying Theorem 1 completes the proof. $\qquad \square$

## C Supporting Lemmas

**Lemma 19.** *Suppose $X_1, X_2, \ldots, X_n$ are $n$ i.i.d. centered sub-Gaussian random variables with Orlicz norm $\|X_1\|_{\psi_2} \leq K$. Then for every $t \geq 0$, we have*

$$\Pr\left(\left|\frac{1}{n}\sum_{i=1}^n X_i\right| \geq t\right) \leq e \cdot \exp\left(-\frac{Cnt^2}{K^2}\right),$$

*where $C$ is an absolute constant.*

*Proof.* See the proof of Proposition 5.10 in [18]. $\qquad \square$

**Lemma 20.** *Suppose $X_1, X_2, \ldots, X_n$ are $n$ i.i.d. centered sub-exponential random variables with Orlicz norm $\|X_1\|_{\psi_1} \leq K$. Then for every $t > 0$, we have*

$$\Pr\left(\left|\frac{1}{n}\sum_{i=1}^n X_i\right| \geq t\right) \leq 2 \cdot \exp\left(-C \min\left\{\frac{t^2}{K^2}, \frac{t}{K}\right\} n\right),$$

*where $C$ is an absolute constant.*

*Proof.* See the proof of Corollary 5.7 in [18]. $\qquad \square$

**Lemma 21.** *Let $X$ be sub-Gaussian random variable and $Y$ be sub-exponential random variable. Then $X - \mathbb{E}[X]$ is also sub-Gaussian; $Y - \mathbb{E}[Y]$ is also sub-exponential. Moreover, we have*

$$\|X - \mathbb{E}[X]\|_{\psi_2} \leq 2\|X\|_{\psi_2}, \quad \|Y - \mathbb{E}[Y]\|_{\psi_1} \leq 2\|Y\|_{\psi_1}.$$

*Proof.* See Remark 5.18 in [18]. □

**Lemma 22.** *Let $X, Y$ be two sub-Gaussian random variables. Then $Z = X \cdot Y$ is sub-exponential random variable. Moreover, there exits constant $C$ such that*

$$\|Z\|_{\psi_1} \leq C\|X\|_{\psi_2} \cdot \|Y\|_{\psi_2}.$$

*Proof.* It follows from the basic properties. We omit the details. □

**Lemma 23.** *Let matrix $\mathbf{X}$ be an $n$-by-$p$ random matrix with i.i.d. rows drawn from $X$, which is zero mean sub-Gaussian random vector with $\|X\|_{\psi_2} \leq K$ and covariance matrix $\mathbf{\Sigma}$. We let $\lambda_1 := \lambda_{min}(\mathbf{\Sigma}), \lambda_p := \lambda_{max}(\mathbf{\Sigma})$.*

*(1) There exist constants $C_i$ such that*

$$\frac{1}{n}\|\mathbf{X}\mathbf{u}\|_2^2 \geq \frac{\lambda_1}{2}\|\mathbf{u}\|_2^2 - C_0\lambda_1 \max\left\{\frac{K^4}{\lambda_1^2}, 1\right\} \frac{\log p}{n}\|\mathbf{u}\|_1^2, \text{ for all } \mathbf{u} \in \mathbb{R}^p,$$

*with probability at least $1 - C_1 \exp\left(-C_2 n \min\left\{\frac{\lambda_1^2}{K^4}, 1\right\}\right)$.*

*(2) In Parallel, there exist constants $C_i'$ such that*

$$\frac{1}{n}\|\mathbf{X}\mathbf{u}\|_2^2 \leq \frac{3\lambda_p}{2}\|\mathbf{u}\|_2^2 + C_0'\lambda_p \max\left\{\frac{K^4}{\lambda_p^2}, 1\right\} \frac{\log p}{n}\|\mathbf{u}\|_1^2, \text{ for all } \mathbf{u} \in \mathbb{R}^p,$$

*with probability at least $1 - C_1' \exp\left(-C_2' n \min\left\{\frac{\lambda_p^2}{K^4}, 1\right\}\right)$.*

*Proof.* It follows by putting Lemma 12 and Lemma 15 in [9] together. □

**Lemma 24.** *Let $X_1$ and $X_2$ be independent random variables with distribution $\mathcal{N}(0, 1)$. For any positive constant $C > 0$, let event $\mathcal{E} := \{C \cdot |X_2| \geq |X_1|\}$. Then we have*

*(a)*

$$\mathbb{E}\left[|X_1| \,\middle|\, \mathcal{E}\right] \cdot \Pr(\mathcal{E}) = \sqrt{\frac{2}{\pi}}\left[1 - \sqrt{\frac{1}{C^2 + 1}}\right].$$

*(b)*

$$\mathbb{E}\left[|X_2| \,\middle|\, \mathcal{E}\right] \cdot \Pr(\mathcal{E}) = \sqrt{\frac{2}{\pi}}\frac{C}{\sqrt{1 + C^2}}.$$

*(c)*

$$\mathbb{E}\left[|X_1 X_2| \,\middle|\, \mathcal{E}\right] \cdot \Pr(\mathcal{E}) = \frac{2C^2}{\pi(1 + C^2)}.$$

*Proof.* (a)

$$\mathbb{E}\left[|X_1| \,\middle|\, \mathcal{E}\right] \cdot \Pr(\mathcal{E}) = 4 \cdot \int_0^\infty \int_0^{uC} \frac{1}{2\pi}\exp(-\frac{1}{2}v^2)\exp(-\frac{u^2}{2})v\,dv\,du = \sqrt{\frac{2}{\pi}}\left[1 - \sqrt{\frac{1}{C^2 + 1}}\right].$$

(b)

$$\mathbb{E}\left[|X_2| \,\middle|\, \mathcal{E}\right] \cdot \Pr(\mathcal{E}) = 4 \cdot \int_0^\infty \int_{v/C}^\infty \frac{1}{2\pi}\exp(-\frac{1}{2}v^2)\exp(-\frac{u^2}{2})u\,du\,dv = \sqrt{\frac{2}{\pi}}\frac{C}{\sqrt{1 + C^2}}.$$

(c)

$$\mathbb{E}\left[|X_1 X_2| \mid \mathcal{E}\right] \cdot \Pr(\mathcal{E}) = 4 \cdot \int_0^\infty \int_{v/C}^\infty \frac{1}{2\pi} \exp(-\frac{u^2}{2}) \exp(-\frac{v^2}{2}) uv\, du\, dv$$

$$= \frac{2}{\pi} \int_0^\infty \exp(-\frac{C^2+1}{2} v^2) v\, dv = \frac{2C^2}{\pi(1+C^2)}.$$

$\square$

**Lemma 25.** *Let $X \sim \mathcal{N}(0, \sigma^2)$ and $Z$ be Rademacher random variable taking values in $\{-1, 1\}$. Moreover, $X$ and $Z$ are independent. Function $f(x, z; a, \gamma)$ is defined as*

$$f(x, z; a, \gamma) = \frac{x + az}{1 + \exp(-\frac{2(1+\gamma)}{\sigma^2} a(x + az))}.$$

*Then for any $a \in \mathbb{R}, \gamma \in \mathbb{R}$, we have*

$$\left| \mathbb{E}\left[f(X, Z; a, \gamma)\right] - \frac{a}{2} \right| \leq \min\left\{ \frac{1}{2} |a\gamma| \exp(\frac{\gamma^2 a^2 - a^2}{2\sigma^2}), \; \frac{\sigma}{\sqrt{2\pi}} + |a| \right\}.$$

*In the special case $\gamma = 0$, we have $\mathbb{E}\left[f(X, Z; a, \gamma)\right] = a/2$.*

*Proof.* First note that

$$\mathbb{E}\left[f(X, Z; a, \gamma)\right] = \frac{1}{2}\mathbb{E}\left[ \frac{X + a}{1 + \exp(-\frac{2(1+\gamma)}{\sigma^2} a(X + a))} + \frac{X - a}{1 + \exp(-\frac{2(1+\gamma)}{\sigma^2} a(X - a))} \right]$$

$$= \frac{1}{2}\mathbb{E}\left[ \frac{X + a}{1 + \exp(-\frac{2(1+\gamma)}{\sigma^2} a(X + a))} + \frac{-X - a}{1 + \exp(-\frac{2(1+\gamma)}{\sigma^2} a(-X - a))} \right],$$

where the first equality is from taking expectation of $Z$, the second equality is from the fact that the distribution of $X$ is symmetric around 0. Let $X' = X + a$, then we have

$$\mathbb{E}\left[f(X, Z; a, \gamma)\right] = \frac{1}{2}\mathbb{E}\left[ \frac{X'}{1 + \exp(-\frac{2(1+\gamma)}{\sigma^2} aX')} + \frac{-X'}{1 + \exp(\frac{2(1+\gamma)}{\sigma^2} aX')} \right]$$

$$= \frac{1}{2}\mathbb{E}\left[ X' - 2\frac{\exp(-\frac{2(1+\gamma)}{\sigma^2} aX')X'}{1 + \exp(-\frac{2(1+\gamma)}{\sigma^2} aX')} \right].$$

Using $\mathbb{E}\left[X'\right] = a$, we have

$$\mathbb{E}\left[f(X, Z; a, \gamma)\right] - a/2 = \mathbb{E}\left[ -\frac{\exp(-\frac{2(1+\gamma)}{\sigma^2} aX')X'}{1 + \exp(-\frac{2(1+\gamma)}{\sigma^2} aX')} \right]$$

$$= \int_{-\infty}^\infty \frac{\exp(-\frac{(x-a)^2}{2\sigma^2})}{\sqrt{2\pi}\sigma} \frac{-\exp(-\frac{2(1+\gamma)}{\sigma^2} ax)x}{1 + \exp(-\frac{2(1+\gamma)}{\sigma^2} ax)} dx = \int_{-\infty}^\infty \frac{\exp(-\frac{x^2+a^2}{2\sigma^2})x}{\sqrt{2\pi}\sigma} \frac{-\exp(-\frac{\gamma ax}{\sigma^2})}{\exp(\frac{a(1+\gamma)x}{\sigma^2}) + \exp(\frac{-a(1+\gamma)x}{\sigma^2})} dx$$

$$= \int_0^\infty \frac{\exp(-\frac{x^2+a^2}{2\sigma^2})x}{\sqrt{2\pi}\sigma} \frac{\exp(\frac{\gamma ax}{\sigma^2}) - \exp(-\frac{\gamma ax}{\sigma^2})}{\exp(\frac{a(1+\gamma)x}{\sigma^2}) + \exp(\frac{-a(1+\gamma)x}{\sigma^2})} dx \qquad (\text{C.1})$$

When $a\gamma \geq 0$, we have $\mathbb{E}\left[f(X, Z; a, \gamma)\right] - a/2 \geq 0$. Under this setting, (C.1) yields that

$$\mathbb{E}\left[f(X, Z; a, \gamma)\right] - a/2 \leq \int_0^\infty \frac{\exp(-\frac{x^2+a^2}{2\sigma^2})x}{2\sqrt{2\pi}\sigma} \left[ \exp(\frac{\gamma ax}{\sigma^2}) - \exp(-\frac{\gamma ax}{\sigma^2}) \right] dx$$

$$= \frac{1}{2} \exp(\frac{\gamma^2 a^2 - a^2}{2\sigma^2}) \int_0^\infty \frac{1}{\sqrt{2\pi}\sigma} \left[ \exp\left(-\frac{(x-\gamma a)^2}{2\sigma^2}\right) - \exp\left(-\frac{(x+\gamma a)^2}{2\sigma^2}\right) \right] x\, dx$$

$$= \frac{1}{2} \exp(\frac{\gamma^2 a^2 - a^2}{2\sigma^2}) \int_{-\infty}^\infty \frac{1}{\sqrt{2\pi}\sigma} \exp\left(-\frac{(x-\gamma a)^2}{2\sigma^2}\right) x\, dx = \frac{1}{2} \exp(\frac{\gamma^2 a^2 - a^2}{2\sigma^2}) \gamma a,$$

| | GMM | MLR(sparse) | MLR(low rank) | MCR |
|---|---|---|---|---|
| $\Delta$ | $0.1(\|\boldsymbol{\beta}^*\|_\infty + \sigma)\sqrt{\frac{\log p}{n}}$ | $0.1(\|\boldsymbol{\beta}^*\|_2 + \sigma)\sqrt{\frac{\log p}{n}}$ | $0.01(\|\boldsymbol{\Gamma}^*\|_F + \sigma)\sqrt{\frac{p_1+p_2}{n}}$ | $0.2\sigma\sqrt{\frac{\log p}{n}}$ |

Table 1: Choice of parameter $\Delta$ in Algorithm 1.

where the first inequality follows from the fact that $x + 1/x \geq 2$ for any $x > 0$, the second equality is from

$$-\int_0^\infty \exp\left(-\frac{(x+\gamma a)^2}{2\sigma^2}\right)x\mathrm{d}x = \int_{-\infty}^0 \exp\left(-\frac{(x-\gamma a)^2}{2\sigma^2}\right)x\mathrm{d}x.$$

When $a\gamma \leq 0$, using similar proof, we have $\frac{1}{2}\exp(\frac{\gamma^2 a^2 - a^2}{2\sigma^2})\gamma a \leq \mathbb{E}\left[f(X,Z;a,\gamma)\right] - a/2 \leq 0$. Combining the two cases, we prove that

$$\left|\mathbb{E}\left[f(X,Z;a,\gamma)\right] - a/2\right| \leq \frac{1}{2}|a\gamma|\exp(\frac{\gamma^2 a^2 - a^2}{2\sigma^2}). \tag{C.2}$$

In the special case when $\gamma = 0$, we thus have $\mathbb{E}(f(X,Z;a,\gamma)) = a/2$.

Note that when $a\gamma \geq 0$, (C.1) also implies that

$$\mathbb{E}\left[f(X,Z;a,\gamma)\right] - a/2 \leq \int_0^\infty \frac{\exp(-\frac{x^2+a^2}{2\sigma^2})x}{\sqrt{2\pi}\sigma}\frac{\exp(\frac{\gamma ax}{\sigma^2})}{\exp(\frac{a(1+\gamma)x}{\sigma^2})}\mathrm{d}x = \int_0^\infty \frac{\exp(-\frac{(x+a)^2}{2\sigma^2})x}{\sqrt{2\pi}\sigma}\mathrm{d}x$$

$$= \int_0^\infty \frac{\exp(-\frac{(x+a)^2}{2\sigma^2})(x+a)}{\sqrt{2\pi}\sigma}\mathrm{d}x - \int_0^\infty \frac{\exp(-\frac{(x+a)^2}{2\sigma^2})a}{\sqrt{2\pi}\sigma}\mathrm{d}x \leq \frac{\sigma}{\sqrt{2\pi}} + |a|.$$

Similarly, when $a\gamma \leq 0$, we have

$$\mathbb{E}\left[f(X,Z;a,\gamma)\right] - a/2 \geq \int_0^\infty \frac{\exp(-\frac{x^2+a^2}{2\sigma^2})x}{\sqrt{2\pi}\sigma}\frac{-\exp(\frac{-\gamma ax}{\sigma^2})}{\exp(\frac{-a(1+\gamma)x}{\sigma^2})}\mathrm{d}x = -\int_0^\infty \frac{\exp(-\frac{(x-a)^2}{2\sigma^2})x}{\sqrt{2\pi}\sigma}\mathrm{d}x$$

$$= -\int_0^\infty \frac{\exp(-\frac{(x-a)^2}{2\sigma^2})(x-a)}{\sqrt{2\pi}\sigma}\mathrm{d}x - \int_0^\infty \frac{\exp(-\frac{(x-a)^2}{2\sigma^2})a}{\sqrt{2\pi}\sigma}\mathrm{d}x \geq -\frac{\sigma}{\sqrt{2\pi}} - |a|.$$

Therefore, we have that

$$\left|\mathbb{E}\left[f(X,Z;a,\gamma)\right] - a/2\right| \leq \frac{\sigma}{\sqrt{2\pi}} + |a|. \tag{C.3}$$

Putting (C.2) and (C.3) together completes the proof. $\qquad\square$

# D   Additional Experiment Setting

In our simulations, parameter $\Delta$ for each model is set according to Table 1.