[Reviews · NeurIPS 2015]

Submitted by Assigned_Reviewer_1

This work addresses the local convergence properties of the EM algorithm in the high-dimensional setting where the number of dimensions is greater than the sample size. The paper builds on a recent work for the low-dimensional setting. The main novelty here is an iterative regularization scheme that shrinks the regularization parameter as the algorithm gets closer to the optimum. This allows the algorithm to converge at a linear rate in the local neighborhood of the optimum. There are several strong conditions on the objective function that the results rely on. As in most theoretical results on EM-type optimization, the algorithm also requires fresh samples at each step.

As this is a "light" review, I was unable to verify the details of this very technical contribution. If the results are correct, they represent a relevant and solid contribution to the theory of EM. The paper will primarily appeal to experts in this area.

Summary: A solid (albeit very technical) contribution on the theory of EM for high-dimensional estimation.

Submitted by Assigned_Reviewer_2

A framework for regularized EM algorithms is proposed and analyzed. The motivation of the regularization is to allow for estimation in high dimensional settings under structural assumptions, where unconstrained estimation is statistically impossible and where ordinary EM may not even be well defined. Key is a regularized EM algorithm, where the ordinary M step is replaced by maximizing the Q function minus a regularizer on the model parameter, scaled by a regularization coefficient.

This is a refinement of previous work along similar lines. The novel contributions are as follows. First, the previous work only considered sparsity as the structural assumption, whereas the present paper allows for more general regularizers, which in particular apply to sparse and low rank structures. The assumptions on the statistical properties of the estimator of the Q function are changed in accordance to this more general regularization framework. Second, a specific choice of the regularization coefficient is given, which is updated at each iteration, and which properly controls the optimization error in relation to the statistical estimation error.

It would be helpful to provide some discussion or examples of the decomposability condition, the subspace compatibility constant, the norm ||_{R^*} (which is used in the key statistical condition 5), and the set C(S,\bar S, R).
Summary: This contribution generalizes and refines previous work on EM algorithms in high dimensions under structural assumptions. The novel results are significant for practical applicability of the theory.

Submitted by Assigned_Reviewer_3

In the EM algorithm for high dimensional setting, the M step is problematic, which needs to be addressed by exploiting the structure of the high-dim paramters. The paper considers to use decompposable regularizers. The main contribution is a way to set the regularizer in the iterative process of EM. It is shown that this leads to local linear convergence, under conditions about the population and empirical estimation of log likelihood using current estimation of the parameters (the basic quantity in EM).

The EM algorithm is used widely in machine learning and it is thus an important topic to identify conditions under which it leads to provable guarantees. The paper gives a priciple way to regularize the EM which is particularly interesting for the high dimensional setting. It further demonstrates the application of the theory to three well known models. The result is interesting and important.

--In the abstract: the authors claim that regularzing the M-step using the state of the art prescriptions ([19]) is not guaranteed to provide the desired bound, but this needs more illustration.

--Condition 5: as noted by the authors, this replaces the term that appears in previous work. It seems to me that this condition is the key to make the algorithm work in the high-dim region, while the analysis follows the previous work.

minor: --line 80: Y Z should be bold face --Eqn (2.2): "f" is missing --line 122: | |_R* is not yet defined here and thus the sentence is a bit confusing.
Summary: This paper studies the regularized EM algorithm. It proposes a way to set the regularizer, and identifies conditions under which the regularization leads to local linear convergence.

Submitted by Assigned_Reviewer_4

The paper introduces the difficulty of EM-algorithm in high-dimensional setting, where M-step can be unsolvable or unstable. Then regularization method is proposed, the key to which seems to be finding a good sequence of regularization coefficients to control the statistical error. Convergence is proved given condition on the initialization, regularization penalty and the likelihood. 3 latent variable models are then analyzed and some simulation result is provided.

Overall it is a good paper which is well motivated with nice theory.

A couple of thoughts: (1) more discussion about the effects of the initialization parameter $\lambda_n^{(0)}$ and contractive factor $\kappa$ can be helpful (2) the theory actually deals with algorithm 2, it might be good to clarify how the theory is linked to algorithm 1. It may also be interesting to show some simulation result for algorithm 2 (3) not sure about what figure 1 is trying to illustrate (probably that the algorithm does converge?). Also the font size for figures should be improved (4) some comparison (either in theory or in simulation) with competing methods (say [20]) can be helpful

Typo: abstract (e.g. a la[19]) equation (2.2) y_{\beta^*} should be f_{\beta^*}; \log_{\beta'} should be \log f_{\beta'}
Summary: Good paper that proposes solving high-dimensional latent variable problems using regularized EM algorithm with varying regularization coefficients. Theory and examples are well explained but simulation part can be improved

Author Feedback
Author rebuttal: We are very grateful to the careful and throughout reviews from all reviewers.

Reviewer_1
The reviewer comments that our novel results are both practically and theoretically important, and also suggests that we provide more discussion and examples on our technical conditions such as the decomposability condition, subspace compatibility constant etc. We have done so, though due to space limitations we have pushed examples illustrating our conditions to the supplementary material, in the section on proof ingredients for the results in section 5. In detail, for each model, we check and illustrate why those aforementioned technical conditions hold in every subsection of Section B. We will try to illustrate these comments in the main paper.

Reviewer_2
The reviewer suggests giving more illustration about why the existing theoretical framework for regularization does not work for inference with latent variables. Briefly, the reason is as follows. If attempting to do ML directly, one runs into the problem of non-convexity which is typical for latent variable models (and why people use EM), hence previous theory on regularization does not seem to apply. In EM, every step corresponds to a convex subproblem, but the loss function here is involved with a current estimation of a model parameter and thus is fundamentally different with standard high dimensional problems that do not have this iterative flavor. We agree this is an important point, and we will include a more explicit discussion in our final paper.

The reviewer is correct that condition 5 is key to our analysis. However, our analysis has significant differences with both standard high dimensional framework (e.g., [19]) and truncated EM algorithm in [20]. In fact, we extend the well known theory for regularization to an important nonconvex setting and overcome the limitations in [20]. Thanks for finding the typos -- these are now fixed.

Reviewer_3
Algorithm 2 is almost the same with algorithm 1 except that we use sample splitting, i.e., we split the whole sample set into several disjoint sets and use every single piece in each iteration of regularized EM. As we pointed out in our paper, the sample splitting does not seem necessary in practice, however, we need it to address one technical challenge in the analysis. That is to say, without sample splitting, the estimator after a few iterations of regularized EM will have a very complex probability dependence on the samples and thus is hard to analyze. Since algorithm 2 is not practical, we omit their experimental results. Based on our results offline, algorithm 2 has similar performance as algorithm 1 but has a slightly worse sample complexity (in theory, it uses an extra log factor because of its need for fresh batches of data).

Figure 1 is intended to show the change of overall estimation error, which is a combination of statistical error and optimization error, and optimization error. As we show in the theory, optimization error should decay at a linear rate to zero. The estimation error should decay to certain level (e.g., \sqrt{s\logp/n}), that is determined by the problem dimensions. This phenomenon is thus verified by figure 1. Experimental results will be enriched according to reviewer's comments.

Reviewer_4
The reviewer comments that our results are important while the details are technical. Indeed this is true, but as at a high level, our key result is theorem 1 that does not require very technical proofs, we also try to illustrate the key ideas conceptually.

Reviewer_6
Our work is in line of [1] and [20]. However, the analytical framework we propose has significant differences. Our conditions 4 and 5 are new to these previous works. Also by using them, we extend the standard regularization analysis to an important nonconvex setting, i.e., parameter estimation with latent variables.

Reviewer_7
We believe it's true that there are many (heuristic) ways to deal with latent variable models and even its high dimensional setting in practice. However, as far as we know, the provable theoretical guarantees are far less understood. Also, while regularization is well studied for high dimensional estimation problems, our setting is fundamentally different for two basic reasons. First, the overall ML problem is non-convex; second, when using EM, where every step corresponds to a convex subproblem, which has an iterative nature. It is this iterative aspect that is the key technical challenge to analysis, choice of regularization parameters, and to which past theory does not provide an answer.

It is mentioned that CV (cross validation) is usually applied for regularization algorithms. It's worth to note our work provides theoretical choices for algorithm parameters. Our guarantees are on the big O level and ignore constants. Roughly speaking, CV can be regarded as a method used in practice to figure out which constant is better for obtaining good empirical performance.